# Pesticide residues alter taxonomic and functional biodiversity in soils

J. Köninger[1,11], M. Labouyrie[2,3,4,11], C. Ballabio[2], O. Dulya[5,6], V. Mikryukov[5,6], F. Romero[3,4], A. Franco[2], M. Bahram[7,8], P. Panagos[2], A. Jones[2], L. Tedersoo[6,9], A. Orgiazzi[10,12 ✉], M. J. I. Briones[1,12 ✉] & M. G. A. van der Heijden[3,4,12 ✉]

Pesticides are widely distributed in soils[1–3], yet their effects on soil biodiversity remain poorly understood[4–7]. Here we examined the effects of 63 pesticides on soil archaea, bacteria, fungi, protists, nematodes, arthropods and key functional gene groups across 373 sites spanning woodlands, grasslands and croplands in 26 European countries. Pesticide residues were detected in 70% of sites and emerged as the second strongest driver of soil biodiversity patterns after soil properties. Our analysis further revealed organism- and function-specific patterns, emphasizing complex and widespread non-target effects on soil biodiversity. Pesticides altered microbial functions, including phosphorus and nitrogen cycling, and suppressed beneficial taxa, including arbuscular mycorrhizal fungi and bacterivore nematodes. Our findings highlight the need to integrate functional and taxonomic characteristics into future risk assessment methodology to safeguard soil biodiversity, a cornerstone of ecosystem functioning.

Belowground life is essential for maintaining critical ecosystem functions and services such as food production, carbon storage, erosion control and water regulation[8]. In addition to hosting nearly 59% of the Earth's biodiversity[9], soils also act as sinks for contaminants, such as pesticides applied aboveground[3]. These pesticides can persist in soils for extended periods[10], depending on their chemical properties[4] and soil adsorption and absorption capacities[11]. A recent pan-European study, LUCAS (Land Use-Land Cover Area Frame Survey) Soil 2018, detected at least one pesticide residue in 87% of the 3,473 sites investigated, with 46% of these sites having pesticide concentrations exceeding 0.05 mg kg$^{-1}$ (ref. 1).

Several studies have demonstrated the negative effects of pesticides on biodiversity aboveground, particularly birds[12,13] and bees[14–16] and other insect taxa[17,18]. By contrast, similar assessments of pesticide effects on soil communities remain scarce, despite the critical role of soil organisms in ecosystem functions, including pesticide degradation[19,20]. For example, previous research has shown that pesticides negatively affected the abundance and diversity of non-target soil organisms, including soil invertebrates[21], such as earthworms[22], and arbuscular mycorrhizal fungi (AMF)[23], whereas others have found positive effects on pesticide degraders[7]. However, these studies have been spatially limited by focusing on specific countries[23,24] and agroecosystems[5,23,24], selected soil biota[6,24–26], and by including a very limited number of pesticide compounds[27]. Therefore, the effects of multiple pesticides on complex soil communities at large geographical scales and across different ecosystem types have not been addressed[28], but are crucially needed to better assess biodiversity under pesticide pressure.

This knowledge gap is due to the fact that little quantitative information on pesticide usage, doses, frequency and residues remaining in soils is currently available[21]. Furthermore, regulatory assessments primarily focus on single substances tested on a few invertebrate species, such as single species of earthworms (*Eisenia fetida*), nematodes (*Caenorhabditis elegans*) and collembolans (*Folsomia candida*)[29], with specific endpoints such as mineralization and nitrogen transformation (for microbes, nitrate formation)[29], and do not consider a wide range of field conditions and the effects of long-term exposure[30]. As a result, the broader ecological impacts of pesticide use on soil life should be better represented in future risk assessments of regulations, moving towards a more holistic approach[30–32].

This study provides a continent-wide evaluation of the impacts of pesticide residues, their active ingredients and metabolites (hereafter pesticides), on soil biodiversity across Europe (the European Union and the UK). Given the documented role of pesticide concentrations in shaping soil biota at smaller scales[5,7,23], we examined their effects on archaea, bacteria, fungi, protists, nematodes and arthropods at the continental scale, using field data from 373 sites across several European landscapes (that is, annual croplands, permanent croplands, former croplands recently converted to grasslands, extensive grasslands and woodlands). Additionally, we evaluated the responses of nine functional groups of soil biota to pesticides on the basis of their ecological roles in soil functioning: archaeal nitrifiers, bacterial chemoheterotrophs, bacterial nitrogen-fixers (N-fixers), AMF, fungal plant pathogens, (animal and plant) parasitic protists and bacterivore and herbivore nematodes. Finally, we examined the biological responses

[1]Departamento de Ecología y Biología Animal, Universidade de Vigo, Vigo, Spain. [2]European Commission, Joint Research Centre (JRC), Ispra, Italy. [3]Department of Plant and Microbial Biology, University of Zurich, Zurich, Switzerland. [4]Plant Soil Interactions, Agroscope, Zurich, Switzerland. [5]Department of Botany, Institute of Ecology and Earth Sciences, University of Tartu, Tartu, Estonia. [6]Mycology and Microbiology Center, University of Tartu, Tartu, Estonia. [7]Department of Ecology, Swedish University of Agricultural Sciences, Uppsala, Sweden. [8]Department of Agroecology, Aarhus University, Slagelse, Denmark. [9]Department of Zoology, College of Science, King Saud University, Riyadh, Saudi Arabia. [10]European Dynamics, Brussels, Belgium. [11]These authors contributed equally: J. Köninger, M. Labouyrie. [12]These authors jointly supervised this work: A. Orgiazzi, M. J. I. Briones, M. G. A. van der Heijden. ✉e-mail: alberto.orgiazzi@gmail.com; mbriones@uvigo.gal; marcel.vanderheijden@botinst.uzh.ch

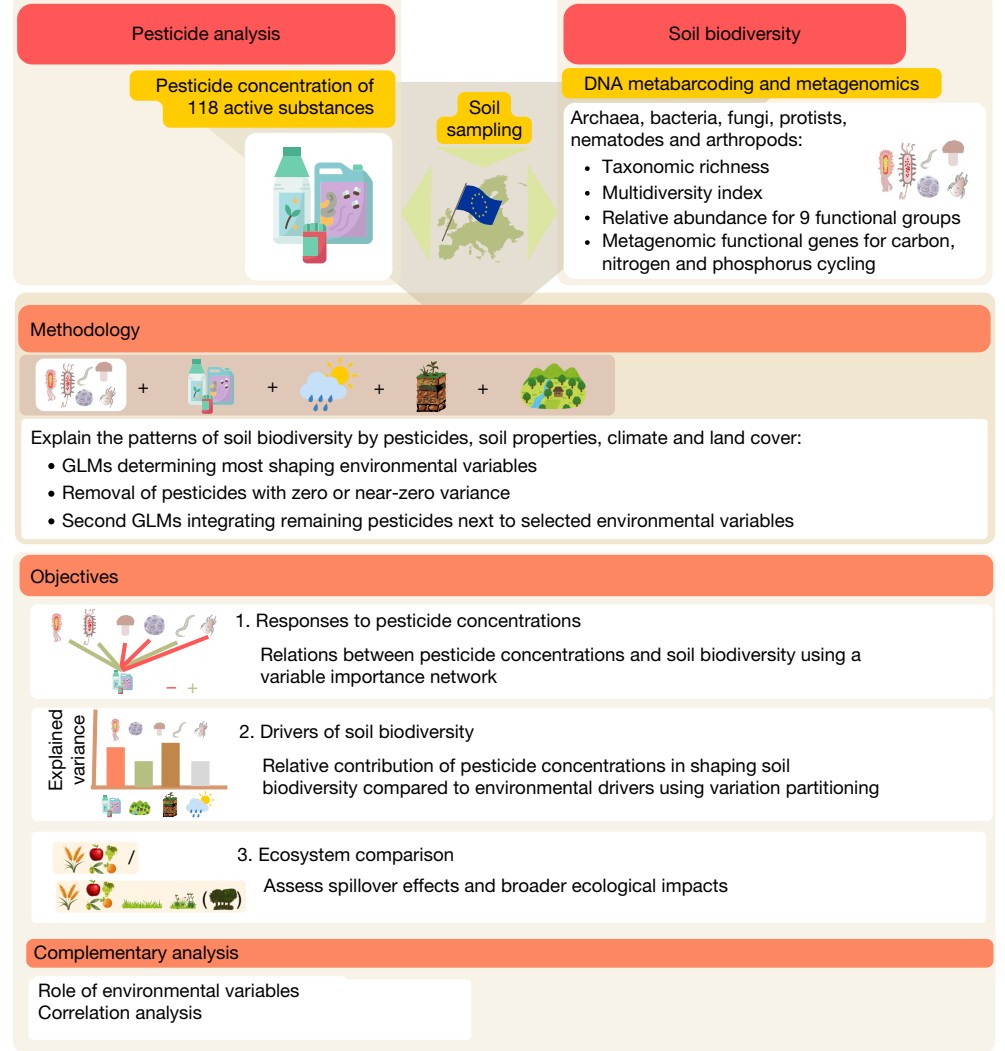

**Fig. 1 | Conceptual diagram to test the effects of pesticides on soil biodiversity.** Using the information derived from generalized linear models (GLMs), including pesticide concentrations, soil properties climate and ecosystem types, we investigated: (1) the effects of pesticides on soil biodiversity (taxonomic diversity, functional groups and functional genes); (2) the contribution of pesticides to soil biodiversity variation compared to environmental variables; and (3) spillover effects and broader ecological effects of pesticides comparing analysis carried out on croplands and all ecosystem types. Complementary analyses identified which environmental conditions could be related to the persistence of individual pesticides in soils among those compounds found to shape soil biodiversity (Kruskal–Wallis test). Correlation analyses were used to investigate the relationships between soil biodiversity and pesticide occurrence and risk.

to pesticides of 48 functional gene groups involved in the carbon (C), nitrogen (N) and phosphorus (P) cycles (for example, mineral nitrogen transformations regulating N loss through leaching and greenhouse gas emissions). All taxonomic and functional groups investigated in this study are hereafter collectively referred to as 'soil biodiversity'.

Our analyses focused primarily on cropland soils (including both annual and permanent crops), on which pesticides are predominantly applied[1,2,28,33]. We hypothesized that pesticides influence soil biodiversity, more so in these intensively managed ecosystems. To test this, we assessed the relationships between each pesticide concentration and: (1) the richness and diversity (Shannon index) of each taxonomic group; (2) their combined diversity (multidiversity); (3) the relative abundance of functional groups; and (4) the diversity of the functional gene groups (Fig. 1, objective 1). These analyses accounted for environmental drivers, including soil properties, climate and ecosystem type. We then quantified the relative importance of pesticide concentrations in shaping soil biodiversity compared to environmental drivers (Fig. 1, objective 2). Although croplands are the primary recipients of pesticide inputs, contamination can extend into surrounding ecosystems.

To evaluate the broader relevance and robustness of pesticide–soil biodiversity relationships, we conducted the same analyses including all ecosystem types (Supplementary Data 3) and compared them to those in croplands (Fig. 1, objective 3).

## Pesticide residues in soils

Across all five ecosystem types, a total number of 63 different pesticides were detected in European soils, with one or more pesticides being detected in 70% of the investigated sites (Fig. 2a). Ten out of the 63 detected pesticides had been discontinued for use in the EU in 2018 (that is, at the time of the survey[34]; Supplementary Table 1).

The majority (54%) of the pesticides detected were fungicides (Fig. 2b), followed by herbicides (34.9%) and insecticides (11.1%). The highest numbers of residues and cumulative pesticide concentration were found in annual and permanent croplands, followed by grasslands and woodlands (Fig. 2c and Extended Data Figs. 2a,b and 3). The most common pesticides detected were glyphosate (a herbicide) and its metabolite aminomethylphosphonic acid (AMPA), followed by boscalid

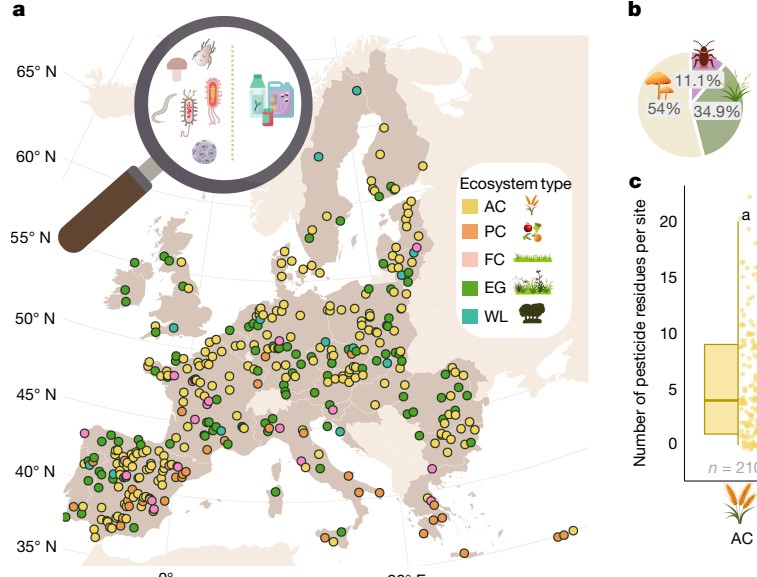

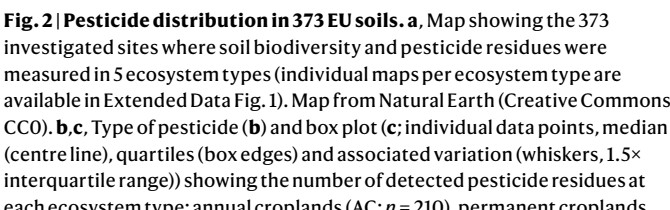

**Fig. 2 | Pesticide distribution in 373 EU soils. a**, Map showing the 373 investigated sites where soil biodiversity and pesticide residues were measured in 5 ecosystem types (individual maps per ecosystem type are available in Extended Data Fig. 1). Map from Natural Earth (Creative Commons CC0). **b,c**, Type of pesticide (**b**) and box plot (**c**; individual data points, median (centre line), quartiles (box edges) and associated variation (whiskers, 1.5× interquartile range)) showing the number of detected pesticide residues at each ecosystem type: annual croplands (AC; $n = 210$), permanent croplands (PC; $n = 34$), former croplands recently converted to grasslands (FC; $n = 19$), extensive grasslands (EG; $n = 97$) and woodlands (WL; $n = 13$). Different letters indicate significant differences (two-sided pairwise Wilcoxon multiple comparison test with Benjamini–Hochberg correction). AC versus PC, $P = 0.0048$; AC versus FC, $P = 0.0003$; AC versus EG, $P < 2 \times 10^{-16}$; AC versus WL, $P = 6.3 \times 10^{-6}$; PC versus FC, $P = 0.2781$; PC versus EG, $P = 0.0020$; PC versus WL, $P = 0.0059$; FC versus EG, $P = 0.0793$; FC versus WL, $P = 0.0249$; EG versus WL, $P = 0.0856$.

(a fungicide), pendimethalin (a herbicide) and epoxiconazole (a fungicide) (Extended Data Fig. 4).

## Soil biota responses to pesticides

Next, we assessed which of the 63 detected pesticide concentrations (alongside key environmental variables) have relevant impacts on richness and diversity of organism groups, multidiversity, relative abundance of each functional group and diversity of each functional gene group (gene orthologues per million reads (pmOGs)).

We found that the effects of pesticide concentrations in croplands (both annual and permanent crops) varied depending on organism taxonomical and functional group, and the pesticide involved (Fig. 3 and Extended Data Fig. 5). Fungi exhibited multiple negative associations, and their richness decreased in relation to four fungicides (boscalid, carbendazim, dimetomorph and fluopyram) and the herbicide diflufenican, whereas other groups displayed more variable patterns. Although multidiversity showed both positive and negative associations with pesticides, this integrative index (aggregating multiple organism groups) masked the complexity of pesticide–soil biodiversity relationships, overlooking distinct associations between individual organism types and a broader range of pesticides.

Increasing fungicide concentrations in croplands negatively related to several non-target organisms (Fig. 3 and Extended Data Fig. 5). Notably, bixafen concentrations were associated with a decrease in fungal plant pathogens, a reduction in the richness of protists, nematodes and arthropods, as well as a reduction in the diversity of archaea, bacteria and arthropods. Higher doses of carbendazim, fenpropidin and epoxiconazole reduced relative abundance of AMF.

Beneficial groups such as AMF and bacterivore nematodes were negatively correlated with higher concentrations of the herbicide pendimethalin, which, by contrast, promoted plant antagonists such as protist plant parasites (Fig. 3 and Extended Data Fig. 5). Other plant antagonists, including fungal plant pathogens and herbivore nematodes, were also positively affected with increasing concentrations of the herbicide glyphosate, whereas the richness of protists and nematodes, the diversity of fungi and arthropods, and the abundance of archaeal nitrifiers and bacterivore nematodes declined (Fig. 3 and Extended Data Fig. 5).

The most sensitive gene groups affected by pesticides (fungicides, herbicides and insecticides) in croplands were bacterial genes involved in the denitrification and chitin degradation (Fig. 3 and Extended Data Fig. 6). Seventy per cent of the observed significant effects on the diversity of bacterial gene groups and 84% of the effects on fungal gene groups were positive. For archaeal and faunal groups, around 50% of relations were negative (Extended Data Fig. 6). Further results on the effects of specific pesticides are presented in Supplementary Results 1.

## Pesticides as a driver of soil biota

We found that pesticide concentrations accounted for up to 29.5% of the explained variance in soil biodiversity community structure in croplands (Fig. 4 and Supplementary Data 2, Tables 1–6) and represented a larger portion of the variance of some soil biodiversity metrics than environmental drivers. Specifically, pesticide concentrations were the most important factor shaping fungal richness (uniquely explaining 12.3% of variance; Fig. 4 and Supplementary Data 2, Table 1). Additionally, the relative abundance of bacterial chemoheterotrophs, AMF, protist plant parasites, herbivore nematodes and bacterivore nematodes were also significantly influenced by pesticide concentrations (accounting for 10.4%, 11.6%, 29.5%, 13.4%, and 8.9% of the explained variance, respectively; Supplementary Data 2, Table 3). Variance in fungal and nematode richness and protist diversity, were equally explained by changes in pesticide concentrations and soil properties. At the metagenomic level, for 20 out of the 48 functional gene groups, pesticides explained as much as, or more of the variance in diversity than soil properties. For example, 15.6% of variability in faunal genes

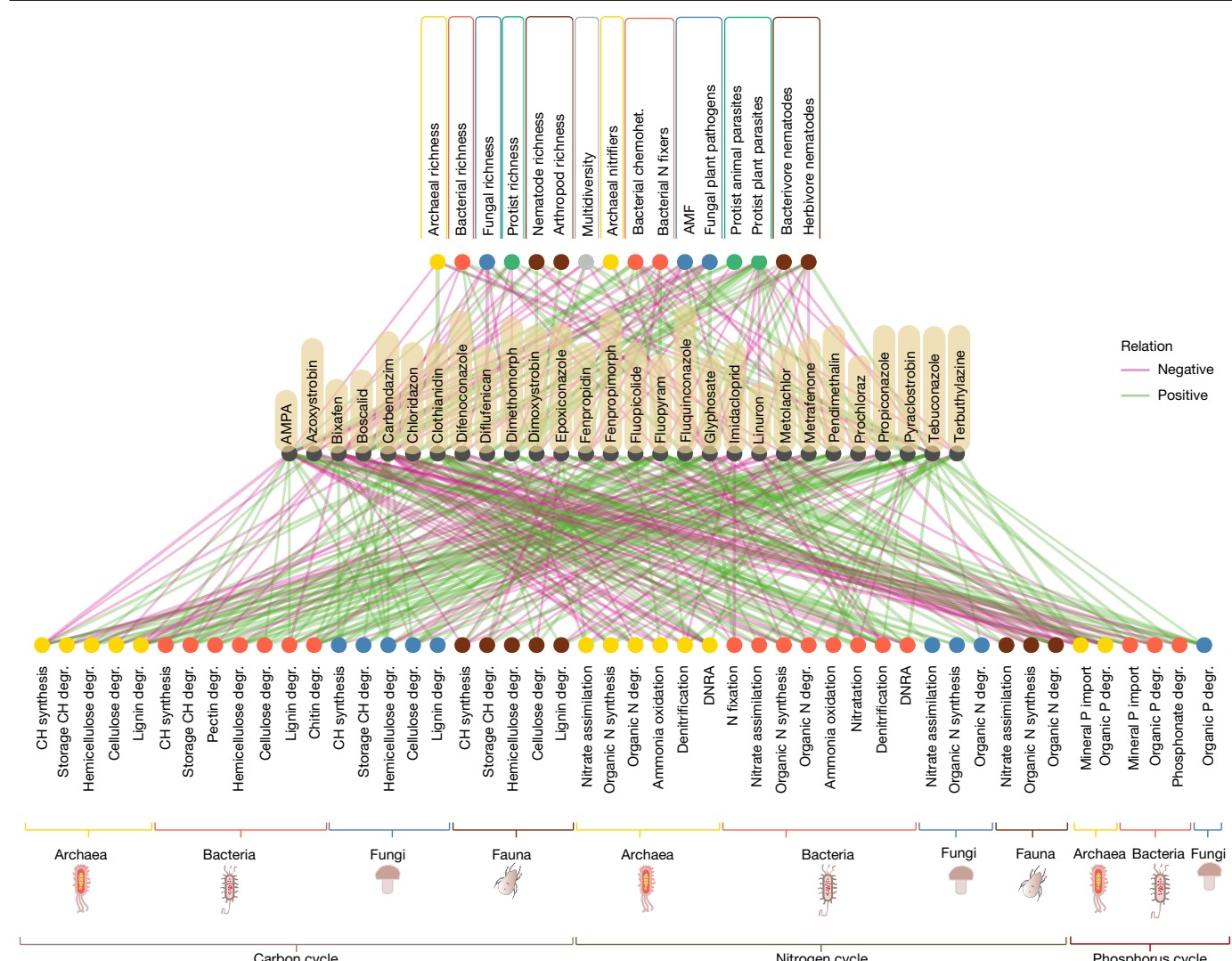

**Fig. 3 | Soil biodiversity responses to pesticide concentrations in croplands.** Variable importance (VIP) network between pesticide concentrations (centre) and the richness of archaea, bacteria, fungi, protists, nematodes and arthropods, the multidiversity index, the relative abundance of functional groups (top) and the diversity of each functional gene group involved in the carbon, nitrogen and phosphorus cycles (bottom). Nodes are coloured according to different organism groups: yellow, archaea; red, bacteria; blue, fungi; green, protists; and brown, fauna. CH, carbohydrates; degr., degradation; DNRA, dissimilatory nitrate reduction. VIP links are coloured on the basis of the coefficient sign of the given pesticide in the GLM, provided it is retained as a predictor for the biodiversity metric. Green links indicate a positive coefficient (positive relationship), and pink links represent a negative coefficient (negative relationship). Underlying VIP plots are displayed in Extended Data Figs. 5 and 6. Results for Shannon diversity can be found in Extended Data Fig. 5. These analyses focus on croplands only ($n$ = 244 sites were used for metabarcoding analyses and $n$ = 234 sites were used for metagenomics analyses); VIP plots including other ecosystem types ($n$ = 373 sites for metabarcoding analyses and $n$ = 349 sites for metagenomics analyses) are presented in Supplementary Data 3, Figs. 1 and 2. An interactive version of this figure is available at https://esdac.jrc.ec.europa. eu/content/pesticides-and-soil-biodiversity. The fitting of the models is discussed in Supplementary Results 2.

involved in organic N degradation were explained by pesticides, against 8.5% only explained by soil properties (Extended Data Fig. 6 and Supplementary Data 2, Tables 4–6).

## Pesticide effects across ecosystem types

Although most pesticide residues were detected in croplands, they were also retrieved in other ecosystems (that is, former croplands recently converted to grasslands, extensive grasslands and woodlands) (Fig. 2 and Extended Data Fig. 4). Overall, associations between pesticides and soil biodiversity were consistent when comparing analyses across all ecosystem types (that is, croplands and other ecosystems; see Supplementary Data 3) with those restricted to croplands, whereas some patterns emerged only when non-cropland ecosystems were included. Notably, associations between pendimethalin – a herbicide

that was detected in all investigated ecosystem types – and both fungal plant pathogens and herbivore nematodes were evident only when grasslands and woodlands were included in models (Supplementary Data 3, Fig. 1). Similarly, a decline in archaeal richness linked to higher bixafen concentrations was only observed in the analysis of all ecosystems (Supplementary Data 3, Fig. 1). Positive associations between the diversity of functional gene groups and pesticide concentrations were more pronounced when including grasslands (Supplementary Data 3, Fig. 2).

As expected, the contribution of pesticides to explaining variation in soil biodiversity was consistently higher in croplands alone than when considering croplands together with other ecosystems. This pattern held across taxonomic groups, functional groups (up to 29.5% of variation explained in croplands and 17.4% across all ecosystems; Fig. 4, Supplementary Data 2, Tables 1–3, Supplementary Data 3, Fig. 4

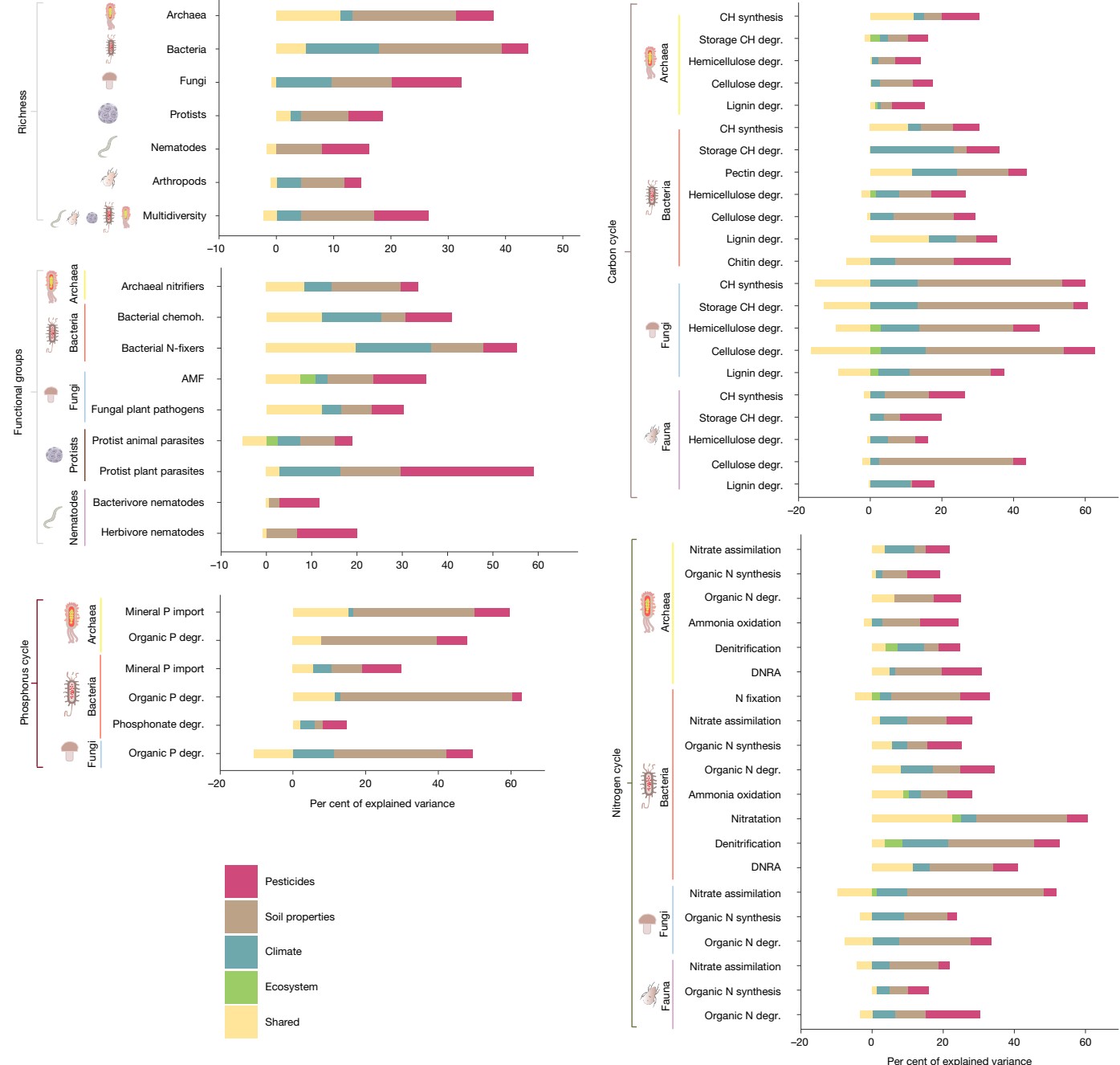

**Fig. 4 | Contribution of pesticide concentrations in explaining soil biodiversity metrics in croplands.** The relative contribution (per cent of explained variance) of pesticide residue concentrations (red), soil properties (brown), climate (blue), ecosystem type (green) and shared variance (yellow) in explaining (1) the richness of archaea, bacteria, fungi, protists, nematodes and arthropods, (2) the multidiversity index, (3) the relative abundance of functional groups, and (4) the diversity of each functional gene group involved in the C, N, P cycles, were obtained by applying variation partitioning on each associated GLM. See Extended Data Figs. 5 and 6 for the detailed pesticide concentrations and Supplementary Data 2, Tables 7 and 8 for the soil properties and climatic variables retained per GLM. Results for Shannon diversity can be found in Extended Data Fig. 8. These analyses focus on croplands only ($n$ = 244 sites were used for metabarcoding analyses and $n$ = 234 sites were used for metagenomics analyses); analyses for all ecosystem types ($n$ = 373 sites were used for metabarcoding analyses and $n$ = 349 sites were used for metagenomics analyses) are presented in Supplementary Data 3, Fig. 4. The fitting of the models is discussed in Supplementary Results 2.

and Supplementary Data 4, Tables 1–3) and functional gene groups (5.5-fold increase in explained variability when focusing on croplands only; Fig. 4, Supplementary Data 2, Tables 4–6, Supplementary Data 3, Fig. 4 and Supplementary Data 4, Tables 4–6).

These results confirm the central influence of pesticides on soil biodiversity in cropland systems and highlight the importance of including non-croplands to detect spillover effects and broader ecological patterns.

## Discussion

It has been widely acknowledged that pesticides negatively affect aboveground diversity[12–18]. This study extends these observations to the belowground environment, indicating that pesticides significantly influence soil biodiversity at both taxonomical and functional levels, with impacts on soil C, N and P cycling. Whereas previous experimental studies and analyses in specific areas have shown that pesticides

influence soil biota[5–7,21–27], to our knowledge, ours is the first study to demonstrate the relative importance of pesticides in comparison to soil properties, ecosystem type and climate at a continental scale. Our results demonstrate that especially in croplands, pesticides are—after soil properties—the second major driver of soil biodiversity patterns. Soil biodiversity responded to pesticides in different ways depending on the ecosystem, organism group, gene function and type of pesticide, with both direct and indirect effects on many non-target groups and their roles in the soil.

## Non-target effects of pesticides

Previous findings derived from controlled experiments[26,35,36] have shown that pesticides can disrupt soil food web functioning by simultaneously affecting several non-target organisms. Similarly, our in-field study at continental scale showed that, beyond their intended targets, residues of herbicides, fungicides and insecticides altered the diversity and functional structure of all studied non-target taxa investigated here. Notably, pesticides negatively affected some key beneficial taxa, such as AMF, which are essential for plant nutrient acquisition and growth[37] and bacterivore nematodes, which regulate soil bacterial populations and the rate of nutrient and organic matter cycling[38], in line with previous findings at smaller geographical scales[6]. Furthermore, higher concentrations of specific fungicides (for example, bixafen) were associated with reduced richness and diversity of archaea, bacteria, protists, nematodes and arthropods. Whereas some studies reported non-target effects of fungicides on nematode[26], protist[35] and bacterial diversity[36], these results are based on a very limited number of fungicides. Our results cover a much broader range of fungicides, and show that these compounds can affect multiple components of the soil community.

By contrast, certain herbicides (such as glyphosate and pendimethalin) were positively related to the relative abundance of undesirable groups in croplands, such as fungal plant pathogens, protist plant parasites and herbivore nematodes, suggesting potential indirect benefits from herbicide application that warrant further investigation. In addition, we found that glyphosate and the fungicide carbendazim—both of which strongly altered fungal communities, a major source of chitin[39]—were also associated with increased diversity of bacterial functional genes encoding chitin-degrading enzymes. Although this suggests that bacteria may gain advantages in pesticide-rich environments whereas fungal communities decline[40–42], such trophic cascades are difficult to confirm in uncontrolled environments. These bacterial benefits may also stem from bacterial capacity to use pesticides as nutrient sources[7]. Supporting this hypothesis, the positive association between the concentration of AMPA (the main aminophosphonate residue resulting from widespread application of glyphosate[43]) and the diversity of bacterial gene groups involved in phosphonate degradation suggests that glyphosate may serve as a significant additional source of phosphorus for bacteria.

We observed that pesticide-driven changes in taxonomic diversity overall corresponded to shifts in functional gene diversity, suggesting a close link between taxonomic and functional diversity. Furthermore, our functional gene analyses revealed that certain pesticides affected particular functional capacities of soil biota. Thus, carbendazim, AMPA, and bixafen decreased the diversity of nitrogen and phosphorus-related genes, supporting experimental findings that microbial nitrogen metabolism can be especially vulnerable to pesticides[44].

Despite these important findings, our assessment of soil biodiversity was limited to major groups of soil biota—namely archaea, bacteria, fungi, protists, nematodes and arthropods. Other groups were not considered (for example, viruses) or not included owing to methodological constraints (for example, rotifers, tardigrades and annelids; Methods and Supplementary Discussion); therefore, the biodiversity assessment presented here is not exhaustive. We also acknowledge that our survey lacked quantitative information on pesticide application rates, timing and composition of pesticide mixtures applied at each site, which prevents us from disentangling the effects of pesticides from those of intensive land management practices. To disentangle them, detailed site-level management data on fertilization, crop rotation and tillage practices, factors that are known to influence soil biota[45,46], are needed and are missing from this study. Additionally, although this study assessed 118 compounds, other harmful substances, such as organophosphate metabolites and heavy metals, were not included. Therefore, the patterns that we report should be interpreted as early warning signals of potential pesticide impacts rather than definitive evidence of direct causation.

## Revising EU risk assessment procedures

In agreement with previous research[7,23], our results highlight the challenge in drawing general conclusions about pesticide effects on soil biodiversity when relying on a small number of taxa that can be cultivated under laboratory conditions[32,47], overlooking key groups such as AMF and diverse microbial communities. Furthermore, the risk quotient based on the no observed effect concentration (NOEC), which is widely used in risk assessment, is derived solely from a reduced number of soil invertebrate species (Supplementary Results 4). As a result, the NOEC does not capture the complexity of field community responses[29]. To improve pesticide risk evaluation, future frameworks should develop protection goals and acceptable exposure thresholds based on community-level endpoints at both taxonomic and functional levels. Notably, controlled laboratory studies using field soil communities could help define pesticide concentration ranges or dose–response relationships where significant losses in taxonomic diversity or functional potential occur. These studies should be refined and validated with soils sampled across environmental gradients accounting for key confounding factors such as land use, soil properties, climate, management intensity and legacy pesticide exposure (Supplementary Results 4). In agreement with previous studies[6,44], our analysis confirmed that microbial nitrogen-related metabolism and phosphorus uptake may be particularly sensitive to pesticide impacts. This underscores the importance of using genetic markers for microbial metabolic pathways as robust indicators of changes in soil nitrogen and phosphorus biogeochemical cycles in future pesticide risk assessments.

## Future research

Our study further emphasizes the need for more field experimental studies that include different pesticide types, concentrations and mixtures, including long-term exposure scenarios. In addition, we found that pesticides applied in croplands can drift into non-agricultural areas. Because the response of soil biodiversity to pesticides was consistent across ecosystem types, our findings represent a benchmark for refining future environmental risk assessments and regulations that consider off-site pesticide effects. Future research should focus on: (1) systematically integrating pesticide use and monitoring data, including from non-cropland areas to better assess the broader exposure and potential impacts of pesticide contamination; and (2) establishing threshold values for pesticide impacts on a wider range of soil biota (for example, by including macrofauna and functional genes) across land use intensity gradients, soil types and climates[39] (see also Supplementary Discussion on robust soil biodiversity and pesticide data baselines).

## Conclusions

Our findings demonstrate that pesticide residues are critical, yet often overlooked, drivers of soil biodiversity. Their effects are complex, organism-specific and extend beyond intended targets, challenging

current regulatory reliance on simplified indicators. To protect soil ecosystems, ecotoxicological assessments must move beyond single-species tests to include community-level and functional responses. This requires transparent pesticide use data and environmental assessments across diverse ecosystems, not just croplands. Only then will we be able to assess to what extent the unintended effects of pesticides on soil organisms may compromise the ecosystem functions that underpin long-term food security. Balancing the immediate need for high crop yields with efforts to enhance environmental sustainability will require investment in sustainable pest management solutions and agro-ecological practices that support both productivity and soil health.

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

## Methods

### Sampling sites

Soil samples were collected from 373 sites in 26 European Union member states and the UK. These sites were a subset of the larger LUCAS Soil dataset[48], and overlapped between LUCAS Soil Pesticides module[49] (measurement of in-field pesticide residue concentrations) and LUCAS Soil Biodiversity module[49] (assessment of soil taxonomical and functional diversity using DNA metabarcoding and metagenomics). Samples were collected at one vegetation growing season (April to October 2018)[28,49]. The sampled sites included 210 annual croplands (for example, maize and wheat), 34 permanent croplands (for example, vineyards, orchards and olive groves), 19 recently converted grasslands (that is, former croplands not cultivated for at least one year and not subjected to crop rotation, abandoned croplands and temporary grasslands), 97 extensive grasslands and 13 woodlands (including 6 coniferous and 7 broadleaved forests). Grasslands and woodlands were included in this study to investigate potential contamination since out of 129 non-cropland sites, 39 sites were situated in cropland polygons (27 extensive grasslands, 11 former croplands recently converted to grasslands, and 1 woodland), based on satellite data from the CORINE land cover data 2018.

### Pesticide residue analyses

In the framework of LUCAS Soil Pesticides module[49], 118 residues of active ingredients of synthetic pesticides and their metabolites (breakdown products) were measured using multi-residue liquid chromatography–tandem mass spectrometry (LC–MS/MS) and multi-residue gas chromatography–tandem mass spectrometry (GC–MS/MS) methods. Glyphosate and AMPA were quantified using a designed LC–MS/MS method[1]. The limit of quantification (LOQ) varied among the different compounds and ranged between 0.001 and 0.025 mg kg$^{-1}$ (Supplementary Table 1). The LOQ was defined as the lowest level tested that still complied with criteria for recovery, repeatability, intra-laboratory reproducibility, and identification as specified by European Commission Directorate-General for Health and Food Safety[50]. Values of the LOQ in this study were comparable to those reported for other soil monitoring studies[33,51,52]. When a pesticide concentration was reported below LOQ, the value was replaced by 0.

Pesticides and their metabolites (hereafter pesticides) were chosen due to their usage frequency, persistence and measurability[1,33]. Of the 118 pesticides investigated (Supplementary Data 1), 63 were detected in at least one site (Supplementary Table 1). Sites where none of these pesticides were detected were also included in the analyses as a reference (baseline) for soil biodiversity values, that is, 28 annual croplands (representing 13% of the annual cropland sites), 13 permanent croplands (38%), 8 recently converted grasslands (42%), 52 extensive grasslands (54%), and 11 woodlands (85%). Pesticides were classified according to their (1) type (fungicide, herbicide and insecticide)[53], (2) chemical groups[54–56], and (3) target modes of action[54–56] (Supplementary Table 1).

### Soil biodiversity analyses

**Metabarcoding analyses.** In the framework of LUCAS Soil Biodiversity module, biodiversity was assessed using DNA metabarcoding methods, targeting DNA regions commonly used as molecular markers: 16S rRNA genes for archaea and bacteria, internal transcribed spacer (ITS) regions for fungi, and 18S rRNA genes for protists, nematodes and arthropods[57–59]. Specific primers were used to amplify these regions: SSU1ArF and SSU1000ArR for archaeal 16S rRNA, 515F and 926R for bacterial 16S rRNA, ITS9mun and ITS4ngsUni for fungal ITS, and Euk575F and Euk895R for eukaryotic 18S rRNA. Bacterial and eukaryotic sequencing was performed using the Illumina MiSeq platform, while archaeal and fungal sequencing used the PacBio Sequel II platform.

Three eukaryotic groups (rotifers, tardigrades and annelids) were excluded due to the small soil volume collected that prevents robust analyses of macrofauna[58] and the limitations of reference databases[60–62].

Sequences were clustered into archaeal operational taxonomic units (OTUs) using a 97% similarity threshold[59], while OTUs of fungal ITS data were analysed with a 98% similarity threshold[57]. Exact sequence variants were identified for bacterial zero-radius OTUs (zOTUs, generated with UPARSE) and 18S eukaryote amplicon sequence variants (ASVs, generated with DADA2) following methodologies established earlier[57–59]. Detailed descriptions of selected bioinformatic tools have been published for archaea[59], bacteria, fungi[57] and non-fungal eukaryotes[58]. ASV (or OTU) counts were normalized using the ranked subsampling method from the R package SRS[63].

A composite soil diversity index (hereafter multidiversity index) was used to account for the effects of pesticides on the overall assessed biodiversity (combined biodiversity). The index was calculated by averaging the standardized richness of each soil organism group (standardized between 0 and 1[64]) using a total of six organism groups (that is, archaea, bacteria, fungi, protists, nematodes and arthropods), as follows:

$$\text{Multidiversity index} = \frac{1}{6}\left(z_{\text{archaea}} + z_{\text{bacteria}} + z_{\text{fungi}} + z_{\text{protist}} + z_{\text{nematode}} + z_{\text{arthropod.}}\right)$$

where $z$ corresponds to the standardized richness of the given organism group.

Potential functional groups were assigned using functional trait databases for archaea and bacteria (FAPROTAX[65] using the cal_spe_func function from the microeco R package)[66], for fungi (FungalTraits[67]) and for nematodes (Global database of soil nematodes)[68]. Functional annotations also relied on previous studies for protists[69]. We considered nine functional groups that contribute to core ecosystem functions and were present across studied ecosystem types (Supplementary Data 4, Table 12), including plant productivity AMF[70], fungal plant pathogens[70], plant-parasitic protists[71] and herbivore nematodes[72], nitrogen loss and retention (archaeal nitrifiers and bacterial N-fixers)[73], and organic carbon release (bacterial chemoheterotrophs)[73]. We also considered bacterivore nematodes and animal-parasitic protists—key regulators of bacterial populations and indicators of soil health state[71,72]. Functional assignments followed established frameworks in soil microbial and faunal ecology. We calculated the relative abundance (%) of these potential functional groups of archaea, bacteria, fungi, protists and nematodes per site, that is, the sum of read counts represented by the functionally assigned ASVs (or OTUs) to a given functional group, divided by the total read counts.

**Metagenomics analyses.** For metagenomic analysis, DNA samples were processed following the protocol of Bahram et al.[74]. In brief, metagenomic libraries were prepared using the Nextera XT DNA Library Prep Kit (Illumina) in accordance with the manufacturer's guidelines. Samples were indexed with the Nextera Index set, and each library was prepared with 5 μl of DNA template at a concentration of 0.2 ng μl$^{-1}$. DNA concentration was measured with the Qubit 1X dsDNA High Sensitivity Kit (Invitrogen, Thermo Fisher Scientific). Sequencing was carried out on an Illumina NovaSeq platform in 2×150 paired-end mode.

Raw metagenomic sequencing reads underwent initial quality processing using fastp v.0.23.4[75] for quality filtering and error correction. Potential contaminants were removed using BBMap v.39.01 following the DOE JGI Metagenome Workflow[76]. Functional gene identification and annotation of metagenome sequencing reads were performed using eggNOG-mapper v.2.1.2[77] with eggNOG orthology database v.5.0.2[78]. High-throughput homology searches were conducted using DIAMOND v.2.1.7[79] in blastx sensitive mode. Annotation results were filtered to retain only high-confidence matches with ≥50% alignment percentage identity and E-value < 10$^{-8}$. Taxonomic origin of functional genes was determined by extracting the taxonomic classification of seed orthologues from the eggNOG-mapper results, and categorizing

genes into broader taxonomic groups (archaea, bacteria, fungi and metazoa, hereafter fauna).

Functional genes were included in analyses based on their biological relevance to biogeochemical cycles of C, N or P for each organism type (archaea, bacteria, fungi or fauna; Supplementary Data 4, Table 13). Given that carbohydrates comprise a major portion of organic matter, we included functional genes involved in the degradation of carbohydrates with varying recalcitrance—from storage carbohydrates such as starch to more complex carbohydrates like hemicellulose (rigid components and more labile pectin), cellulose, lignin and chitin. N-related functional genes represent processes such as nitrogen fixation, denitrification, nitrification, dissimilatory nitrate reduction, organic N degradation and mineral N assimilation and synthesis—key pathways contributing to greenhouse gas emissions, nitrogen leaching, competition with crops for mineral nitrogen and organic nitrogen mineralization. The selected phosphorus-related genes are involved in organic phosphorus mineralization (with phosphatases and phosphonatases) and assimilation (via P import); the balance of these processes determines phosphorus availability for plants and, consequently, plant productivity. Out of 502 selected genes, the genes involved into similar environmental function were grouped. For each functional gene group, the number of gene orthologous groups was normalized by dividing it by the number of functionally annotated reads and then multiplied by $10^6$, resulting in functional gene diversity expressed as pmOGs (number of orthologous groups per million of functional reads). In total, 48 functional gene groups (Supplementary Data 4, Table 14) were included in the analysis based on: (1) their prevalence across all ecosystem types (Supplementary Data 4, Table 15); and (2) the quality of data available for modelling purposes. Analyses included 349 sites, of which 200 were annual croplands, 34 permanent croplands, 96 croplands recently converted to grasslands and 19 extensive grasslands (woodlands were not part of the analyses).

### Environmental properties

A total of 11 soil properties were measured, including soil water content (per cent of fresh soil weight), soil texture (per cent of coarse fragments, sand and clay), soil pH ($H_2O$), electrical conductivity (dS m$^{-1}$), organic carbon to total nitrogen ratio (Corg:N ratio), available phosphorus content (mg kg$^{-1}$), bulk density (0–20 cm, g cm$^{-3}$) and extractable potassium content[48] (mg kg$^{-1}$). For the statistical analyses, pH was negatively exponentially transformed, and electrical conductivity was log-transformed. A total of seven climatic variables were used, including mean annual temperature, mean diurnal temperature range, precipitation seasonality, temperature seasonality, averaged monthly precipitation and averaged monthly temperature over the 1970–2000 period for the month of sampling, and annual aridity, all derived from the Global Climate Data[80]. Additional details about data acquisition are available from Köninger et al.[58].

### Statistical analyses

The richness and Shannon diversity of all individual taxonomic groups (archaea, bacteria, fungi, protists, nematodes and arthropods), the soil multidiversity index, the relative abundance of functional groups (archaeal nitrifiers, bacterial chemoheterotrophs, bacterial N-fixers, AMF, fungal plant pathogens, protist animal parasites, protist plant parasites, bactivore nematodes and herbivore nematodes) and the diversity of functional gene groups involved in the C, N and P cycles were included in all statistical analyses (Fig. 1).

**Pesticide concentrations in GLMs, alongside relevant soil properties, climate and ecosystem type.** We built models in which distinct pesticide concentrations (that is, not lumped together) were selected as relevant predictors of soil biodiversity responses next to soil properties, climate and ecosystem type.

For this, we first used GLMs, including variables for soil properties, climate and ecosystem type (one model per soil biodiversity metric).

To further reduce redundancy and multicollinearity, we coupled the GLMs with an Akaike information criterion (AIC)-based stepwise regression analysis, including both forward and backward selections. These models identified the most relevant soil and climatic properties impacting soil biodiversity, and if ecosystem type was a relevant predictor. As pesticides may be more prevalent in certain environments, this step prevented the biased selection of pesticide concentrations associated with those environmental conditions. Only the second (final) set of GLMs including the pesticide concentrations was used for the downstream analyses (for example, variation partitioning).

Before integrating pesticide concentrations into the models, we removed the pesticides with zero or near-zero variance along the sites (using nearZeroVar function from caret R package[81]), as they present little to no variation across sites. Specifically, pesticides with a ratio superior to 95/5 between the most common value to the second most common value were not considered, as well as pesticides for which the percentage of distinct concentration values out of the number of total samples was inferior to 5%. This led to 20 pesticide residues remaining in the dataset for all ecosystem types (AMPA, azoxystrobin, bixafen, boscalid, carbendazim, chloridazon, clothianidin, difenoconazole, diflufenican, epoxiconazole, fenpropidin, fluopicolide, fluopyram, glyphosate, imidacloprid, metolachlor, pendimethalin, prochloraz, propiconazole and tebuconazole), and 28 in croplands only (dimethomorph, dimoxystrobin, fenpropimorph, fluquinconazole, linuron, metrafenone, pyraclostrobin and terbuthylazine, in addition to the previously mentioned).

Finally, we built a second set of feature-selected GLMs (one model per soil biodiversity metric) that incorporated the remaining pesticide concentrations together with the previously selected soil properties, climate, and/or ecosystem type. In these models, we defined the scope of the selection such that the variables retained during the previous step were all included in the subsequent models, while pesticide variables were added only if they improved the model (that is, resulted in a lower AIC). These models, which included relevant environmental and pesticide predictors, were used for further analyses (that is, variation partitioning and VIP analyses, environmental differences using Kruskal–Wallis test).

Although GLMs provide an approximation to the nonlinear and complex interactions between pesticides and soil organisms, these models were proved reliable for analysing the effects of pesticides on biodiversity[82–84]. In our models, we assumed a nonlinear relationship (Gaussian log-link) between predictors and the response variable to account for potential outliers. Model diagnostics included QQ plots (using the simulateResiduals function from the DHARMa package)[85] to assess the normality of residuals (see DHARMa plots files in the outputs folders together with the R scripts from the Code availability section). Collinearity within all fitted models was assessed using the vif function from the car package[86]. Multicollinearity was detected in two models, and this is further discussed in Supplementary Results 2. Spatial autocorrelation in the model residuals was tested using the geoR package[87], revealing no clear geographical trends.

All GLM performances were assessed using the squared correlation between predicted and observed values ($r^2$), the mean absolute error, the mean squared error and the root mean squared error (see Supplementary Data 2, Tables 12–16 and Supplementary Data 4, Tables 16–20). The fitting of the models is discussed in Supplementary Results 2.

Using previously established GLMs including relevant pesticide concentrations and environmental properties, we performed VIP analyses and determined whether each pesticide concentration had a beneficial or detrimental effect on soil biodiversity (objective 1). Then, to understand the relative contribution of pesticide concentrations in shaping soil communities compared to other drivers, we used variation partitioning on the same GLMs (objective 2). Variance partitioning allows to decompose the total variance explained by a model into unique components attributable to distinct sets of predictors

(for example, specific drivers), and components of variance shared by two or more sets of predictors. The unexplained variance, represented by the residuals of the model, reflects the potential influence of unmeasured factors. To assess spillover and broader ecological effects of pesticides, we performed analysis on croplands (both annual and permanent croplands together) and compared them to all ecosystem types (objective 3, see Supplementary Data 3 for associated supplementary figures).

**The positive or negative effect of pesticide concentrations on soil biodiversity (objective 1).** We ranked the most influential pesticides affecting soil biodiversity responses using VIP calculation[88] (using the varImp function from the caret package). The positive or negative effect of each retained pesticide concentration was determined based on the sign of its coefficient in the associated GLM. These pesticide concentrations, which explain soil biodiversity patterns, were visualized with VIP plots, that incorporated their positive or negative effects (Extended Data Figs. 5 and 6 underlying Fig. 3, and Supplementary Data 3, Figs. 1 and 2). In Results, we discuss the relationships between a given pesticide concentration and each soil biodiversity metric when the relation is supported by a clear trend on the corresponding partial plot (all partial plots are available in Supplementary Data 5 and 6).

**The relative contribution of pesticide concentrations compared to soil properties, climate and ecosystem type in explaining soil biodiversity (objective 2).** We applied variation partitioning[89] to rank the contribution of the different drivers (pesticides, soil properties, climate and ecosystem type) to soil biodiversity metrics based on their portion of explained variance (for example, Fig. 4). For this, we adapted the *varpart* function from the *vegan* R package[90], comparing the $r^2$ values of sub-models (GLMs) where the target variable was explained by one driver type alone, combinations of two, or three drivers, or all drivers together (full model), and thereby isolating unique and shared contributions to explained variance (see all R scripts with names containing 'Variation partitioning' from Code availability). In our analyses, for each model, the pesticide driver refers to all pesticide concentrations selected as relevant predictors in the model. Soil properties and climate drivers grouped all edaphic and climatic variables from the same model, respectively (see Supplementary Data 2, Tables 7 and 8 and Supplementary Data 4, Tables 7 and 8). Similarly, the ecosystem driver identified whether the ecosystem type was selected as a relevant predictor in the model. Variation partitioning allowed us to determine which driver best explained the variations in soil organism richness and diversity, functional group relative abundance, functional gene group diversity, and combined soil diversity (multidiversity index).

### Complementary analyses
**Environmental conditions related to the presence of pesticides shaping soil biodiversity.** We investigated the environmental conditions in which a pesticide concentration impacting soil biodiversity was found. For each pesticide concentration retained in previous GLMs, we assessed the existence of significant differences in the mean values of the environmental variables selected in the same model between sites in which the pesticide was present versus sites in which it was not detected above LOQ, using a Kruskal–Wallis test (kruskal.test from stats package[91]). These analyses allowed us to identify which environmental conditions could be related to the persistence of individual pesticides in soils (Supplementary Results 3).

**Pesticide aggregated metrics and one-to-one correlations with soil biodiversity.** Using the 63 detected pesticide residues, we investigated the relationship between soil biodiversity metrics and pesticide occurrence using Spearman correlations, to include nonlinear correlations (Supplementary Results 4). We used several pesticide aggregated metrics including: (1) the number of pesticides detected

at each site (that is, pesticide residue with a positive concentration); (2) the number of herbicides; (3) insecticides; or (4) fungicides detected at the site, by counting the number of pesticides detected per site and separating them based on their type[53]. While these pesticide metrics do not account for potential synergistic or antagonistic effects between pesticide residues by lumping them into simplified descriptive variables, they provide a consistent basis for comparing pesticide trends across studies. These correlations were conducted for cropland sites only and when including other ecosystems.

**Pesticide risk analyses.** Following Franco et al.[28], we assessed the ecotoxicological risk of each pesticide based on the NOEC. When multiple NOEC values were reported for different endpoints on the same substance and species, the lowest NOEC value was selected ($NOEC_{min}$). For each pesticide, a risk quotient equal to the pesticide concentration divided by its $NOEC_{min}$ was calculated, and further multiplied by an assessment factor of 5, in line with regulatory risk assessment practices[92]. This risk quotient was calculated for a total of 61 pesticides for which a $NOEC_{min}$ was reported (chloridazon and terbutryn were not included in the calculations). A cumulative pesticide risk per site was calculated by summing all risk quotients of pesticides detected in a given site. Cumulative risk metrics were also calculated separately for insecticides, fungicides and herbicides. All risk metrics were included into abovementioned Spearman correlation analyses next to pesticide occurrence, and associated results are presented and discussed in Supplementary Results 4.

### Reporting summary
Further information on research design is available in the Nature Portfolio Reporting Summary linked to this article.

## Data availability
Pesticide data supporting this study are available from European Soil Data Centre (ESDAC) (https://esdac.jrc.ec.europa.eu/content/pesticides-and-soil-biodiversity), subject to registration and a data sharing agreement, owing to the confidential nature of the measurements. The Pesticide Properties Database is accessible at https://sitem.herts.ac.uk/aeru/ppdb/. The database from the Herbicide Resistance Action Committee (HRAC) is accessible at https://hracglobal.com/files/2024-HRAC-GLOBAL-HERBICIDE-MOA-CLASSIFICATION-POSTERold.pdf, the database from the Fungicide Resistance Action Committee (FRAC) is accessible at https://www.frac.info/fungicide-resistance-management/by-frac-mode-of-action-group/#open-tour, and the one from Insecticide Resistance Action Committee (IRAC) is accessible at https://irac-online.org/mode-of-action/. The raw data (DNA sequences) generated in this study have been deposited in the Sequence Read Archive (SRA) database under BioProject ID PRJNA1118194 for archaeal 16S data, BioProject ID PRJNA952168 for bacterial 16S and fungal ITS data, BioProject ID PRJNA985135 for eukaryotic 18S data and BioProject ID PRJNA1032917 for metagenomic data. The Global database of soil nematodes is available at https://github.com/hooge104/2020_global_nematode_dataset/blob/master/data/nematode_full_dataset_wBiome.csv. The sampling site environmental metadata used in this study are available from ESDAC (https://esdac.jrc.ec.europa.eu/content/soil-biodiversity-dna-eukaryotes).

## Code availability
All R scripts relating pesticide analysis to soil biodiversity are available from ESDAC (https://esdac.jrc.ec.europa.eu/content/pesticides-and-soil-biodiversity).

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

**Acknowledgements** The LUCAS Survey is coordinated by Unit E4 of the Statistical Office of the European Union (EUROSTAT). The LUCAS Soil sample collection and laboratory analysis are supported by the Directorate-General Environment (DG-ENV), Directorate-General Agriculture and Rural Development (DG-AGRI), Directorate-General Climate Action (DG-CLIMA) and Directorate-General for Health and Food Safety (DG-SANTE) of the European Commission. We also thank EFSA for supportive advice. M.G.A.v.d.H. and F.R. acknowledge funding of the Swiss National Science Foundation (grant 310030_188799). M.B. was funded by the Swedish Research Councils Formas (grant 2020–00807). This work was realized in collaboration with the European Commission's Joint Research Centre under the Collaborative Doctoral Partnership Agreements no. 35533 with the Universidade de Vigo (UVIGO) and no. 35594 with the University of Zurich (UZH).

**Author contributions** M.G.A.v.d.H., M.J.I.B. and A.O. conceptualized the study. J.K., M.L., C.B., M.G.A.v.d.H., M.J.I.B. and A.O. undertook the design and methodology. J.K., M.L., C.B., O.D. and V.M. were involved in data analysis. J.K., M.L, C.B., O.D., V.M., A.F. and M.B. were involved in data interpretation. F.R., P.P., A.J., L.T., A.O., M.G.A.v.d.H. and M.J.I.B supervised the work. J.K. and M.L. wrote the original draft. All authors contributed to reviewing and editing the final manuscript.

**Competing interests** The authors declare no competing interests.

**Additional information**
**Correspondence and requests for materials** should be addressed to A. Orgiazzi, M. J. I. Briones or M. G. A. van der Heijden.

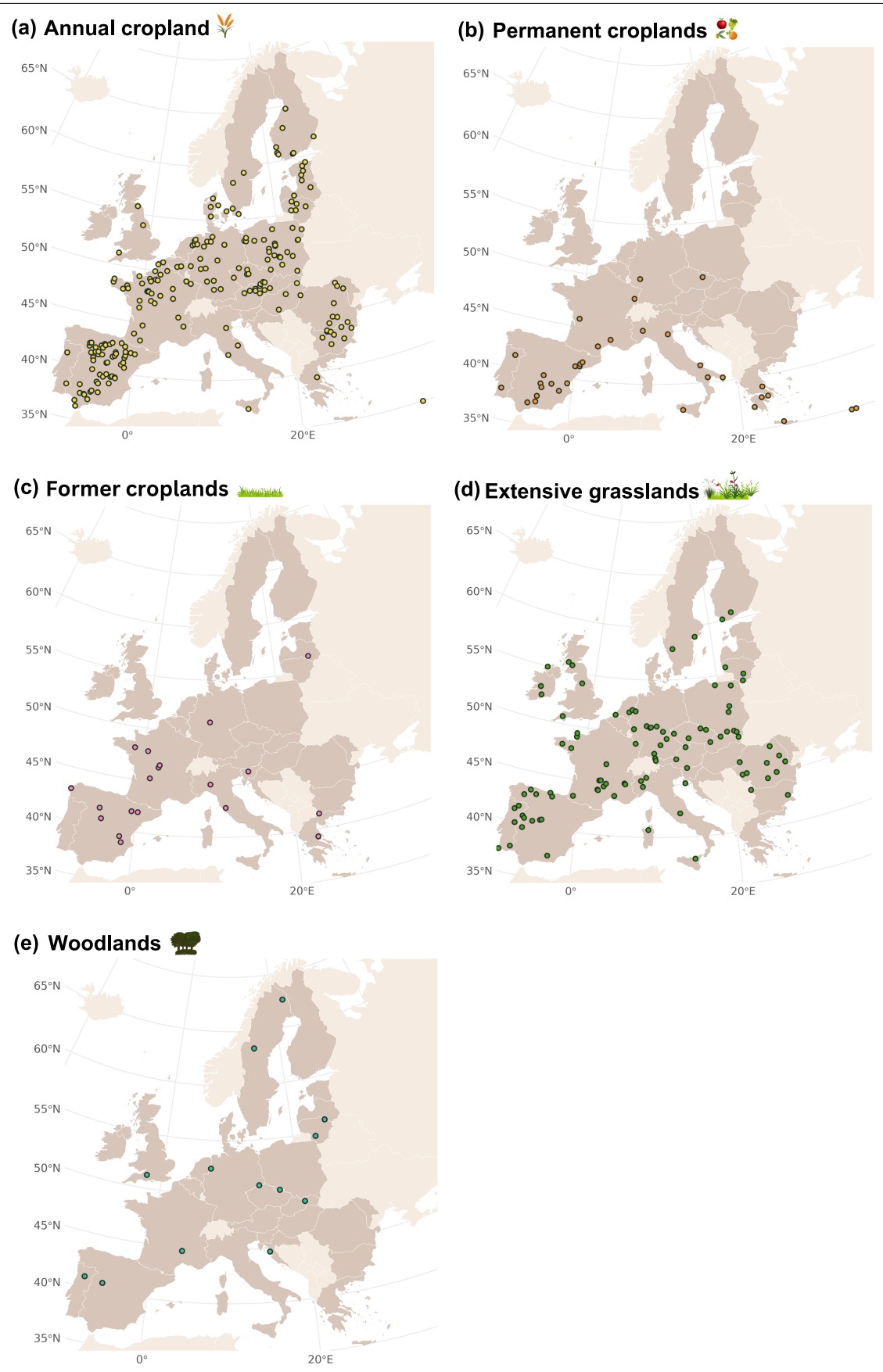

**Extended Data Fig. 1 | Sampling sites.** Maps showing locations where soil biodiversity and pesticide residues were measured across five ecosystem types for (**a**) annual croplands (AC; n = 210), (**b**) permanent croplands (PC; n = 34), (**c**) former croplands recently converted to grasslands (FC; n = 19), (**d**) extensive grasslands (EG; n = 97) and (**e**) woodlands (WL; n = 13) respectively. Twenty-six countries in darker shades include sampling sites among the 373 investigated sites. Map from Natural Earth (Creative Commons CC0).

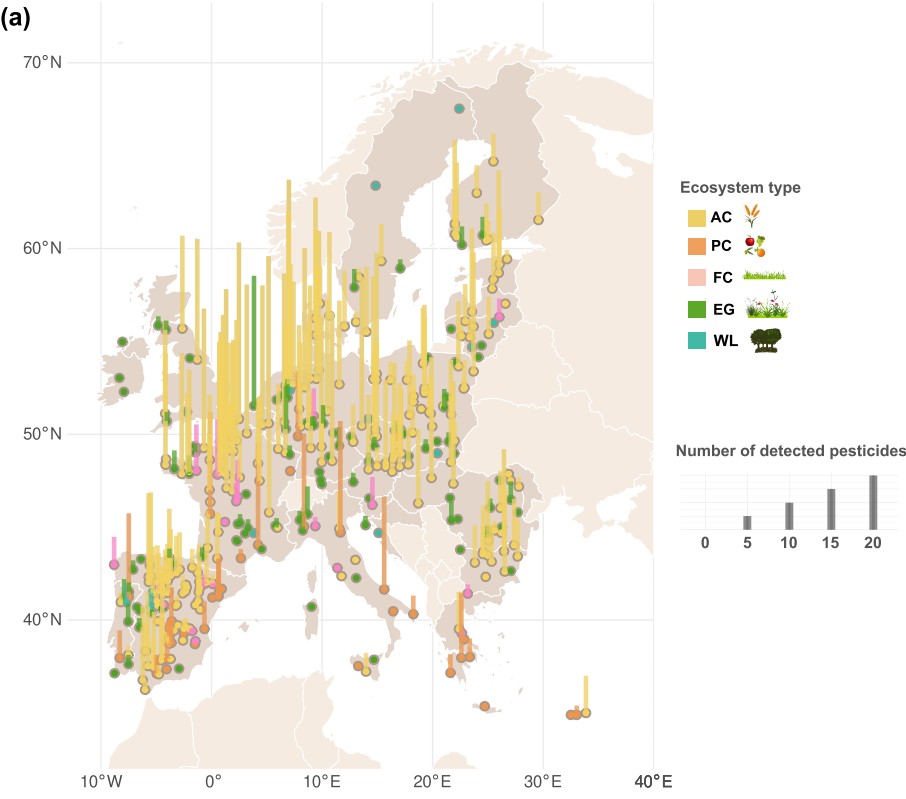

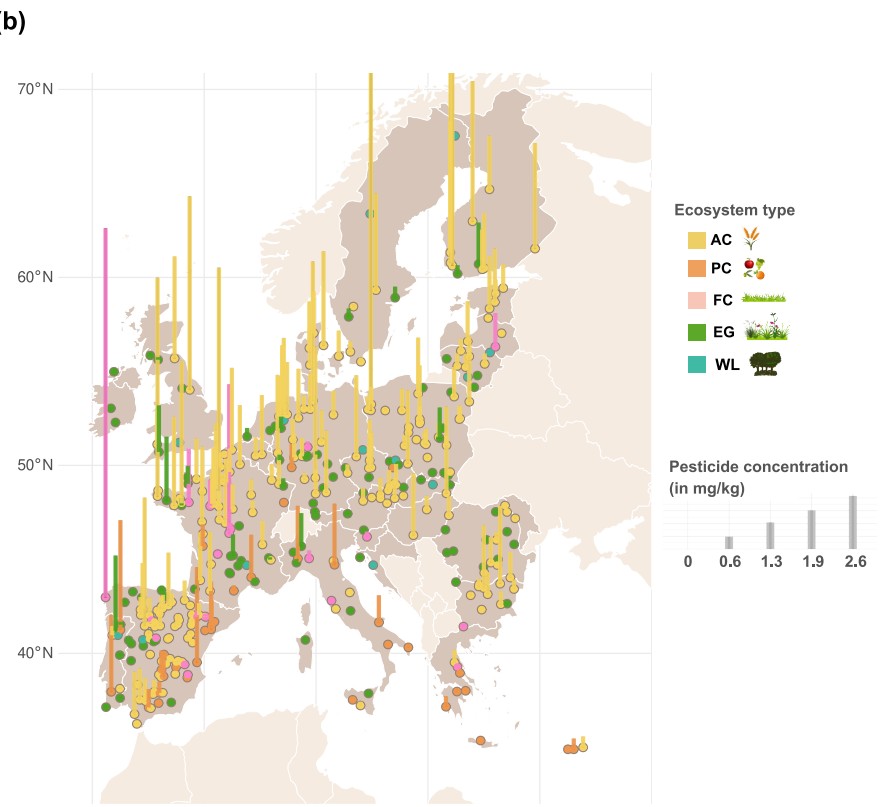

**Extended Data Fig. 2 | Maps of pesticide distribution per ecosystem type.**
Maps showing (**a**) the number of pesticides and (**b**) cumulative concentration of pesticide residues (in mg/kg) per ecosystem type across Europe (n = 373 sites): annual croplands (AC), permanent croplands (PC), former croplands recently converted to grasslands (FC), extensive grasslands (EG) and woodlands (WL).

For each location, the height of the bar is proportional to the number of pesticide residues (or cumulative concentration) detected per site, respectively. For better visualisation, the pesticide number bars were divided by two, while the concentration bars multiplied by 20. Map from Natural Earth (Creative Commons CC0).

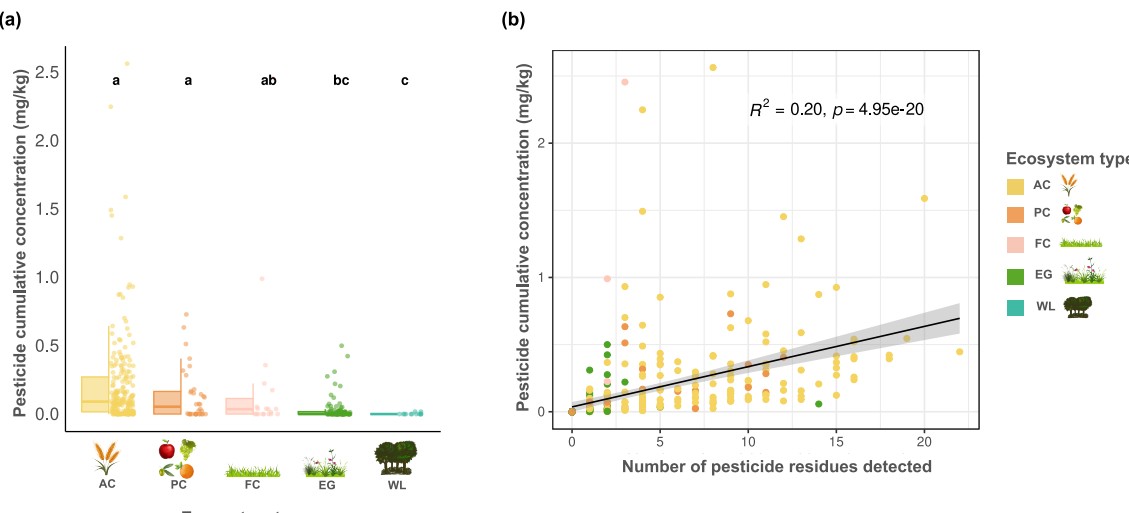

**Extended Data Fig. 3 | Pesticide cumulative concentration and regression with number of pesticides.** At n = 373 sites: (**a**) Boxplots (individual data points together with the median and its associated variation (1.5 x interquartile)) of pesticide cumulative concentration (in mg/kg) across ecosystem types. Different letters indicate significant differences (two-sided pairwise Wilcoxon multiple comparison test (with a Benjamini & Hochberg's correction)) between ecosystem types (for p-values, AC-PC = 0.069, AC-FC = 0.051, AC-EG<2e-16, AC-WL = 4.9e-06, PC-FC = 0.681, PC-EG = 0.005, PC-WL = 0.005, FC-EG = 0.069, FC-WL = 0.017, EG-WL = 0.054). (**b**) Regression models between the pesticide cumulative concentration and the number of pesticides detected per site. R-squared ($R^2$) and p-value (p) are derived from a linear model (grey zone indicates the 95% confidence interval).

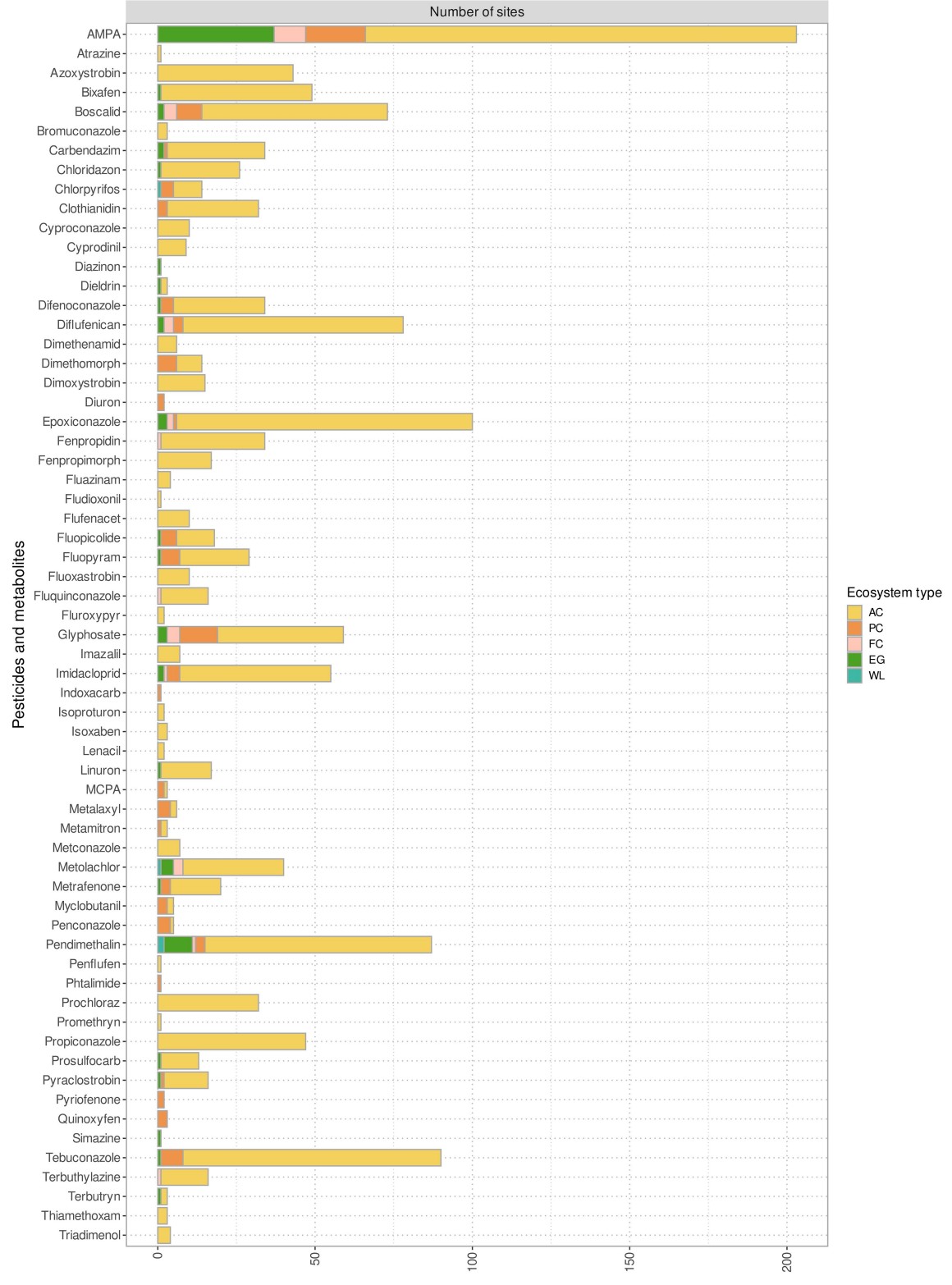

**Extended Data Fig. 4 | Pesticide distribution boxplot per ecosystem type.** Number of sites in which each pesticide was detected (summed values per pesticide) per ecosystem type (annual croplands (AC), permanent croplands (PC), former croplands recently converted to grasslands (FC), extensive grasslands (EG) and woodlands (WL); n = 373 sites).

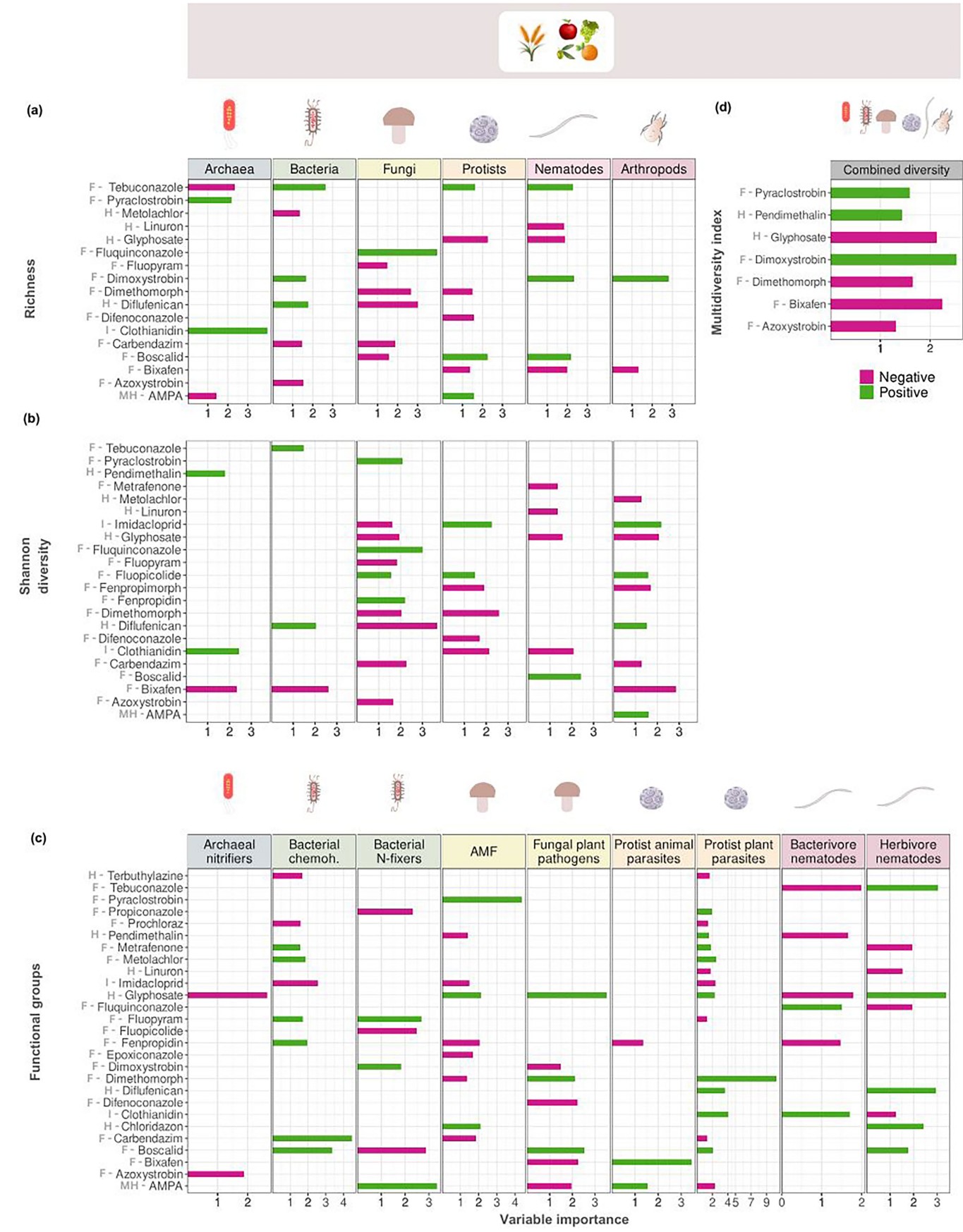

**Extended Data Fig. 5** | See next page for caption.

**Extended Data Fig. 5 | Soil biodiversity responses to pesticides in croplands.** For croplands only (n = 244 sites), soil biodiversity (assessed by metabarcoding) responses to pesticide concentrations (herbicides H, metabolite of a herbicide MH, fungicides F or insecticides I). Positive or negative relationship of each pesticide concentration retained in the models with each (**a**) soil organism observed richness, (**b**) Shannon diversity, (**c**) functional group relative abundance and (**d**) multidiversity. Horizontal bars correspond to the variable importance (VIP) coloured in green (positive relationship) or pink (negative), according to the coefficient sign of each pesticide in the associated generalised linear model (GLM). Variable importance was calculated on the feature-selected GLM including pesticide concentrations, soil properties, climate and ecosystem type information. A correlation matrix for initial environmental and predictors is available in Extended Data Fig. 7 below. These analyses focus on croplands only (n = 244 sites), while analyses for all ecosystem types (n = 373 sites) are presented in Supplementary Data 3, Fig. 1).

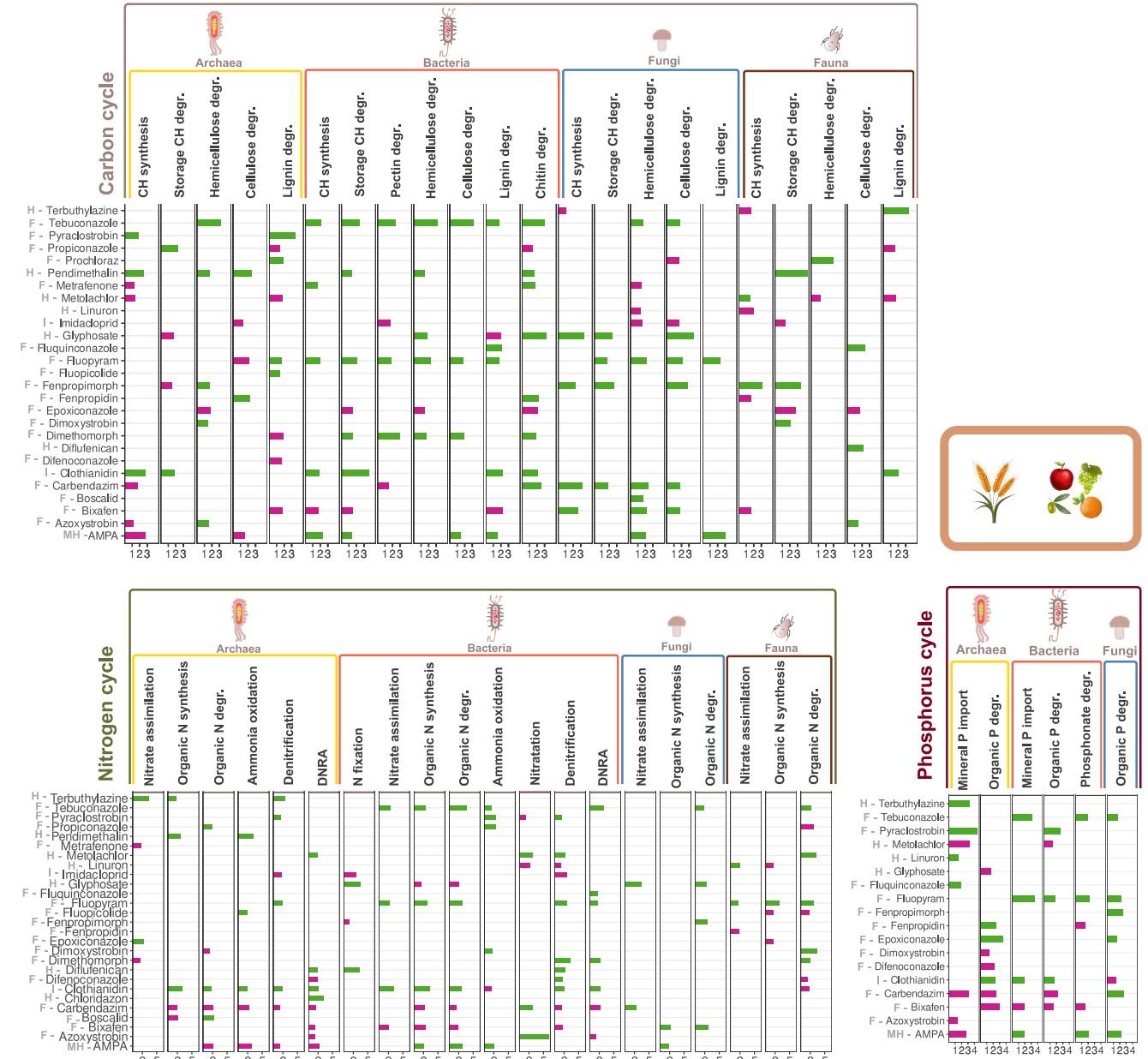

**Extended Data Fig. 6 | Soil C, N, P functional gene responses to pesticides in croplands.** For croplands only (n = 234 sites), soil C, N, P functional gene groups responses to pesticide concentrations (herbicides H, metabolite of a herbicide MH, fungicides F or insecticides I). Positive or negative relationship of concentration of pesticides retained in the GLMs with the diversity of each functional gene involved in the C, N, and P cycles. Horizontal bars correspond to the variable importance (VIP) coloured in green (positive relationship) or pink (negative), according to the coefficient sign of each pesticide in the associated GLM. Variable importance was calculated based on the feature-selected GLM, including pesticide concentrations, soil properties, climate, and ecosystem type information. These analyses focus on croplands only (n = 234 sites), while analyses including croplands and grasslands (n = 349 sites) are presented in Supplementary Data 3, Fig. 2.

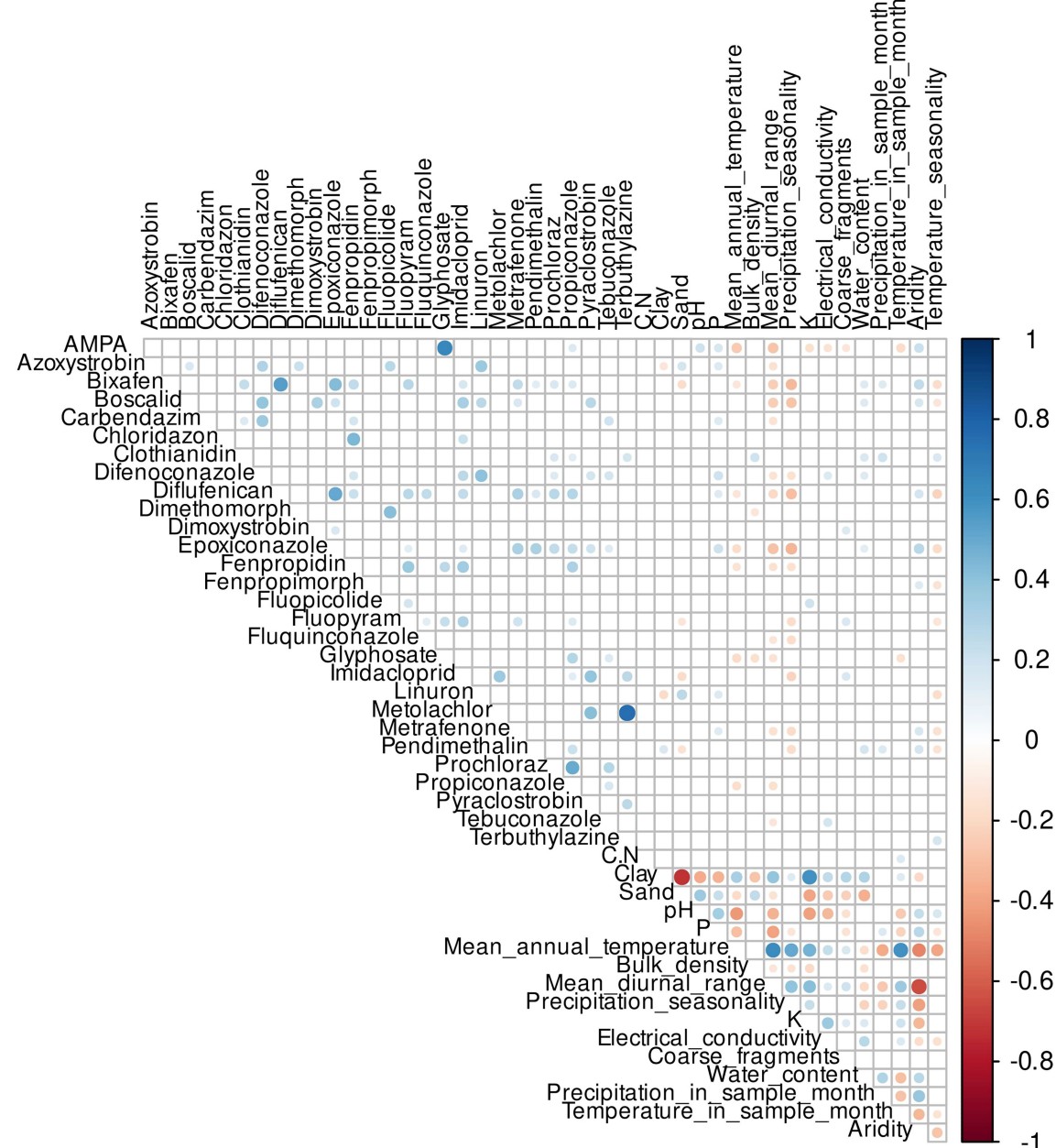

**Extended Data Fig. 7 | Correlation matrix of initial predictors for croplands.** For croplands only (n = 244 sites), Pearson (linear) correlation matrix between the initial set of predictors (i.e., before feature-selection) used in the generalised linear models, i.e., all environmental variables next to the most occurring pesticides across croplands (28 pesticides out of 63, all other pesticides with near-zero variance are not kept).

**Extended Data Fig. 8 | Explained variance of Shannon diversity in croplands.**
For croplands only (n = 244 sites), explained variance (in %) of Shannon diversity by selected variables: pesticide residue concentrations (red), soil properties (brown), climate (blue), ecosystem type (green) together with the shared variance (yellow). See Extended Data Fig. 5 for the detailed pesticide concentrations and Supplementary Data 2, Table 7 for the selected soil properties and climatic variables retained per GLM. This figure shows the results when the data of croplands only are analysed (n = 244 sites) while the results in Supplementary Data 3, Fig. 4 are based on all ecosystem types (n = 373 sites for metabarcoding analyses).

# Reporting Summary

## Statistics

For all statistical analyses, confirm that the following items are present in the figure legend, table legend, main text, or Methods section.

| n/a | Confirmed | |
|---|---|---|
| ☐ | ☒ | The exact sample size (*n*) for each experimental group/condition, given as a discrete number and unit of measurement |
| ☐ | ☒ | A statement on whether measurements were taken from distinct samples or whether the same sample was measured repeatedly |
| ☐ | ☒ | The statistical test(s) used AND whether they are one- or two-sided *Only common tests should be described solely by name; describe more complex techniques in the Methods section.* |
| ☐ | ☒ | A description of all covariates tested |
| ☐ | ☒ | A description of any assumptions or corrections, such as tests of normality and adjustment for multiple comparisons |
| ☐ | ☒ | A full description of the statistical parameters including central tendency (e.g. means) or other basic estimates (e.g. regression coefficient) AND variation (e.g. standard deviation) or associated estimates of uncertainty (e.g. confidence intervals) |
| ☐ | ☒ | For null hypothesis testing, the test statistic (e.g. *F*, *t*, *r*) with confidence intervals, effect sizes, degrees of freedom and *P* value noted *Give P values as exact values whenever suitable.* |
| ☒ | ☐ | For Bayesian analysis, information on the choice of priors and Markov chain Monte Carlo settings |
| ☒ | ☐ | For hierarchical and complex designs, identification of the appropriate level for tests and full reporting of outcomes |
| ☒ | ☐ | Estimates of effect sizes (e.g. Cohen's *d*, Pearson's *r*), indicating how they were calculated |

*Our web collection on statistics for biologists contains articles on many of the points above.*

## Software and code

Policy information about availability of computer code

| Data collection | We used Rstudio (version 2021.09.0) and R (version 4.2.1). |
|---|---|
| Data analysis | We used Rstudio (version 2021.09.0) and R (version 4.2.1). All R scripts relating pesticide analysis to soil biodiversity are available from European Soil Data Centre - ESDAC (https://esdac.jrc.ec.europa.eu/content/pesticides-and-soil-biodiversity).<br><br>The R packages needed are the following:<br>#library(BiocManager) v1.30.18<br>#library(devtools) v2.4.3<br>library(SRS) v0.2.3<br>library(microeco) v0.15.0<br>library(car) v3.0.12<br>library(caret) v6.0.94<br>library(corrplot) v0.92<br>library(cowplot) v1.1.3<br>library(dplyr) v1.0.8<br>library(geoR) v1.9.2<br>library(ggord) v1.1.5 – instructions of installation here: https://fawda123.github.io/ggord/<br>library(ggordiplots) v0.4.1 – instructions of installation here: https://rdrr.io/github/jfq3/ggordiplots/<br>library(ggplot2) v3.3.5<br>library(ggpmisc) v0.5.5<br>library(ggpol) v0.0.7 |

```
library(ggpubr) v0.6.0
library(ggrepel) v0.9.5
library(magrittr) v2.0.3
library(multcompView) v0.1.8
library(patchwork) v1.2.0
library(plyr) v1.8.7
library(readxl) v1.4.0
library(rnaturalearth) v1.0.1
library(rnaturalearthdata) v1.0.0
library(sf) v1.0.7
library(stringr) v1.4.0
library(tibble) v3.2.1
library(tidyr) v1.2.0
library(vegan) v2.5.7
library(DHARMa) v0.4.7
library(Metrics) v0.1.4
library(ggraph) v2.2.1
library(visNetwork) v2.1.2
library(stats) v4.5.0
```

For manuscripts utilizing custom algorithms or software that are central to the research but not yet described in published literature, software must be made available to editors and reviewers. We strongly encourage code deposition in a community repository (e.g. GitHub). See the Nature Portfolio guidelines for submitting code & software for further information.

## Data

Policy information about availability of data

All manuscripts must include a data availability statement. This statement should provide the following information, where applicable:

- Accession codes, unique identifiers, or web links for publicly available datasets
- A description of any restrictions on data availability
- For clinical datasets or third party data, please ensure that the statement adheres to our policy

Pesticide data supporting this study are available from European Soil Data Centre (ESDAC) (https://esdac.jrc.ec.europa.eu/content/pesticides-and-soil-biodiversity), subject to registration and a data sharing agreement, owing to the confidential nature of the measurements. The Pesticide Properties Database is accessible at https://sitem.herts.ac.uk/aeru/ppdb/. The database from the Herbicide Resistance Action Committee (HRAC) is accessible at https://hracglobal.com/files/2024-HRAC-GLOBAL-HERBICIDE-MOA-CLASSIFICATION-POSTERold.pdf, the database from the Fungicide Resistance Action Committee (FRAC) is accessible at https://www.frac.info/fungicide-resistance-management/by-frac-mode-of-action-group/#open-tour, and the one from Insecticide Resistance Action Committee (IRAC) is accessible at https://irac-online.org/mode-of-action/. The raw data (DNA sequences) generated in this study have been deposited in the Sequence Read Archive (SRA) database under BioProject ID PRJNA1118194 for archaeal 16S data, BioProject ID PRJNA952168 for bacterial 16S and fungal ITS data, BioProject ID PRJNA985135 for eukaryotic 18S data and BioProject ID PRJNA1032917 for metagenomic data. The Global database of soil nematodes is available at https://github.com/hooge104/2020_global_nematode_dataset/blob/master/data/nematode_full_dataset_wBiome.csv. The sampling site environmental metadata used in this study are available from ESDAC (https://esdac.jrc.ec.europa.eu/content/soil-biodiversity-dna-eukaryotes).

## Human research participants

Policy information about studies involving human research participants and Sex and Gender in Research.

| | |
|---|---|
| Reporting on sex and gender | N/A |
| Population characteristics | N/A |
| Recruitment | N/A |
| Ethics oversight | N/A |

Note that full information on the approval of the study protocol must also be provided in the manuscript.

# Field-specific reporting

Please select the one below that is the best fit for your research. If you are not sure, read the appropriate sections before making your selection.

☐ Life sciences  ☐ Behavioural & social sciences  ☒ Ecological, evolutionary & environmental sciences

For a reference copy of the document with all sections, see nature.com/documents/nr-reporting-summary-flat.pdf

# Ecological, evolutionary & environmental sciences study design

All studies must disclose on these points even when the disclosure is negative.

| | |
|---|---|
| Study description | We investigated the impact of pesticide concentrations on soil communities of archaea, bacteria, fungi, protists, nematodes and arthropods across Europe. Nine functional groups were also investigated: archaeal nitrifiers, bacterial chemoheterotrophs, bacterial Nitrogen-fixers (N-fixers), arbuscular mycorrhizal fungi (AMF), fungal plant pathogens, (animal and plant) parasitic protists, and bacterivore and herbivore nematodes, as well as the diversity of 48 functional gene groups involved in the carbon (C), nitrogen (N) and phosphorus (P) cycles, derived from archaeal, bacterial, fungal and faunal metagenomes. We explored the effect of pesticides on the richness and diversity of the organisms, their combined diversity (using a multidiversity index), the relative abundance of the functional groups, and the diversity of each of the functional gene groups, in five ecosystem types spanning annual croplands, permanent croplands, former croplands recently converted to grasslands, extensive grasslands and woodlands. |
| Research sample | We analysed pesticide concentrations and archaeal, bacterial, eukaryotic (fungi, protists, nematodes, arthropods) DNA sequences from 373 soil samples collected all over Europe as part of LUCAS (Land Use/Cover Area frame Survey), the largest European soil survey coordinated by the European Commission. We also analysed metagenomes of archaea, bacteria, fungi and fauna in 349 soil samples. |
| Sampling strategy | Sampling points were selected based on their overlapping between LUCAS Soil Pesticides and LUCAS Soil Biodiversity modules, both presented in Orgiazzi, A. et al. LUCAS Soil Biodiversity and LUCAS Soil Pesticides, new tools for research and policy development. European Journal of Soil Science 73, e13299 (2022). |
| Data collection | At each location, five subsamples covering a depth of 20 cm were collected and mixed together. One subsample was collected at the precise geographical location of the pre-selected point while four additional subsamples were collected at the four cardinal directions (North, East, South and West), at a distance of 2m from the first subsample location in each direction. Data collection was performed by trained surveyors. Surveyor instructions for data collection are presented in Fernández-Ugalde O., Orgiazzi A., Jones A., Lugato E., Panagos P., LUCAS 2018 – SOIL COMPONENT: Sampling Instructions for Surveyors, EUR 28501 EN, doi 10.2760/023673 https://esdac.jrc.ec.europa.eu/public_path/shared_folder/doc_pub/JRC105923_LUCAS2018_JRCTechnicalReport.pdf |
| Timing and spatial scale | Data collection took place across Europe from April to October 2018. The final 373 selected soil samples were part of 26 European countries. |
| Data exclusions | Certain eukaryotic groups (e.g. rotifers, tardigrades and annelids) were excluded due to the small soil volume collected that prevents robust analyses of macrofauna and the limitations of reference databases. |
| Reproducibility | To date, there is no repeat of the experiment, but a LUCAS biodiversity campaign has been performed in 2022 and will allow in the future to verify the patterns found for 2018. Analysis and results for the latest campaign will be available in 2025-2026. |
| Randomization | Samples were characterized by their meta-information (e.g., ecosystem type) but not allocated into further groups for the analyses. |
| Blinding | *Describe the extent of blinding used during data acquisition and analysis. If blinding was not possible, describe why OR explain why blinding was not relevant to your study.* |

Did the study involve field work?    ☒ Yes    ☐ No

## Field work, collection and transport

| | |
|---|---|
| Field conditions | Soil samples were collected from April to October 2018. A full list of metadata (e.g., date, coordinates) were collected. Climatic conditions were accounted for into the models. |
| Location | Soils samples were collected across European Union and United Kingdom. Sampling locations extended from Sweden to Cyprus (latitudinal gradient) and from Portugal to Cyprus (longitudinal gradient). |
| Access & import/export | Samples were collected following national rules. In some cases (private land) owners were duly informed before collection. No permission was sought for land in public ownership. |
| Disturbance | No disturbance were implied. |

# Reporting for specific materials, systems and methods

We require information from authors about some types of materials, experimental systems and methods used in many studies. Here, indicate whether each material, system or method listed is relevant to your study. If you are not sure if a list item applies to your research, read the appropriate section before selecting a response.

## Materials & experimental systems

| n/a | Involved in the study |
|-----|----------------------|
| ☒ | Antibodies |
| ☒ | Eukaryotic cell lines |
| ☒ | Palaeontology and archaeology |
| ☒ | Animals and other organisms |
| ☒ | Clinical data |
| ☒ | Dual use research of concern |

## Methods

| n/a | Involved in the study |
|-----|----------------------|
| ☒ | ChIP-seq |
| ☒ | Flow cytometry |
| ☒ | MRI-based neuroimaging |

