## [Peer Review File · Nature]

Pesticide residues alter taxonomic and functional biodiversity in soils

Corresponding Author: Professor Marcel van der Heijden

Version 0:

Reviewer comments:

Referee #1

(Remarks to the Author)

The manuscript "Pesticides are a main driver of soil biodiversity across Europe" by Koninger et al examined the effects of pesticides on soil microbial and some faunal communities across European woodlands, grasslands, and croplands. They found 70% studied sites contained pesticide residues and different soil biota respond differently to pesticide residues i.e. positive relationship for bacteria and negative for soil fungi. They concluded that these findings support for urgent consideration of pesticides on soil biodiversity in regulatory decision making.

Studies on effects of pesticides on soil biodiversity and functions is not new, a lot of data have been reported for plot scale and some meta-analysis to upscale. Some examples include

1. <https://besjournals.onlinelibrary.wiley.com/doi/full/10.1111/1365-2664.14437>
2. <https://www.frontiersin.org/journals/environmental-science/articles/10.3389/fenvs.2021.643847/full>
3. <https://iopscience.iop.org/article/10.1088/1748-9326/abe5d6/meta>
4. <https://link.springer.com/article/10.1007/s00244-014-0124-5>
5. <https://www.sciencedirect.com/science/article/pii/S004565352202313X>
6. <https://www.sciencedirect.com/science/article/pii/S0038071722003170>
7. <https://www.nature.com/articles/s41559-022-01799-8>

The strength of this manuscript is obtaining empirical evidence at for multiple pesticides, along with various soil communities simultaneously in standardized experimental design. Such data are needed for developing management and policy decision systems.

A. Title: In my opinion, title of the manuscript is non-informative in its current form. I suggest following title for authors to consider

Contrasting response of soil biota to pesticide contaminations, or
Pesticide contaminations reduce diversity of fungi but not for other soil taxa.

B. Introduction: from the presentation side, I think manuscript needs improvement in articulation of why this study is needed, and what are current knowledge gaps, other than scale of the study. For example, magnitude and quantification of contaminations are not known or is it how different communities respond to different pesticides are not known. What are management and policy data requirements which are currently not available.

Line 47-48- is not appropriate. The persistence depends on chemical structures (which determines half-life) of pesticides. Most of currently used pesticides will have half-life in weeks and few months. Saying that it is also likely some 'short half-life' pesticides can persist longer in soils which have high absorption/ adsorption or low microbial activities. But these need to be articulate.

C. Results: Most of results obtained were as expected. For example, fungicide negatively correlated with total fungi, AMF and

soil fungal pathogens. They are fungicides and they are expected to have this negative impact. Some of the findings are unexpected for example, why bacterivorous nematodes are negatively correlated with fungicide residues. Fungicides neither impact nematodes, nor bacteria. Authors did not provide rationale for these findings. Another thing I found strange that no nematicide was detected (see Supplementary Table 1)? I was wondering if this is because the detection system was not able to do so? Given the focus on nematode community, and widespread use of nematicides, I would argue this dataset can be important. Other strange finding was that insecticide has negative impacts on fungi but not arthropods (which they are designed to kill)? Author does not provide explanation for this.

Figure 1e and multiple figures in supplementary materials- there is no text/ word on X-axis (which is replaced by photos), I thought text or both text + photos will be better for audience

D. Discussion: As mentioned above the strength of this manuscript is bringing together empirical evidence for multiple pesticides, along with various soil communities, simultaneously. However, in my opinion this section needs to briefly include couple of strange findings outline above, but most importantly, in current form it lacks impact statements. In other words, how are these findings going to impact management and policy strategies. For example, do results from this study require change in policy, if yes which ones. There is a long paragraph (L337-353) on future research needs, which is good, but focus should be on impacts of current findings.

Any treatment (chemical or otherwise) is likely to have some effects on diversity of various communities. What are consequences for these changes- will it compromise ecosystem functions and how- should be focus of the discussion. Additionally, are these negative effects transient? If yes, why should we worry about? If not, what should be done to mitigate the risks without significantly compromising the agriculture productivity. For example, to maintain agriculture yields we need to use fungicides (which has the strongest impact on biodiversity) to control pathogens. What is trade off here-protecting soil fungi (without knowing functional consequences) or agricultural yields?

D. Methods. Data generated and statistical analyses are appropriate in my opinion, but I do suggest additional analyses for conclusive evidence. Can further statistical analyses be used to improve findings. For example, bacterial diversity was positively related with pesticide residue here. Previous studies have also shown that agricultural soils have relatively high bacterial diversity. Can effect of overall agriculture from pesticide application be distinguished? Will random forest model and/ or structural equation models be useful here?

One critical lack of measures, in my opinion, is some biomass/ abundance data. It is possible that changes in diversity in some cases are compensated by change in the biomass. For example, change in bacterivorous nematodes diversity could be compensated by abundance data? In that case, it is changes in niches that impacts diversity rather than direct toxic or stimulatory effects of pesticides. Some QCR data for key taxa (e.g. fungi, AMF, bacteria) can help to improve mechanistic understanding.

(Remarks on code availability)

It seems all appropriate to me but I am not expert on creating code. Hopefully, other reviewers can provide better and detailed assessments.

Referee #2

(Remarks to the Author)

The manuscript entitled "Pesticides are a main driver of soil biodiversity across Europe". This study comprehensively describes the effects of various pesticide metrics on soil biota across 373 sites in 26 European countries, including woodlands, grasslands, and croplands.

The manuscript addresses an interesting and urgent topic. However, I have some reservations regarding its novelty. There have been several studies on the effects of pesticides on belowground biodiversity, but the authors did not express well how this study is innovative compared to previous publications¹⁻⁵. In addition, this manuscript is more like a report than a research article, as it lacks detailed methodology and discussions.

Furthermore, the authors set out to study the effects of "pesticides" on "soil biodiversity". I see at least two critical issues with their main objective and this study, in that they are too vague, and let me be precise.

1. Pesticides

The authors introduced the concepts of "pesticide occurrence" and "pesticide concentration" to state the impact of pesticides on soil biodiversity. However, the definition and calculation methods for "pesticide occurrence" are lacking. It is useless to simply lump pesticide metrics into the term "pesticide occurrence". Pesticide occurrence does not reflect the amount used, toxicity, pesticide persistence, etc. For example, what does the "increased pesticide occurrence" mentioned in the abstract mean? with high resolution analysis tools you could surely "see" more pesticides in each sample.

The selection of pesticide metrics is not convincing. One of the metrics for pesticide occurrence, cumulative risk, is calculated on the basis of the no observed effect concentration (NOEC) for soil arthropods, suggesting that the ecotoxicological risk discussed in the manuscript may not be applicable to the soil microbiota and their interactions (e.g. network analysis).

Additionally, the authors evaluated the impact of pesticide concentrations relative to key environmental factors. The impact of pesticides on the soil biota largely depends on the type of active ingredients, not just the pesticide concentrations. Different active ingredients have varying ecotoxicities to the soil biota. Hence, the authors might also need to consider the

impact of pesticide active ingredients.

2. Soil biodiversity

Although the soil biodiversity in this study included soil archaea, bacteria, fungi, protists, nematodes, and arthropods, it is still incomplete and generic. For example, earthworms and plant roots, which are also key soil biota that affect vital soil processes, are not included.

Furthermore, the definition of functional groups, one of the metrics for soil biodiversity, is too vague. The functional group identifies bacterial or fungal functions on the basis solely of genus taxonomy rather than confirming microbial functional potential at the functional gene level.

References:

1. Liao, H. et al. Herbicide Selection Promotes Antibiotic Resistance in Soil Microbiomes. *Molecular Biology and Evolution* 38, 2337–2350 (2021).
2. Maggi, F., Tang, F. H. M. & Tubiello, F. N. Agricultural pesticide land budget and river discharge to oceans. *Nature* 620, 1013–1017 (2023).
3. Zheng, X. et al. Organochlorine contamination enriches virus-encoded metabolism and pesticide degradation associated auxiliary genes in soil microbiomes. *ISME J* 1–12 (2022).
4. Liu, Y.-R. et al. Soil contamination in nearby natural areas mirrors that in urban greenspaces worldwide. *Nat Commun* 14, 1706 (2023).
5. Edlinger, A. et al. Agricultural management and pesticide use reduce the functioning of beneficial plant symbionts. *Nat Ecol Evol* 6, 1145–1154 (2022).

Line-by-line feedback

Line 1: The title is too broad. Is pesticide occurrence or pesticide concentration the main driver of soil biodiversity?

Moreover, the data reported in this study do not prove that pesticides are the main driver. In Figure 3, the soil physicochemical properties are shown to be the primary factors.

Line 41: This manuscript uses pesticide residue concentrations. Are there any reports on pesticide residue concentrations?

Line 60: The article only uses the glyphosate metabolite, aminomethylphosphonic acid (AMPA), and the authors should explain why.

Lines 65-68: What were the reasons or criteria that the authors used to select the nine functional groups?

Line 88: Is it appropriate to use pesticide persistence as metrics? Persistence does not directly reflect biological effects, merely the residence time.

Line 131: The selection of broad-spectrum pesticides for analysis appears to be rational for assessing their widespread biological impacts. However, how is the “broad spectrum” defined? None of the pesticides mentioned are broad-spectrum for archaea or bacteria, which are two important members of “soil biodiversity”.

Line 133: The relationship between each pesticide and microbiota is not a simple linear relationship but characterized by distinct dose thresholds. Summing concentrations of different types of pesticides or even aggregating concentrations of all pesticides for correlation analysis could lead to confounding results.

sticide occurrence?

Line 138: How is the multidiversity index calculated? Please provide the calculation formula for multidiversity index.

Lines 140--141: Arbuscular mycorrhizal fungi (AMF) have been defined in the introduction.

Line 143: Please provide the R and p values of the correlation analysis.

Line 127-167: The authors claim there are complex correlations between soil biota to pesticide occurrence. However, the $|r|$ values for the Spearman correlation results in Figure 2 are all less than 0.4. This is not a notable correlation, even if the p value is significant. The data tell us the opposite of what they claim in the manuscript. Note: $|r| > 0.95$ indicates a significant correlation; ≥ 0.8 denotes a high correlation; $0.5 \leq |r| < 0.8$ indicates a moderate correlation; $0.3 \leq |r| < 0.5$ suggests a low correlation; and $|r| < 0.3$ indicates a very weak relationship, which is considered noncorrelated.

Line 152: In Figure 2, it is evident that cumulative concentration in croplands shows a positive correlation with archaeal richness, whereas insecticide concentration exhibits a negative correlation with archaeal richness. The authors should explain these results.

Line 176: Based on a linear model or a unimodal model? Was detrended correspondence analysis conducted for model selection? Was collinearity among factors assessed? The authors should provide detailed descriptions of these methods in the methodology section.

Line 252: The author needs to further describe and discuss the fitting results of the generalized linear model.

Lines 258-267: Environmental factors can influence the persistence of pesticides in soil, as mentioned in this section. A relevant question is, how did authors obtain pesticide persistence data (lines 83--167)? Did they directly use the data from soil samples?

Additionally, the use of the Kruskal–Wallis test to analyse potential environmental conditions that may enhance the persistence of individual pesticides in soil is not rigorous. The results of the Kruskal–Wallis test can indicate significant differences, but the actual connection between environmental conditions and pesticide persistence requires experimental validation or more detailed data analysis, such as soil organic matter contents etc..

Lines 277-279: The effect of pesticide concentration is likely one of the most complex factors. The authors need to clarify the nonlinear and intricate interactions between pesticides and microbiota. Is there a threshold for pesticides to impact the soil biota?

Reference: <https://doi.org/10.1038/s41467-023-41258-x>

Lines 285-291: This statement seem to be a repetition with the results? The discussion should be a further analysis of the results, exploring deeper scientific issues as well as the significance of the study. It has been widely reported that pesticides can increase or decrease diversity. The authors need to further discuss this result.

References:

1. Zhao, F., Yang, L., Yen, H. et al. Reducing risks of antibiotics to crop production requires land system intensification within thresholds. *Nat Commun* 14, 6094 (2023).
2. Ke, M. et al. Development of a machine-learning model to identify the impacts of pesticides characteristics on soil microbial communities from high-throughput sequencing data. *Environ Microbiol* 1462-2920.16175 (2022)
3. Guinet, M., Adeux, G., Cordeau, S. et al. Fostering temporal crop diversification to reduce pesticide use. *Nat Commun* 14, 7416 (2023).
4. Wang, C. N. et al. Effects of pesticide residues on bacterial community diversity and structure in typical greenhouse soils with increasing cultivation years in Northern China. *Sci Total Environ*, 710, 136321 (2020).

Line313: What is the connection between bacterivore nematodes and plant productivity?

Lines 549-553: Why use Amplicon Sequence Variants (ASVs) for 18S analysis and operational taxonomic units (OTUs) for others? There are two completely different algorithms: OTUs are based on clustering, whereas ASVs involve denoising.

Lines 559-561: Please provide the calculation formula.

(Remarks on code availability)

Referee #3

(Remarks to the Author)

This manuscript describes field monitoring of combined soil pesticide residues and soil biota in over 300 sites in 26 European countries covering different ecosystems (annual and perennial cropland, grassland and woodland) and part of the LUCAS monitoring campaign in 2018. The paper attempts to relate the composition (richness, diversity and functional group abundance) of the soil biota to different pesticide residue metrics (number of detected pesticides, cumulative (additive) concentration and risk and individual pesticide residue concentrations), as well as ecosystem type, soil properties and climate variables and determine the most important variables for structuring the soil community. I think it is a potentially important study, though I have some issues with the inclusion of an excessive number of both dependent and predictor variables that in some cases are very correlated, many many different models are formulated rather than composing models that considers the data complexity (hierarchical or multivariate), the lack of considering other (agricultural) management related variables that are not monitored but of potential high importance in shaping soil communities and possibly correlated with the pesticide residues as well as the method to infer organismal abundance underlying the functional groups.

To summarize my major criticisms:

1) There are an excessive number of variables used for both the dependent side, but in particular for the predictor side of the x-y relations. It is evident from supplementary figure S6 that there is high correlation (mostly $r > 0.8$) among number of pesticides, herbicides, fungicides, mode of action, chemical groups, persistence, broad-spectrum, meaning that these variables would mostly explaining the same variance in dependent variables. Thus only one of these should be used in analyses (I suggest number of pesticides detected for simplicity and comparability with others work) and it should be made clear that it would represent a number of other pesticide related metrics that are impossible to separate. The other metrics (cumulative concentration and risk + number of detected insecticides) are not as correlated and could be considered for inclusion. In addition to these metrics, individual concentrations are also included in some of the analyses, without considering their interrelatedness and correlation with the compounded pesticide metrics. This is a monitoring study with the strength of measuring the cumulative (mixture of) exposure to pesticide, it is not an experiment suitable for linking soil biota metrics to specific concentrations of individual substances (this is better done in e.g. empirical papers underlying the topically related meta-analysis by Beaumelle et al. 2024 (<https://doi.org/10.1111/1365-2664.14437>)). Inclusion of this part risk to be misleading rather than informative, based on both the correlation among pesticide metrics, the lack of including management related variables (point 3 below) and the uncertainties in the abundance inference underlying functional groups (point 4 below). For the dependent side, is there really value in presenting both richness and diversity metrics? At least consider putting one of these in the supplementary part and keeping the other in the main manuscript OR make a point of keeping them both in the manuscript and be clear about why including them both are informative (e.g. better highlight IMPORTANT differences and discuss why they may differ). In addition, perhaps Beaumelle et al. (2024) could be informative in guiding the use of richness, diversity and abundance/biomass, with their conclusion that richness and diversity were more impacted by pesticides compared to abundance and biomass.

2) I can't even count how many individual statistical analyses that are done for this work; partly because there are so many dependent and predictor variables and partly because models are built for parts of the data rather than building fewer models that include interactions to evaluate e.g. the importance of sampled ecosystem or organism group in relation to the pesticide metrics using general/generalized mixed models or multivariate analysis approaches. Also the GLMs and variance partitioning to identify influential variables need additional methods details, e.g. always stating the R package and function(s) used. In addition, I can't see any detailing of how well the chosen model(s) fit the data (residues) and if/how this has been evaluated even if this box is ticked in the reporting summary ("A description of any assumptions of corrections, such as test of normality..."). Let me know where it can be found if I have overlooked this.

3) It is great that you have included considerations of ecosystem, soil and climatic properties in addition to the pesticide residues. However, there is a lack of information and considerations of other, potentially related, agricultural management variables in addition to the pesticide use aspect. Since other agricultural management activities such as fertilizer regime and soil disturbance could be related to pesticide use regimes, caution should be taken when concluding on causal impact of pesticide residue on the soil biota community composition. If I understand correctly, agricultural management aspects where not monitored in LUCAS 2018, so this information can't be explored or considered? In that case, it is rather a caution in formulations and bringing this aspect up for discussion that is needed in the manuscript – something that I can't find currently.

4) Organismal abundance is inferred based on read counts from the DNA metabarcoding. I'm not an expert on this topic, but even I know that this is a highly debated topic because of methodological challenges, e.g. biomass across and within species, taxon specific primer efficiency and amplification and post lab data processing (e.g. reviewed and discussed for terrestrial arthropods in Sickel et al. 2023 (<https://doi.org/10.3897/mbmg.7.112290>)) as well as the challenge of relict DNA remains in soil samples. I understand that you are trying to make the best out of this data, but I lack any discussion on this caveat (can be brief in the manuscript and more extended in the supplement) and much more details on the underlying methods. You are referring to other papers (62-64) but that doesn't exclude the need for a method summary also here, that includes the main considerations that the reader should keep in mind. Here I'm also wondering if the richness (i.e. number) of ASVs/OTUs could be used in place to form the functional groups and the dependent variables used in the analysis since these metrics are less questioned than the methods for abundance estimation.

That said, this seems like an interesting, timely and potentially significant study that could substantially move along our understanding of how real-world pesticide exposure via soil might be affecting the soil biota community in Europe – though the above and below points should be addressed. Since environmental risk assessment (ERA) of pesticides usually consider the potential impacts of exposure to one (or perhaps a few if co-formulated in a single product) active ingredients for a set of model species (mainly annelids), the reality is that organisms encounter pesticide exposure as mixtures. Studies like can provide insights to how the ERA process could be revised or supplemented to cover mixtures and the diversity of the soil biota. In relation to this, it would be interesting to have an improved and extended discussion in relation to the conclusions made in Franco et al. (2024, <https://doi.org/10.1002/ieam.4917>) in a mixture ERA context. E.g. the ecological risk concluded there based on standard toxicity information vs the soil biota data that you have included here. It is really a pity that annelid data is not included in the current manuscript, since most standardized toxicity data comes from this group. Another paper that could be worth relating to is Alsani et al. (2024, <https://doi.org/10.1016/j.soilbio.2024.109459>), since this has authors from the group, and specifically tackle land use intensification in relation to part of the soil biota community. Pesticide use is often an inseparable part of land use intensification, so they often have to be discussed as a bundle (e.g. as discussed in Rigal et al. 2023, <https://doi.org/10.1073/pnas.2216573120>). The authors have though done a good job of setting the work into the appropriate context and cite relevant literature.

Abstract

L35-38: Consider caveating with the lack of agricultural management practice information possibly related to pesticide use and soil residues.

Main manuscript and figures

L115: number of pesticide residues = number of detected pesticides?

L116-118: Are the different ecosystem sites evenly distributed over countries and latitude/longitude? Could this be presented in some way in the methods? I can see that there are figures with the point distribution, but it is hard to see if the different ecosystem sites are well distributed since the maps are very dominated by the annual crop land sites. Consider including supplementary maps that show the site distribution by ecosystem (i.e. one map for each ecosystem type).

L134: For the cumulative risk metric, consider including a reference to where this same approach has been used before (i.e. in Franco et al. 2024).

L136: Replace responses with relations.

L144: Why the changed direction (from positive to negative relation) for bacterial N-fixers when using all sites vs only cropland sites?

L149-150: This is an important conclusion, though it can be made just by looking at the strong correlation in supplementary figure S6. Consider using your own conclusion to simplify the manuscript.

L160-162: Since you are relying on composite pesticide metrics here, there appears to be no need, and actually contra productive, to only include the top 20/28 occurring pesticides. Would the result be different if you included all detected pesticides? Based on supplementary figure S7, that would not be the case, so I suggest to just rely on data from all detected pesticides (and in addition possibly reduce the number of predictors based on the strong correlations in supplementary figure S6).

Figure 2: Herbicides, Insecticides, Fungicides – are these referring to the number of detected compounds of these groups or something else?

L180-181: Here is a place where caution is needed in relation to pesticide residues vs other, possibly related, land use intensification metrics that are unmeasured. It is rather pesticide concentration and other related variables.

L185-194: Also here there is need for nuance, since it could equally well be unmeasured variables that correlate with pesticide residues. This is highly likely given the relation between pesticide use and other aspects of (agricultural) land use intensification.

L201-203: Could the negative shared variance be related to model overfitting?

L210-: I'm considering this part inappropriate given the nature of the data with monitoring of pesticide residues rather than experimental + potential for correlation among residue concentrations. It may be more misleading than informative, which is why I suggest to exclude this part where individual pesticide concentrations are included as predictors. Such information could rather be used to point to which pesticides drive potential exposure and risk overall or in particular ecosystems (perhaps this is already done in previous publications?).

L211-212: Are the concentrations of individual (or several) pesticides correlated?

L233: Delete "very".

L234: Delete "significantly" – if not significant, there is no relation to mention. Makes me wonder about the significance of the "negatively related" in the following sentence.

L246: responses = relations

L507-508: Are these sites isolated from cropland? That would be a requirement for considering these for background contamination and possible benchmarking. I'm curious to know the distance of isolation from cropland and in general on the landscape context land use/cover composition around these sampled points.

L521-523: Add information on what percent of the sites that these numbers represent for each of the ecosystems (e.g. 13% for cropland).

L528-529: Systemic vs non systemic is a debated character since it can refer to both the target pest and the crop plant to be protected. What is the reference here?

L534: NOEC for which organisms?

L547-549: What are the methodological challenges here? Is it just few taxonomic units within these groups that is the limitation. It is a pity that annelids can't be included since much of the standard toxicity testing is based on this group and the cumulative risk metric use here should be particular relevant to explore in relation to this group.

L560: Could the same approach be used for the functional groups? In such case, the questionable use of abundance proxies could be avoided.

L562-564: How reliable is this method for inferring abundance for these organism groups? It would be good with a supplementary section on this and possibly a caveat somewhere in the main text.

L573-: This would be an appropriate place to also mention the lack of agricultural management data. Additionally, it would be informative to have a correlation matrix (similar to supplementary figure S6) that also included soil and climatic properties, in addition to the pesticide metrics to understand the interrelationship among all explanatory variables. Finally, was there any systematic difference in soil and climatic properties among ecosystems?

L599-600: Again, these data are unsuitable for exploring the many individual pesticide concentrations as predictors.

L602-604: To support the use of separate models here, ecosystem could be included as a predictive factor with inclusion of the interaction(s) with other predictors. If $p < 0.05$, it would support the use of post hoc testing of differences in relation within ecosystems.

L607-615: It is evident from supplementary figure S6 that there is high correlation (mostly $r > 0.8$) among number of pesticides, herbicides, fungicides, mode of action, chemical groups, persistence, broad-spectrum, meaning that these variables would mostly explaining the same variance in dependent variables. Thus only one of these should be used in analyses (I suggest number of pesticides detected for simplicity and comparability with others work) and it should be made clear that it would represent a number of other pesticide related metrics that are impossible to separate. The other metrics (cumulative concentration and risk + number of detected insecticides) are not as correlated and could be considered for inclusion. Another solution to focusing on the most relevant variables would be to use multivariate statistics.

L615-628: Again, I find it inappropriate to conduct predictive testing for the individual concentrations of pesticides based on the underlying monitoring data. For the compounded pesticide metrics, it defeats their purpose to exclude some pesticides.

L661-662: Would it be possible to here also model the importance of unmeasured variables? For example by using counterfactual analysis?

L676: I'm wondering if it would not be more informative to replace the cumulative pesticide concentration here and in related figures with the number of detected pesticides since this metric was so strongly correlated to many other of the explored pesticide metrics?

L692-: This details the underlying datasets of pesticide residues and soil biota DNA metabarcoding, but not the dataset that matches the two for the 373 included sites. So in addition to the underlying dataset, I suggest to also archive the specific pesticide-soil biota dataset used in this manuscript.

L700-: Consider also archiving the R code used for the data analyses done in this manuscript. The referenced code is just for part of the underlying data, if I understand correctly.

Supplementary material

Supplementary figure S1a: Number of pesticide residues = number of detected pesticides

Supplementary figure S1: Possibly also cumulative pesticide risk on a map, but that may be very similar to the map for number of detected pesticides. Again, consider mapping each ecosystem in a separate map panel since it is hard to see anything but the abundant cropland sites.

Supplementary results S1: It is actually Roundup and not glyphosate that is studied in ref 88 – please correct accordingly.

Supplementary figure S3: Please give units on the axes when appropriate.

Supplementary figure S4 and others: N values are missing in many of the figure texts – please include these consistently.

Supplementary figure S7: I'm wondering about the inclusion of panel a here, since panel b should be sufficient and they are very similar. Is there a specific reason for including both? And for the identification of the top 20-28 pesticide compounds?

Supplementary figure S8: Also here it would probably be more interesting to have number of detected pesticides rather than cumulative concentration since the former related strongly to many of the other pesticide metrics. Why was the cumulative concentration chosen over the other metrics?

Supplementary table S6-7: Where are the model specifications that these numbers come from?

(Remarks on code availability)

Version 1:

Reviewer comments:

Referee #1

(Remarks to the Author)

The revised manuscript "Pesticide residues in soils differentially affect soil biodiversity" by Koninger et al evaluated the effects of pesticides on soil microbial and faunal communities across multiple ecosystems in Europe. They report that ~70% sites contained pesticide residues and different soil communities responded differentially to pesticide residues. In revised version they also analysed functional genes involved in carbon, nitrogen and phosphorus cycle which mainly responded

positively while soil fungi community declined. Authors concluded that these findings highlight the need for comprehensive assessment of pesticide effects on soil biodiversity for improved regulation.

In my opinion, authors have addressed most technical issues raised by reviewers by including new data from metagenomics, utilising other statistical tools (e.g. Structural equation models) and providing more details where needed. Two key weaknesses remain in my opinion

a. There is a need for better articulation of scientific needs and socio-economic and regulation policy. Combined with space constraint, the authors have understandably taken a more cautious approach, and rather focused on research requirements for future regulation/ policy impact. However, in my opinion, there is still some room for this to be strengthened and I have highlighted below where and how.

b. The Discussion section seems disjointed. This is partially due to address multiple issues raised by reviewers. Below I have provided some notes to improve this.

Other comments

1. L44. It should be '59% of the Earth's biodiversity—'

2. L46. Please provide reference for impact of soil adsorption and absorption impact on pesticide residue persistence.

3. L52. In my opinion, it should read- '--soil organisms in ecosystem functions including pesticide degradation'.

4. Line 66. This will be good place to include one or two sentence on limitation of current regulations i.e. based on assay of single compounds and on specific organisms.

5. L183-185. Impact on N2 fixing taxa- have authors accounted for presence of legume plants? Presence of legume plants will increase relative abundance of symbiotic N2 fixer which can potentially bias this dataset.

6. L302-313. Could this finding be linked to microbial activity, and pesticide adsorption? For example, lower temperature means lower microbial activities that in turn can reduce rate of pesticide degradation.

7. L316-317. You have already said this earlier, suggest to delete.

8. L330-333. This belongs to results - not in discussion section. Authors should highlight rationale or implication of this finding.

9. L336. Suggest use the term – with reduced diversity of 'some' beneficial organisms,--

10. L343- suggest – soil food web and 'potentially' ecosystem functioning by—

11. L428-435. Not sure what does this section add. There are too many – 'future research is needed' in the discussion section. This is good that authors should highlight those but that should be based on following in my opinion. How does the present work advance the discipline and how does this open up new area of research (future research) and what other evidence are needed for societal impact- in this case regulation policy. So, my suggestion to authors will be to consolidate future research requirements in one paragraph (may be paragraph just before conclusions section).

12. L445-458. Suggest to reducing this section to half and bring about 2- 3 sentence on requirements for improving specific regulations.

Overall, I believe that the revision has significantly improved the manuscript and, in my opinion, if articulated appropriately, it can have significant impacts on both the scientific discipline and relevant policies.

(Remarks on code availability)

The analyses seem straight forward to me and believe any one can repeat those analyses.

Regarding code, I do not consider my self expert on this and hopefully other reviewers can provide more informed advice.

Referee #2

(Remarks to the Author)

I appreciate the authors' efforts to address the concerns pointed out in the first-round review, particularly in clarifying the novelty of study, expanding methodological details, and strengthening the discussion. However, I still suggest rejecting this manuscript because some of the explanations are not reasonable, and does not resolve the fundamental science questions. Therefore, the new version does not reach the quality standard of Nature and requires more-than-major revisions to do so.

1. "Large scale"

The authors repeatedly emphasize the "large scale" of their study, considering multiple pesticides, species, countries, and ecosystems. However, I believe this approach, as presented, is not innovative enough to justify publication in Nature. For instance, while the authors analyze the responses of multiple species to pesticides, do the conclusions differ from those of studies focusing on a single species? Have any new scientific insights been gained? Conclusions such as "Soil biodiversity (taxonomic and functional) responded differently to pesticides depending on organism type" and "the effects of soil pesticide residues may extend beyond their target organisms" are just expected findings on the basis of prior single-species studies. Moreover, the authors claim their study is unprecedented in scale and attempt to consider multiple species, yet they overlook annelids, despite offering some justification for this omission. Additionally, soil properties, climate, and other factors can influence the impact of pesticides on soil organisms. Can the European sampling sites truly represent global climate types (e.g., as classified by the Köppen Climate classification) or soil properties (e.g., as described by SoilGrids)?

2. Pesticide

The authors acknowledged that "pesticide occurrence" is misleading and replaced it with "pesticide metrics." However, this change in terminology does not resolve the fundamental issue: the manuscript still lacks a clear and rigorous definition of how pesticide presence is quantified and interpreted. Simply renaming the term does not clarify the methodology or address

my concern that lumping various pesticide-related parameters into a single term obscures their distinct ecological implications. The authors should explicitly define how each pesticide metric is measured, normalized, and integrated into their analysis. Additionally, their response does not clarify how increased detection (e.g., owing to more sensitive analytical techniques) is distinguished from actual changes in pesticide presence in the environment.

The authors argue that the cumulative risk is based on multiple taxa (annelids, collembolans, mites) rather than just arthropods. However, this still does not justify the application of NOEC-derived risk assessments to soil microbiota, which differ significantly in sensitivity and ecological function. The fact that NOEC is derived from a limited number of model organisms makes its extrapolation to the broader soil community problematic. Simply acknowledging this limitation in the Methods and Discussion does not resolve the core issue: the risk metric does not adequately capture microbial responses to pesticides. This methodological weakness undermines the reliability of the ecological risk assessments presented. The authors state that they initially tested risk quotient models (pesticide concentration/NOEC_{min}) but did not include them owing to distortions caused by NOEC limitations. However, this does not address my concern: the impact of different pesticide active ingredients varies widely, and merely considering pesticide concentrations without accounting for specific toxicities is insufficient. The manuscript lacks an analysis of how different active ingredients contribute to the observed biodiversity changes. Without this, the study provides an oversimplified view of pesticide effects. A more robust approach would involve grouping pesticides on the basis of their mode of action or toxicity profiles to different soil biota rather than treating them as a uniform risk factor.

3. Soil biodiversity

The authors acknowledge that certain groups are excluded owing to methodological constraints but do not provide a convincing justification for omitting them entirely. Other studies have successfully incorporated earthworm communities via DNA metabarcoding or morphological approaches¹¹.

The additional analysis of annelids is appreciated, but the authors ultimately decided to exclude them on the basis of low dataset quality. This exclusion, rather than improving the study, highlights a major limitation: the study still lacks a true representation of soil macrofauna, which weakens its conclusions about soil biodiversity.

The authors now recognize the limitations of assigning functions on the basis solely of taxonomic classification, yet their response does not sufficiently address my original concern. Inferring microbial functional roles on the basis solely of genus-level taxonomy is problematic because functional potential can vary significantly within a genus. Without direct evidence from functional gene profiling or metabolic analyses, these assignments remain speculative.

Although the authors mention incorporating metagenomic insights, they do not clarify whether these were used to redefine functional groups or simply to provide additional results. If functional annotations are still largely based on taxonomy rather than gene function, the issue remains unresolved.

The authors state that they have expanded their analyses to include 48 functional gene groups related to biogeochemical cycles, but there are still major concerns:

The selection of functional genes appears arbitrary, and it is unclear how these gene groups were defined or validated. Without a clear methodological explanation, the robustness of these functional assignments is questionable.

4. Analysis

The new correlation analyses and GLMs do not fundamentally resolve the problem of pesticide impact assessment, as functional group definitions remain vague. The relationship between pesticides and functional diversity needs more rigorous exploration, potentially through direct metagenomic or transcriptomic analysis rather than inferred functions.

The conclusions of studies on the relationship between pesticides and microbial diversity contradict widely accepted perspectives. Specifically, the analysis indicated that bacterial richness is positively correlated with most pesticides, which deviates from mainstream findings. While the authors propose a reasonable explanation—that pesticide exposure reduces fungal diversity, alleviates competition and promotes pesticide-degrading bacterial communities—this hypothesis lacks robust data support. The current discussion is largely speculative and lacks concrete statistical or experimental validation. The authors are encouraged to provide more detailed supporting data.

5. Line-by-line feedback:

Lines 192-221: The authors have replaced sales statistics with pesticide residue concentration data from Vieira et al. (2023), which is a more relevant dataset. However, the manuscript does not clarify whether measured concentrations or modeled estimates were used for analysis. Furthermore, pesticide concentrations alone do not reflect actual exposure levels to soil biota—other factors such as bioavailability, degradation kinetics, and interactions with soil organic matter should be considered.

Lines 76-83: The authors cite their presence across ecosystem types, but they do not justify why these specific groups were prioritized over others.

Their relevance as biological indicators is cited from Karpouzias et al. (2022), but the authors do not provide evidence that these functional groups respond strongly to pesticide exposure. The reliance on taxonomy-based functional assignments (rather than metagenomic/metatranscriptomic evidence) is problematic, as it does not accurately reflect functional capabilities.

Line 88: The authors have removed pesticide persistence from the analysis, which is appropriate, as persistence alone does not equate to biological effects. However, the manuscript should still discuss why persistence is not a reliable indicator of pesticide impact on biodiversity—for example, highly persistent pesticides may have low bioavailability, and rapidly degrading pesticides can still exert significant effects on microbial communities.

Line 139: The Pesticide Properties Database categorization only applies to target organisms (weeds, fungi, insects), not to

nontarget soil biota (archaea, bacteria, protists, etc.). The authors admit that the pesticides analyzed do not directly target archaea or bacteria, yet they still assume broad-spectrum effects on microbial communities. The cited studies (Ma et al. 2021; Puglisi et al. 2012) suggest potential indirect effects of fungicides on bacterial diversity, but this does not justify treating these pesticides as broad-spectrum agents for all soil biotas. If the authors want to claim broad-spectrum effects, they should provide quantitative toxicity data for archaea, bacteria, and fungi, rather than assuming indirect effects.

Lines 152-155: The authors acknowledge that summing pesticide concentrations leads to potential confounding but still argue that it is a useful approach for quantifying overall pesticide effects on biodiversity. However, this method introduces serious statistical biases: 1) Nonadditive toxicity: Different pesticides have distinct mechanisms of action and toxicity thresholds. Summing concentrations ignore potential synergistic or antagonistic effects and assume a linear dose–response relationship, which is biologically unrealistic. 2) Varying environmental persistence: Pesticides degrade at different rates, and their bioavailability depends on soil properties. Aggregating concentrations does not account for differential persistence and degradation kinetics, leading to misleading conclusions.

The authors claim to have conducted generalized linear models (GLMs) for individual pesticides, but they do not present a direct comparison between the aggregated and individual models. Without such a comparison, the aggregated pesticide approach remains fundamentally flawed.

Lines 140-155: the authors report that all Spearman correlation coefficients in Figure 2 are below 0.4, which is considered "low" or "very weak" based on standard statistical guidelines. Despite this, they argue that statistical significance (p values) is more important than the strength of the correlation. This is a fundamental misinterpretation of statistical significance. A statistically significant result ($p < 0.05$) merely indicates that the observed correlation is unlikely to be due to chance; it does not imply a strong or biologically meaningful relationship. The effect size (correlation coefficient $|r|$) determines the biological relevance of the relationship. Correlations below 0.3 suggest a negligible association between pesticides and biodiversity metrics. The authors attributed the weak correlations to high spatial and temporal variability in the field data. While variability is expected, the lack of strong correlations suggests that pesticides are not the main driver of soil biodiversity, which contradicts the manuscript's primary claim.

To justify the weak correlations, the authors conduct a subset analysis in Italy, reporting higher correlations (ranging from -0.65--0.72). However, this analysis is flawed for several reasons: 1) small sample size bias: the Italy dataset includes only 18 sites, making it highly susceptible to random variation and outliers. High correlations in small samples do not generalize to the entire dataset. 2) selection bias: The authors do not explain how this subset was selected. If the region was chosen post hoc because it shows stronger correlations, this introduces confirmation bias. 3) Ignoring confounding factors: The greater correlations in Italy could be due to local environmental or land-use factors rather than pesticide effects. Without controlling for these factors, the results are inconclusive.

A more appropriate statistical approach would involve multivariate models with interaction terms rather than selectively reporting high correlations from small subsets.

References:

1. Pansu, J. et al. Landscape-scale distribution patterns of earthworms inferred from soil DNA. *Soil Biology and Biochemistry* 83, 100-105 (2015).
2. Kotttek, M., J. Grieser, C. Beck, B. Rudolf, and F. Rubel, 2006: World Map of the Köppen-Geiger climate classification updated. *Meteorol. Z.*, 15, 259-263 (2006).

(Remarks on code availability)

Referee #4

(Remarks to the Author)

We have reviewed the manuscript with special emphasis to the comments of Ref 3 and the rebuttal. In general, we agree with the conclusion of Ref 3 that 'this seems like an interesting, timely and potentially significant study that could substantially moves along our understanding of how real-world pesticide exposure via soil might be affecting the soil biota community in Europe'. The paper shows how widespread agricultural pesticide residues persist in soils and its analyses suggest that soil microbial communities are strongly modified in response. From a cautionary principle these results are of great importance for policy decisions and regulations, although more definite proofs will require further work (as detailed below).

We also agree with Referee #3 that the manuscript presents an overly complex and apparently incoherent narrative, largely due to the sheer number of analyses and datasets included, a problem that only has been partly corrected in the rebuttal and the new version of the manuscript. A few further changes, and some deliberate choices, are advisable to improve the transparency and clarity of the paper.

(1) Bring overview and structure in the many analyses for the purpose of answering the main question of the paper – evaluating the impacts of pesticides on soil biodiversity

The two paragraphs introducing the approach and methods (ll. 67-91) only partly solve the problem of bringing coherence in the many analyses done. The authors have constructed a Methodology diagram which is now hidden as Fig S15 in the Supplementary Material. In our view, this diagram (in modified form) is needed up front to bring clarity and justify the choices made. The left part of the diagram portrays very well the complexity of both pesticides and soil biodiversity. The diagram can

line out the approach of tackling this complexity, dealing with colinearities (middle part), resulting in downstream analyses addressing key aspects of the main question of the paper (right part). Here analyses on the full dataset vs. croplands only should be clearly separated. Hence, a conceptual rather than methodological diagram (although methods can be summarized in the caption) will explain the hierarchy of the analyses, and how and why subsequent steps contribute to addressing the main objective of the paper. It also presents a 'roadmap' as a guidance for the subsequent results to come. A place for the diagram in the main text would be very helpful.

We recommend that the authors use this diagram to explain their deliberate choices about which analyses are essential to the central narrative and which could be relegated to supplementary materials. For example, the title emphasizes the effects of pesticide residues on soil biodiversity, yet the inclusion of functional gene analyses is not well justified; it simply pops up in the manuscript.

(2) Highly correlated pesticide metrics

The high correlation between variables, acknowledged in the rebuttal, should be addressed in the main text or in the caption of Fig. 2. Although the authors note that Figure 2 is intended to illustrate general positive correlations and is not used in GLMs, we still find it problematic. Including all highly correlated variables risks overstating the implications. We understand the choice of showing the metrics as they are often used in scientific studies and policy making. However, it should be made clear, early in the caption of Fig 2, that for columns with a similar colour, we are essentially looking at one and the same effect, referring to the relevant Suppl Fig that gives the correlations between the variables.

Otherwise, the solution for handling the correlations between the metrics and the collinearities are appropriate, with a two-step GLM where the first step selects the pesticide concentrations to be included.

(3) Interpreting results for croplands in relation to agricultural intensification

Pesticide use is typically embedded within broader land management regimes, often co-occurring with practices like excessive use of manure and chemical fertilizer, which also strongly affect soil biota. Although the authors attempted to include some management-related variables, these do not capture long-term land-use intensity. The Discussion (II.449-458) states the problem as focal points for future studies, not as consequential for the interpretation of the current results. This is an important omission. It should be more clearly acknowledged that the observed patterns represent early-warning signals of effects of pesticides rather than evidence for direct causation. Works such as Tsiafouli et al. (2015) and de Vries et al. (2013) have shown how land-use intensity affects soil biodiversity, and with the seminal dataset that the authors present, the question remains, are effects of pesticides in fact caused by agricultural intensification or, vice versa, are pesticides a causal factor behind land-use intensification effects on soil biodiversity.

Despite these limitations, we acknowledge the strength of the study in terms of its broad taxonomic scope and geographic coverage. In particular, the consistent effects observed across ecosystems, including woodlands, are compelling, as they may allow for interpretation independent of land-use intensity. Comparing these effects more systematically with those for croplands would strengthen the case that the paper makes.

Tsiafouli, M. A., Thébault, E., Sgardelis, S. P., De Ruiter, P. C., Van Der Putten, W. H., Birkhofer, K., ... & Hedlund, K. (2015). Intensive agriculture reduces soil biodiversity across Europe. *Global change biology*, 21(2), 973-985.

De Vries, F. T., Thébault, E., Liiri, M., Birkhofer, K., Tsiafouli, M. A., Bjørnlund, L., ... & Bardgett, R. D. (2013). Soil food web properties explain ecosystem services across European land use systems. *Proceedings of the National Academy of Sciences*, 110(35), 14296-14301.

(4) Microbial richness and diversity metrics

While the authors provide a valid (albeit theoretical) rationale in the rebuttal for including both richness and diversity, the different results for richness vs diversity are unconvincing and the main text does not adequately discuss the contrasting richness and diversity patterns. Presenting both metrics thus remains unwarranted as it stands.

We would suggest transferring the results on diversity to the Supplementary. The differences between richness and diversity responses in Fig. 3 are quantitative rather than qualitative. In Fig. 4, the different lines to richness and diversity can hardly be distinguished. Removing diversity would increase its readability and its value to the paper. It thus seems that the added value of including diversity is limited; excluding it from the main text would simplify and strengthen the narrative of the paper.

The rebuttal states: '... in the presence of certain pesticides, soils may become dominated by a few more resilient taxa (with increased biomass of these groups), while more sensitive or less competitive taxa decline or disappear, leading to reduced OTU richness and diversity.' This would be a valuable addition to the main text (and also suggests that responses of richness and diversity are not divergent as suggested beforehand).

(Remarks on code availability)

NA

Referee #5

(Remarks to the Author)

I co-reviewed this manuscript with one of the reviewers who provided the listed reports.

(Remarks on code availability)

Version 3:

Reviewer comments:

Referee #1

(Remarks to the Author)

The manuscript 'Pesticide residues in soils affect soil taxonomic and functional biodiversity' by Königer et al investigated the effects of multiple pesticides on soil biodiversity and key functional gene groups across Europe. They report pesticide residues from ~70% sites and identify complex and widespread non-target impacts on soil biodiversity. Pesticides also impacted functional microbial groups specifically phosphorus and nitrogen cycling and suppressed many beneficial taxa. They concluded that key component of soil biodiversity should be included pesticide regulation to protect soil biodiversity.

I believe that the manuscript advances the discipline significantly and will have impact beyond academia and research. In my opinion methodological and statistical approaches are appropriate and robust. Authors have addressed all issues raised by me.

I also believe, Reviewer 2 comments were appropriately addressed in the revised version. I also agree with authors the including annelids based on data generated will be misleading and the methodological approach can not cover annelid diversity or community composition. Constructive comments such as incorporating pesticide properties to explain observed effects were satisfactorily addressed by carrying out additional statistical analyses. They have also revised sentences for clarity and limitations of current manuscript, and in my opinion, these are appropriate responses.

Some minor comments

L 39. integrate functional and taxonomic endpoints- 'should this be integrate functional and taxonomic characteristics---

L67-68. Will this be better expression –“highlighting a need for holistic approach in a in environmental risk assessment frameworks”

Figure 1: It has too much text and most of them too small fonts to read. Either improve the quality or better move this to supplementary materials.

L131. Relevant impacts?

L182. May be appropriate title would be 'Pesticide concentrations explain a significant variation in soil biodiversity of croplands' ?

L268. What does 'for a limited set of substances'—mean?

L 337-345. Conclusion section should include a statement on trade off. For example, what impact will it have on food security if we do not use pesticides, and how to balance competing demands of food security and environmental sustainability.

(Remarks on code availability)

This is not my strength. I hope other reviewers will be able to do that. Saying that, in previous reviews, no body question the code choices.

Referee #2

(Remarks to the Author)

I appreciate the authors' efforts to address the concerns I raised. I acknowledge the significant effort that authors have invested in the “large-scale” aspect of their work, including the number of pesticides analyzed, the variety of ecosystem types, and the wide range of soil organisms. However, a large scale does not in itself constitute innovation, I remain concerned about its novelty in theory is sufficient for publication in Nature (I do not, by all means, intend to discount the quality of this work). The conclusions drawn — even in their revised form, such as “Pesticides altered microbial functions, including phosphorus and nitrogen cycling and suppressed beneficial taxa, including arbuscular mycorrhizal fungi and bacterivore nematodes.” — are almost verified by the previous literatures (see more details in the following references).

What I think could be more significant and novel is to address functional traits using real measurements, and delineate the link between pesticides diversity and biodiversity and food web dynamics.

In addition to its limited novelty, this paper also does not present significant technical breakthroughs: the methods used

(DNA metabarcoding, metagenomics, and GLMs) are all common tools in research related to microbial community (see more details in references). While the paper reported the adverse effects of use on the soil microbiome, it fails to comprehensively quantify these impacts. A dataset of such "large scale" is immensely valuable to propose more meaningful criteria for identifying "healthy soil microbiota" and then quantitatively synthesizing the impacts of pesticides on it. But regrettably, such a valuable dataset has resulted in a simple report rather than targeted, constructive, and policy-relevant recommendations.

References (just for example):

Pesticides alter microbial functions

Zhang, X. et al. The fate and ecological risk of typical diamide insecticides in soil ecosystems under repeated application. *Journal of Hazardous Materials* 494, 138440, (2025). (Pesticides application inhibited core nitrogen and carbon cycling microbes)

Wu, G. et al. Impacts of organophosphate pesticide types and concentrations on aquatic bacterial communities and carbon cycling. *Journal of Hazardous Materials* 475, 134824, (2024). (Pesticides application inhibited both Calvin-Benson-Bassham cycle and Wood-Ljungdahl pathway)

Sim, J. X. F. et al. Impact of twenty pesticides on soil carbon microbial functions and community composition. *Chemosphere* 307, 135820, (2022). (Pesticides application influenced soil carbon cycling; also analyzed with considerable number of pesticides)

Sim, J. X. F. et al. Pesticide effects on nitrogen cycle related microbial functions and community composition. *Science of The Total Environment* 807, 150734, (2022). (Pesticides application inhibited nitrogen cycle)

Liu, Y.-R. et al. Soil contamination in nearby natural areas mirrors that in urban greenspaces worldwide. *Nature Communications* 14, 1706, doi:10.1038/s41467-023-37428-6 (2023). (Multiple soil contaminants, including pesticides, influence microbial functional traits in global scale)

Pesticides suppressed beneficial taxa

Edlinger, A. et al. Agricultural management and pesticide use reduce the functioning of beneficial plant symbionts. *Nature Ecology & Evolution* 6, 1145-1154, doi:10.1038/s41559-022-01799-8 (2022). (In their own previous work, the authors have already addressed the negative impacts of pesticides on arbuscular mycorrhizal fungi. Does the present study provide any new data or insights regarding this specific topic?)

Romero, F., Jiao, S. & van der Heijden, M. G. A. Impact of microbial diversity and pesticide application on plant growth, litter decomposition and carbon substrate use. *Soil Biology and Biochemistry* 208, 109866, (2025). (In this study from the authors' research group, the negative impacts of pesticides on organic matter degradation by beneficial taxa have already been discussed. Does the current paper provide any new findings or insights regarding this topic?)

Using DNA metabarcoding, metagenomics, and GLMs in research related to microbial community

Zhang, X. et al. The fate and ecological risk of typical diamide insecticides in soil ecosystems under repeated application. *Journal of Hazardous Materials* 494, 138440, (2025). (DNA metabarcoding, metagenomics)

Wu, G. et al. Impacts of organophosphate pesticide types and concentrations on aquatic bacterial communities and carbon cycling. *Journal of Hazardous Materials* 475, 134824, (2024). (DNA metabarcoding, RT-qPCR)

Sim, J. X. F. et al. Impact of twenty pesticides on soil carbon microbial functions and community composition. *Chemosphere* 307, 135820, (2022). (DNA metabarcoding, RT-qPCR, metagenomics)

Sim, J. X. F. et al. Pesticide effects on nitrogen cycle related microbial functions and community composition. *Science of The Total Environment* 807, 150734, (2022). (DNA metabarcoding, RT-qPCR)

Liu, Y.-R. et al. Soil contamination in nearby natural areas mirrors that in urban greenspaces worldwide. *Nature Communications* 14, 1706, doi:10.1038/s41467-023-37428-6 (2023). (metagenomics)

Edlinger, A. et al. Agricultural management and pesticide use reduce the functioning of beneficial plant symbionts. *Nature Ecology & Evolution* 6, 1145-1154, doi:10.1038/s41559-022-01799-8 (2022). (DNA metabarcoding)

Romero, F., Jiao, S. & van der Heijden, M. G. A. Impact of microbial diversity and pesticide application on plant growth, litter decomposition and carbon substrate use. *Soil Biology and Biochemistry* 208, 109866, (2025). (DNA metabarcoding)

Kang, L. et al. Metagenomic insights into microbial community structure and metabolism in alpine permafrost on the Tibetan Plateau. *Nature Communications* 15, 5920, (2024). (metagenomics, GLMs)

Yu, Z., Zeng, X.-m., Cheng, X., Zhang, Q. & Zhang, K. Patterns and Environmental Drivers of Soil Microbial Succession. *Global Change Biology* 31, e70475, (2025). (DNA metabarcoding, GLMs)

Healthy soil microbiota under pesticides application (mentioned in previous studies, but can be improved by your datasets)

Swaine, M. et al. Impact of pesticides on soil health: identification of key soil microbial indicators for ecotoxicological assessment strategies through meta-analysis. *FEMS Microbiology Ecology* 101, fiae052, (2025). (ammonia-oxidizing microorganisms as indicators of the toxicity of pesticides on soil microbiota)

Xu, N. et al. Integrating Anthropogenic–Pesticide Interactions Into a Soil Health-Microbial Index for Sustainable Agriculture at Global Scale. *Global Change Biology* 30, e17596, (2024). (propose Health-Microbial Index and analyzed the impacts of pesticides on microbiota in global scale).

(Remarks on code availability)

Referee #4

(Remarks to the Author)

The authors have majorly revised their manuscript on the impact of pesticides on soil biodiversity, resulting in a more accessible paper, a better exposure of its insistent message with clear account of how it is based on the impressive

European-wide database and series of analyses.

Our comments were well implemented, although I have a few minor additional suggestions. In particular

- The new conceptual Fig 1 is very welcome, giving overview of the analyses, their hierarchy and interrelationships. However, better match the description in the text (II.84-96) with the figure. It is confusing to have 4 objectives (i – iv) in the text and 3 in the figure. Explain in the text why the complementary analyses (and refer to them as such) are needed.
- Analyses of cropland vs all ecosystems are now better in balance
- The role of land use intensification and other factors are now appropriately discussed (II 294-303), but I miss a note in the section on future research work. Establishing threshold values for pesticide impacts is an important research avenue but will only be meaningful if combined with effects of land use intensity measures.
- Several analyses moved to Suppl (Correlation analysis; diversity measures), SEM omitted, making the paper more straightforward.

(Remarks on code availability)

Referee #5

(Remarks to the Author)

I co-reviewed this manuscript with one of the reviewers who provided the listed reports.

(Remarks on code availability)

Point by point response to the comments by the reviewers.

We would like to thank the three reviewers for carefully reviewing our manuscript and providing many constructive comments and recommendations. We have addressed all these comments (see our point-by-point responses below) and performed new analyses. Please note that we added various related manuscript files including an interactive version of figure 4, enlarged versions of Figure 2 and Figure S5 as well as outputs for random forest analysis.

Referee expertise:

Referee #1: environmental ecology/soil

Referee #2: soil science

Referee #3: environmental science/pesticide impacts

Referees' comments:

Referee #1 (Remarks to the Author):

The manuscript "Pesticides are a main driver of soil biodiversity across Europe" by Koninger et al examined the effects of pesticides on soil microbial and some faunal communities across European woodlands, grasslands, and croplands. They found 70% studied sites contained pesticide residues and different soil biota respond differently to pesticide residues i.e. positive relationship for bacteria and negative for soil fungi. They concluded that these findings support for urgent consideration of pesticides on soil biodiversity in regulatory decision making.

Studies on effects of pesticides on soil biodiversity and functions is not new, a lot of data have been reported for plot scale and some meta-analysis to upscale. Some examples include

1. <https://besjournals.onlinelibrary.wiley.com/doi/full/10.1111/1365-2664.14437>
2. <https://www.frontiersin.org/journals/environmental-science/articles/10.3389/fenvs.2021.643847/full>
3. <https://iopscience.iop.org/article/10.1088/1748-9326/abe5d6/meta>
4. <https://link.springer.com/article/10.1007/s00244-014-0124-5>
5. <https://www.sciencedirect.com/science/article/pii/S004565352202313X>
6. <https://www.sciencedirect.com/science/article/pii/S0038071722003170>
7. <https://www.nature.com/articles/s41559-022-01799-8>

Thank you for highlighting these previous studies and their importance in contextualizing our work. We now better acknowledge previous research citing these studies in the introduction and discussion. We also use these and other studies to point to knowledge gaps. Additionally, we have expanded the discussion section to include potential limitations related to policy. In agreement with our study, these articles also highlighted large research gaps in pesticide research, and this is now more clearly addressed in the manuscript:

- **Beaumelle et al. (2023):** This study was used to highlight the well-documented adverse effects of pesticides on various soil organisms, such as earthworms, bacteria, and fungi: *"Moreover, previous research has highlighted that pesticides can have unintended side*

effects on non-targeted soil organisms, including earthworms (Beaumelle et al., 2023; Pelosi et al., 2014), bacteria, and fungi (Riedo et al., 2021; Walder et al., 2022)." (lines 51-53)

Additionally, Beaumelle et al. (2023) were used to discuss regulatory challenges: *"This lack of large-scale analyses is partly due to limited quantitative information on pesticide usage, applied doses and pesticide residues remaining in soils (Beaumelle et al. 2023)."* (lines 58-59)

- **Gunstone et al. 2021:** This reference was used in the introduction to indicate that previous work has largely focused on a limited range of soil organism groups (line 54).
- **Tejada et al. 2015:** Similarly, this study was cited to highlight that earlier research examined only a narrow subset of pesticide compounds (line 55).
- **Sim et al. (2022):** In our discussion, we are mentioning this reference that also found soil properties to shape the effects of pesticides (lines 432-435): *"Moreover, our findings reveal that soil type and climate correlate with pesticide presence in shaping soil biodiversity. Future studies should cover a wide range of soil types across different climatic zones to experimentally validate their interactive role in pesticide persistence in soils (Sim et al. 2022)"*
- **Mackay et al. (2023):** This study was referenced in the discussion to illustrate key interactions between management practices and pesticide effects: *"Additionally, conventional tillage can increase pesticide leaching (Summerton et al. 2023), while conservation tillage practices —often involving pesticide-treated seeds—may shift fungal communities (Mackay et al. 2023)."* (lines 454-456).
- **Edlinger et al. (2022):** This study was cited in two key contexts. Firstly, in the introduction, to demonstrate that previous work focused on specific organism groups (line 54). Secondly, in the discussion, where we compared our findings on negative effects on AMF with the results reported by Edlinger et al. (2022) (line 337).

The strength of this manuscript is obtaining empirical evidence at for multiple pesticides, along with various soil communities simultaneously in standardized experimental design. Such data are needed for developing management and policy decision systems.

B. Title: In my opinion, title of the manuscript is non-informative in its current form. I suggest following title for authors to consider

Contrasting response of soil biota to pesticide contaminations, or Pesticide contaminations reduce diversity of fungi but not for other soil taxa.

Thank you for your suggestions, we agree that it is important to address that there are contrasting responses of different groups of soil biota to pesticides. We changed the title into: "Pesticide residues in soils differentially affect soil biodiversity."

Introduction: from the presentation side, I think manuscript needs improvement in articulation of why this study is needed, and what are current knowledge gaps, other than scale of the study. For example, magnitude and quantification of contaminations are not known or is it how different communities respond to different pesticides are not known. What are management and policy data requirements which are currently not available.

Thank you for your valuable feedback regarding the articulation of the study's needs and the identification of current knowledge gaps beyond its scale. We have revised the introduction and discussion sections to better explain why this study is needed, while addressing the knowledge gaps in contamination quantification, differential community responses, and policy/data limitations.

1. Articulation of why this study is needed and what are the knowledge gaps:

In the revised manuscript, we explicitly outline the current knowledge gaps and their relevance to the study objectives. Therefore, we have revised the introduction to emphasise the lack of understanding of contamination magnitudes and their impacts on diverse soil communities:

“Therefore, the impacts of multiple pesticides on complex soil communities at larger geographical scales and across different ecosystem types have never been studied” (lines 55–57).

Furthermore, we highlight the knowledge gaps in understanding differential responses of soil communities to pesticides, as well as the importance of ecosystem-specific contexts:

“This study provides the first continent-wide evaluation of the impacts of pesticide residues and their active ingredients and metabolites (hereafter pesticides) on soil biodiversity across Europe (EU+UK). By using field data collected from 373 sites across diverse European landscapes, we examined the effects of specific pesticide concentrations and other metrics (pesticide number, cumulative concentration, cumulative risk following Franco et al., 2024, pesticide type – fungicides, herbicides or insecticides, and the diversity of modes of action or chemical groups) on archaea, bacteria, fungi, protists, nematodes and arthropods across five distinct ecosystem types including annual croplands, permanent croplands, former croplands recently converted to grasslands, extensive grasslands, and woodlands” (lines 66–74).

2. Incorporation of the political and regulatory context:

We have placed our study into the broader context of regulatory and data availability limitations, which constrain both policy and research advancements. Thus, we now address missing regulatory data that hinder comprehensive assessments of pesticide impacts:

“This lack of large-scale analyses is partly due to limited quantitative information on pesticide usage, applied doses and pesticide residues remaining in soils” (lines 58-59).

Additionally, we highlight the limitations of current regulatory frameworks that restrict pesticide research and monitoring:

*“Furthermore, current regulatory assessments primarily focus on single substances tested on a few invertebrate species, such as single species of earthworms (*Eisenia fetida*), nematodes (*Caenorhabditis elegans*) and collembolans (*Folsomia candida*), with narrow endpoints like mineralisation and nitrogen transformation” (lines 61-64).*

3. Strength of the manuscript:

We have restructured the introduction to clearly show the novel contributions of our work. Specifically, the study’ strength lies in its large-scale, multi-organism, multi-ecosystem approach, which we have now emphasised in the introduction (lines 64-74).

Line 47-48- is not appropriate. The persistence depends on chemical structures (which determines half-life) of pesticides. Most of currently used pesticides will have half-life in weeks and few months. Saying that it is also likely some 'short half-life' pesticides can persist longer in soils which have high absorption/ adsorption or low microbial activities. But these need to be articulate.

We agree that the persistence of pesticides depends on their chemical structures and on soil properties. In the revised manuscript, we have included a sentence to define pesticide persistence by their chemical properties including their hydrophobicity (Wang et al. 2020) and soil absorption/adsorption capacity (line 45).

C. Results: Most of results obtained were as expected. For example, fungicide negatively correlated with total fungi, AMF and soil fungal pathogens. They are fungicides and they are expected to have this negative impact. Some of the findings are unexpected for example, why bacterivorous nematodes are negatively correlated with fungicide residues. Fungicides neither impact nematodes, nor bacteria. Authors did not provide rationale for these findings.

Thank you for your observation regarding the expected and unexpected results. We appreciate the opportunity to clarify and contextualise these findings.

Addressing expected results: We agree that the negative correlation between fungicide residues and fungi, including soil fungal pathogens, is expected (line 417-418). However, we also highlighted that AMF, as beneficial soil fungi, are not direct targets of fungicides and are excluded from current risk assessment procedures (lines 388-390). Moreover, evidence on their impacts, especially at larger scales, remains limited, with the exception of the study by Edlinger et al. (2022). Our study confirms the study by Edlinger et al. 2022, now at a European scale, which is an important conclusion (lines 335-337).

Explaining unexpected findings: For the observed negative correlation between fungicides and bacterivorous nematodes, previous studies have suggested that fungicides can indirectly affect nematodes. For example, they may influence nematode biomass and abundance through changes in soil microbial communities (e.g. food of the nematodes), as observed in both laboratory and field studies (Gunstone et al., 2021). Additionally, fungicides have been reported to alter bacterial diversity metrics, potentially because some bacteria can metabolise pesticide residues as a food source (Walder et al., 2022; Zhang et al., 2024).

Regarding the effects of fungicides on bacteria, and nematode, we found contrasting responses for specific products:

- **Bacteria:** The number of fungicides was positively associated with bacterial richness and diversity (Figure 2). For example, tebuconazole was linked to an increase in bacterial richness, while difenoconazole and bixafen were associated with decreases in bacterial richness and diversity, respectively (Figure 4).
- **Nematodes:** The fungicide bixafen was associated with a decrease in nematode richness and diversity (Figure 4).

To address this point, we have expanded the discussion section (lines 364-370) to incorporate these interpretations and stress the need of further research to disentangle direct and indirect effects of fungicides on soil biota. We now wrote: "Another unexpected example was that more fungicides in

soils might benefit bacterial communities while negatively affecting bacterivore nematodes suggesting a relaxed competition with fungi to bacteria (Bahram et al. 2018) (Fig. 2). Conversely, high concentrations of bixafen were negatively associated with both bacterial and nematode diversity (Fig. 4). These findings highlight the need for caution when interpreting results, since increased diversity does not necessarily indicate higher absolute abundance, greater evenness, or improved food availability.”

Bahram, M. et al. Structure and function of the global topsoil microbiome. *Nature* 560, 233-237 (2018).

Gunstone, T., Cornelisse, T., Klein, K., Dubey, A., & Donley, N. (2021). Pesticides and soil invertebrates: A Hazard assessment. *Frontiers in Environmental Science*, 9. <https://doi.org/10.3389/fenvs.2021.643847>

Walder, F. et al. Soil microbiome signatures are associated with pesticide residues in arable landscapes. *Soil Biology and Biochemistry* 174, 108830 (2022).

Zhang, L., Zuo, Q., Cai, H., Li, S., Shen, Z., & Song, T. Fungicides reduce soil microbial diversity, network stability and complexity in wheat fields with different disease resistance. *Appl. Soil Ecol.* 201, 105513 (2024).

Another thing I found strange that no nematicide was detected (see Supplementary Table 1)? I was wondering if this is because the detection system was not able to do so? Given the focus on nematode community, and widespread use of nematicides, I would argue this dataset can be important.

Only few active ingredients labelled as nematicides (NE) are available in the EU database. The same substance can have several uses, e.g. as fungicide but also as nematicide. In our dataset, two nematicides, carbofuran and fluopyram, are part of the targeted pesticides, and also classified as insecticide and fungicide, respectively. Thus, nematicides were not left out. However, we agree that the nematicides are rather underrepresented in the current data set, as not much information is available on the usage of nematicides at the EU level (neither Eurostats nor EFSA have included them in their latest pesticide report). We added this limitation in the discussion (lines 440-442).

Other strange finding was that insecticide has negative impacts on fungi but not arthropods (which they are designed to kill)? Author does not provide explanation for this.

It is true that we did not find any significant correlation effect between pesticide metrics and arthropods. However, we found one insecticide concentration (i.e., imidacloprid) to be a relevant predictor for arthropod richness and diversity (e.g., Fig. 4). This is an important finding showing that looking at pesticide metrics only (lumping all pesticides together) does not provide enough information. Also, Beaumelle et al. (2023) hypothesised that "the exoskeleton of arthropods may act as a protective barrier against penetration of pesticides (Balabanidou et al., 2018; Jänsch et al., 2006)" and thus lowering the risk to be harmed by pesticides. We now discuss these points in lines 356-363. In addition, it is important to note that Insecta is a class and Arthropod is a phylum and although it is true that insects is the largest arthropod group we can also find collembolans and mites as an example in soils. This is why we now added to the manuscript: "This may be attributed to

the protective function of the arthropod exoskeleton, or to potential differences in responses between insect arthropods and non-insect arthropods, such as collembolans or mites, which were not distinguished in our analyses.” (lines 357-360)

Additionally, we looked into the target organisms of the two insecticides identified as relevant predictors by our GLMs: imidacloprid and clothianidin. Their targets are framed very broadly and do not specifically refer to certain taxa. Thus, it is not possible to compare the targets of those insecticides with detected taxa by LUCAS, preventing a narrower analysis.

Dybey et al. Ecological impacts of pesticide seed treatments on arthropod communities in a grain crop rotation. *Journal of Applied Ecology*, 2020, vol. 57, no 5, p. 936-951.

Figure 1e and multiple figures in supplementary materials- there is no text/ word on X-axis (which is replaced by photos), I thought text or both text + photos will be better for audience

Many thanks for pointing this out. We added the text to all x-axis icons to avoid confusion.

D. Discussion: As mentioned above the strength of this manuscript is bringing together empirical evidence for multiple pesticides, along with various soil communities, simultaneously. However, in my opinion this section needs to briefly include couple of strange findings outline above, but most importantly, in current form it lacks impact statements. In other words, how are these findings going to impact management and policy strategies. For example, do results from this study require change in policy, if yes which ones. There is a long paragraph (L337-353) on future research needs, which is good, but focus should be on impacts of current findings.

We included the impacts of current findings for policies along our discussion section, by mentioning:

- the need to expand the scope of tested organisms in risk assessments (lines 384-396) and integrate potential functional endpoints such as genetic markers of nitrogen related metabolism (lines 396-401)
- the lack of comprehensive risk assessments of pesticide mixtures and potential cumulative effects of multiple pesticide residues (lines 402-410)
- the need to include other ecosystem types (grasslands, woodlands) beyond croplands, as well as spanning more soil types under different climatic contexts. This would allow to better integrate potential drift effect of pesticides, and environmental influence on pesticide persistence in soils (lines 411-435).

Any treatment (chemical or otherwise) is likely to have some effects on diversity of various communities. What are consequences for these changes- will it compromise ecosystem functions and how- should be focus of the discussion. Additionally, are these negative effects transient? If yes, why should we worry about? If not, what should be done to mitigate the risks without significantly compromising the agriculture productivity. For example, to maintain agriculture yields we need to use fungicides (which has the strongest impact on biodiversity) to control pathogens. What is trade off here-protecting soil fungi (without knowing functional consequences) or agricultural yields?

Our study highlights the complex effects of pesticides on the soil food web, which vary significantly depending on the specific pesticide used. Nevertheless, our data does not allow any conclusions that could be used for decision-making, as the available data does not allow us to clearly demonstrate

underlying mechanisms nor their impacts on soil functions. Further experimental studies performed at multiple sites are needed to show how pesticides influence soil life and what this means for soil functioning. Our new data demonstrating a link between the occurrence of specific pesticides and functional genes point to potential mechanisms that need further substantiation.

Setting thresholds for acceptable pesticide concentrations, as well as the minimum tolerable effects on non-target organisms, pathogens, and beneficial groups, largely depends on knowing the applied pesticide concentration. However, this information is currently unavailable (as reporting pesticide use and dosage isn't legally mandatory). This information would be crucial for understanding the broader impact of pesticides. This critical gap that must be urgently addressed is discussed in our updated manuscript in lines 58-59 and line 473. Moreover, in future experiments, thresholds and potential trade-offs need to be considered regarding unwanted effects on soil organisms and associated ecosystem services (lines 473-474).

D. Methods. Data generated and statistical analyses are appropriate in my opinion, but I do suggest additional analyses for conclusive evidence. Can further statistical analyses be used to improve findings. For example, bacterial diversity was positively related with pesticide residue here. Previous studies have also shown that agricultural soils have relatively high bacterial diversity. Can effect of overall agriculture from pesticide application be distinguished? Will random forest model and/ or structural equation models be useful here?

We have now included Structural equation modeling (SEM) as requested: we thank you for suggesting Structural Equation Models (SEM) to disentangle effects of agriculture from that of pesticides. Using selected pesticide metrics (number of herbicides, number of insecticides, number of fungicides, and cumulative risk), we applied SEMs including various soil properties (pH, potassium (K) content, phosphorus (P) content, soil bulk density, and water content) along the climate humidity gradient. For most organism groups, we found pesticides to have a stronger impact than changes in soil properties introduced by agriculture. In the revised manuscript, these results follow the correlation analysis, while an associated figure has been added as Supplementary Fig. S9.

We did not further pursue SEMs with distinct pesticide concentrations and environment due to the relatively small number of observations compared to the number of predictors and numerous submodels that need to be considered into the SEM.

Instead of a **random forest model**, we choose to perform GLMs with two steps:

- a first set of GLMs selected the most important environmental variables (soil properties, climate, ecosystem type) for each soil biodiversity metric;
- a second set of GLMs used the previously-selected variables as a minimum model to which (one to several) pesticide concentrations were only added if they improved the model (lower AIC).

These two steps were taken to prevent biases in variable selection that could favor pesticide concentrations highly correlated with environmental variables, potentially leading to an overestimation of the importance attributed to pesticides.

We did this because with random forest (RF), it is not possible to impose a minimum model similar to the second step of GLMs. Consequently, when applying RF, we cannot ensure that pesticides are not interacting with key soil properties or climatic conditions affecting soil biota, which may result in

pesticides being favored as predictors. Furthermore, we believe it is crucial to have interpretable results: using RF instead of GLMs would prevent us from assessing whether pesticide concentrations have a positive or negative relationship with the soil biodiversity metrics of interest.

However, to tackle your point, we performed an additional analysis using RF, applying feature selection with the RFE package after removing the predictors with negative %IncMSE values (identified with the rfPermute package). In these models, pesticide concentrations were mostly similarly selected as important features. However, for some groups, a different set or number of pesticides was chosen compared to GLMs, particularly for those organism groups with lower explained variances (i.e., lower r^2 values). We could include these RF results in the supplementary material or as a supplementary data file if it helps to put our findings into perspective. For now, these results can be found in the supplementary data file named "Supplementary_data_random_forest".

One critical lack of measures, in my opinion, is some biomass/ abundance data. It is possible that changes in diversity in some cases are compensated by change in the biomass. For example, change in bacterivorous nematodes diversity could be compensated by abundance data? In that case, it is changes in niches that impacts diversity rather than direct toxic or stimulatory effects of pesticides. Some QCR data for key taxa (e.g. fungi, AMF, bacteria) can help to improve mechanistic understanding.

We obtained bacterial and fungal biomass data for a subset of the LUCAS sites, using the same soil samples that were employed for soil biodiversity assessment. The biomass data were derived from fatty acid methyl ester (FAME) analyses conducted by Siles et al. (2024) and are available through the European Soil Data Center (ESDAC). In their study, 513 sites were analysed, of which 307 overlap with our 373 study sites. Among the 307 sites, only 3 sites corresponded to woodlands, and 58 to grasslands, which could lead to a strong unbalanced design for statistical analyses performed on all ecosystem types. Consequently, we present the results of analyses performed exclusively for croplands (n=233 sites with 201 annual croplands and 32 permanent croplands) below, that we believe are more appropriate for comparison.

CL_annual	CL_permanent	GL_extensive	GL_recently_converted	WL
201	32	58	13	3

We aimed to compare soil bacterial and fungal biomass (separately) and the fungi-to-bacteria biomass ratio to the results already obtained for bacterial and fungal richness and diversity (for 244 cropland sites, see Fig. 2 and 3 and Supplementary data file S3). For this, we performed correlation analyses (similar to Figure 2 of the revised manuscript), as well as variable importance analyses derived from GLMs (i.e., including pesticide concentrations and environment) for biomass data, defining one model for bacterial biomass, one for the fungal one, and one for their ratio.

We found that there was no direct link between richness (or diversity) and biomass for both bacteria and fungi (Pearson correlations between -0.03 and 0.15 , and Spearman correlations between 0.034 and 0.19). Quadratic models revealed only very weak relationships between variables (see figure below).

Figure: Quadratic models linking microbial biomass to richness and diversity

For the correlation analyses (similar to Figure 2 of the revised manuscript), only the fungal-to-bacterial biomass ratio was found positively correlated with increases in pesticide metrics such as the number of pesticides (Spearman's $\rho = 0.26$) or the cumulative pesticide concentration ($\rho = 0.26$).

However, using GLMs, we were able to reveal that the same (individual) pesticide concentration can be positively associated with biomass but negatively with richness and diversity for the same organism type (see figure below). This was particularly the case for tebuconazole, positively associated with an increase in bacterial richness and diversity but a decrease in bacterial biomass; carbendazim, associated with an increase in fungal biomass but a decrease in fungal richness and diversity; and glyphosate, associated with an increase in fungal biomass but a decrease in fungal diversity. These results were supported by corresponding partial plots, while the rest of the trends were weaker.

Figure: Variable importance of the pesticide concentrations selected as relevant predictors in the GLMs designed for bacterial biomass, fungal biomass and their ratio (left), compared to the variable importance obtained for bacterial and fungal richness (top right) and diversity (bottom right) in the submitted manuscript (adapted from new Supp. Figure S2 from Supp. Data File S3)

This suggests that in the presence of such pesticides, soils may become dominated by fewer more resilient taxa (increased biomass of these groups), while more sensitive and/or less competitive taxa decline or disappear, leading to a decrease in OTU richness/diversity. While these results suggest potentially interesting patterns, we believe that the numerous metabarcoding analyses presented in the initial version of the manuscript, along with the new metagenomics analyses (addressing points from reviewers 2 and 3), provide sufficient complexity. This is particularly true given that the biomass dataset represents only a subset of our sites (croplands) and a subset of our organism groups of interest (bacteria and fungi). Therefore, we prefer to omit these biomass insights in the interest of maintaining clarity and focus.

Referee #1 (Remarks on code availability):

It seems all appropriate to me but I am not expert on creating code. Hopefully, other reviewers can provide better and detailed assessments.

Referee #2 (Remarks to the Author):

The manuscript entitled "Pesticides are a main driver of soil biodiversity across Europe". This study comprehensively describes the effects of various pesticide metrics on soil biota across 373 sites in 26 European countries, including woodlands, grasslands, and croplands.

The manuscript addresses an interesting and urgent topic. However, I have some reservations regarding its novelty. There have been several studies on the effects of pesticides on belowground biodiversity, but the authors did not express well how this study is innovative compared to previous

publications1-5. In addition, this manuscript is more like a report than a research article, as it lacks detailed methodology and discussions.

Thank you for your helpful comments. We have made substantial revisions to address these concerns, as outlined below.

Highlighting the novelty of the study: In the revised manuscript, we have clarified how our study fills critical gaps in the existing literature. Specifically, we are now highlighting the shortcomings of previous studies, such as their geographical limitations, focus on specific organism groups, or restricted to agricultural lands:

“However, these studies have been spatially limited by focusing on specific countries (Riedo et al. 2021, Rivera-Becerra et al. 2017) and agroecosystems (Panico et al., 2022; Rivera-Becerra et al. 2017; Riedo et al. 2021), selected soil biota (Edlinger et al. 2022, Ke et al. 2020; Rivera-Becerra et al. 2017; Gunstone et al. 2021), and on a limited number of pesticide compounds (Tejada et al. 2015).” (lines 53-55):

Also, we have added new data on the effects of pesticides on functional gene diversity, which has never been done before, especially at this large scale (lines 448-449). We also underscored the need for our study:

“Therefore, the impacts of multiple pesticides on complex soil communities at larger geographical scales and across different ecosystem types have never been studied” (lines 55-57). As far as we know, this is the first study at such a large scale and targeting so many different groups of soil biota.

Additionally, we emphasised the innovative aspects of this research by pointing to its empirical, large-scale approach across diverse ecosystems (lines 66-74).

Enhancing methodological details: We expanded the Methods section to improve transparency and provide more comprehensive details. Updates include:

- A detailed description of how GLM models were built and tested (lines 838-886).
- Specifications of the primers and sequencing platforms used for metabarcoding analyses (lines 703-708).
- Clarification on handling pesticide data, such as using 0 as a replacement for concentrations below LOQ (lines 660-661).

To facilitate understanding, we also included a graphical abstract of the methods (Supp. Figure S15), providing an overview of the methodological approach.

Strengthening the discussion: We restructured the Discussion section to incorporate key impact statements and connect findings to policy and regulatory frameworks (lines 384-435). This includes addressing the implications of large-scale assessments for biodiversity conservation and highlighting regulatory gaps that impede comprehensive pesticide monitoring. Additionally, we included the suggested references to contextualise our results within the broader literature.

Furthermore, the authors set out to study the effects of "pesticides" on "soil biodiversity". I see at least two critical issues with their main objective and this study, in that they are too vague, and let me be precise.

1. Pesticides

The authors introduced the concepts of "pesticide occurrence" and "pesticide concentration" to state the impact of pesticides on soil biodiversity. However, the definition and calculation methods for "pesticide occurrence" are lacking. It is useless to simply lump pesticide metrics into the term "pesticide occurrence". Pesticide occurrence does not reflect the amount used, toxicity, pesticide persistence, etc. For example, what does the "increased pesticide occurrence" mentioned in the abstract mean? with high resolution analysis tools you could surely "see" more pesticides in each sample.

We agree that pesticide occurrence is misleading and not well representing all the different metrics evaluated. Thus, in the revised manuscript we replace the term "pesticide occurrence" by "pesticide metrics" instead. We included the definition and calculation methods in lines 669 to 680 and lines 799 to 808.

The selection of pesticide metrics is not convincing. One of the metrics for pesticide occurrence, cumulative risk, is calculated on the basis of the no observed effect concentration (NOEC) for soil arthropods, suggesting that the ecotoxicological risk discussed in the manuscript may not be applicable to the soil microbiota and their interactions (e.g. network analysis).

The no observed effect concentration (NOEC) is not calculated for soil arthropods but rather for several annelids, collembolans, and mites (Franco et al., 2024). We have now included the list of organisms on which the NOEC is established in the Methods section, highlighting the lack of representativeness of several of our organism groups of interest in its calculation (lines 683-687).

There are indeed significant limitations to using the NOEC for risk assessment, which we now discuss in lines 390-396 and in Supplementary results S1. In Figure 2, we explore the relationships (both direct, but also indirect) between biodiversity metrics and pesticide risk using correlations, which make this information limited but still appropriate to our purpose.

Franco, A., Vieira, D., Clerbaux, L-A., Orgiazzi, A., Labouyrie M., Köninger, J., Silva, V., van Dam, R, Carnesecci, E., Dorne J-J. CM, Vuaille, J., Vicente, J. L., Jones, A. Evaluation of the ecological risk of pesticides residues from the European LUCAS soil monitoring 2018 survey. Integrated Environmental Assessment and Management (2024).

Additionally, the authors evaluated the impact of pesticide concentrations relative to key environmental factors. The impact of pesticides on the soil biota largely depends on the type of active ingredients, not just the pesticide concentrations. Different active ingredients have varying ecotoxicities to the soil biota. Hence, the authors might also need to consider the impact of pesticide active ingredients.

Many thanks for this comment. Initially, we also tried GLMs based on the risk quotient (pesticide concentration/NOEC_{min}). These models also displayed varied relationships with different organism groups. However, in order to avoid distortion effects caused by the fact that NOEC is calculated using a very limited number of model organisms, we have not included those models in the revised manuscript.

2. Soil biodiversity

Although the soil biodiversity in this study included soil archaea, bacteria, fungi, protists, nematodes, and arthropods, it is still incomplete and generic. For example, earthworms and plant roots, which are also key soil biota that affect vital soil processes, are not included.

Furthermore, the definition of functional groups, one of the metrics for soil biodiversity, is too

vague. The functional group identifies bacterial or fungal functions on the basis solely of genus taxonomy rather than confirming microbial functional potential at the functional gene level.

We agree that including 6 soil biodiversity groups does not cover the huge diversity present in soils, but our study is the first most comprehensive assessment of soil biodiversity including a wide range of groups of soil biota (e.g. soil archaea, bacteria, fungi, protists, nematodes, and arthropods) as well as new analyses on diversity at the functional gene level that we now added. As we stated in the introduction, previous studies have focused on single or a few organisms from one or a few ecosystem types and have not attempted to cover such a great range of soil types and climates. However, we have added a definition of soil biodiversity in our study: *“This broad scope enabled us to investigate pesticide effects on each selected soil taxonomic and functional groups (hereafter collectively referred to as “soil biodiversity”) at an unprecedented scale and depth.”* (lines 80-82)

In a **Supplementary discussion** (please see Supplementary material), we now acknowledge that other taxa are still missing for a comprehensive overview of the whole soil biodiversity: *“Improved methods (e.g., by considering larger volumes of soil) could capture a wider spectrum of taxonomic groups, including larger and more mobile organisms like annelids (Lilja et al., 2023)”*. The absence of certain organism groups is due to methodological constraints for soil fauna, as well as limited reference databases, which hinder our ability to identify a sufficient number of organisms. This is particularly the case for groups such as tardigrades and rotifers (lines 709-710 of the manuscript). The need for better knowledge on functional annotations was added to the Supplementary discussion: *“Also, deeper knowledge is required at the family, genus or species levels to infer functions, particularly for protists and fauna (Louca et al., 2016)”*.

Regarding annelids, Köninger et al. (2023) reported methodological limitations in metabarcoding analyses of soil samples for detecting macrofauna, i.e. the amount of soil sampled (e.g. 0.6 gram) to isolate DNA was probably too low to obtain a realistic estimate of annelid diversity, and this has also been observed before (Lilja et al. 2023). As a result, Köninger et al. (2023) found little to no variation in annelid richness and diversity across different ecosystem types, also based on the LUCAS Biodiversity dataset (18S data).

However, to tackle your point, we conducted additional analyses accounting for annelids in our investigations and didn't find any significant differences in richness/diversity between ecosystem types (see Pairwise-Wilcoxon tests below) either. Due to the limited quality of the annelid data, 43 sites were discarded from the overall dataset after normalisation of the associated ASV table, resulting in pesticide analyses being conducted on 330 sites instead of 373. Across these 330 sites, the average annelid richness was 5.8 (median = 6), and the average diversity was 1.01 (median = 1.03), indicating a low representativeness of annelids.

Additionally, our dataset only contained one earthworm species, while the remaining organisms were enchytraeids. Given the limited quality of the data, we adopted a more cautious approach until more complete data for annelids becomes available. Therefore, we propose to exclude annelids from our analyses.

```
> pairwise.wilcox.test(df_info_classif_biodiv_env_373sites$anneli_obs, df_info_classif_biodiv_env_373sites$LC1_2018)
```

Pairwise comparisons using Wilcoxon rank sum test with continuity correction

data: df_info_classif_biodiv_env_373sites\$anneli_obs and df_info_classif_biodiv_env_373sites\$LC1_2018

	AC	PC	FC	EG
PC	1.0000	-	-	-
FC	1.0000	1.0000	-	-
EG	0.0018	0.8551	0.8551	-
WL	0.8215	1.0000	1.0000	1.0000

P value adjustment method: holm

```
> pairwise.wilcox.test(df_info_classif_biodiv_env_373sites$anneli_H, df_info_classif_biodiv_env_373sites$LC1_2018)
```

Pairwise comparisons using Wilcoxon rank sum test with continuity correction

data: df_info_classif_biodiv_env_373sites\$anneli_H and df_info_classif_biodiv_env_373sites\$LC1_2018

	AC	PC	FC	EG
PC	1.000	-	-	-
FC	1.000	1.000	-	-
EG	0.027	1.000	0.655	-
WL	0.972	1.000	0.971	1.000

P value adjustment method: holm

Finally, while we discuss the limitations of functional annotations based on taxonomy (Supplementary discussion), we have also expanded the scope of our analyses to metagenomics insights including new data (e.g., see Figures 2 and 4). We considered 48 functional gene groups of archaea, bacteria, fungi and fauna involved in the carbon, nitrogen and phosphorus cycles and performed our analyses (correlation matrix, GLMs with environment and pesticide concentrations and derived outputs) on each functional gene group diversity. This led to additional results and expanded discussion on pesticides' influence on soil community functional profile (e.g., lines 371-383), as well as proposal for future regulatory frameworks (lines 396-401).

Köninger, J., Ballabio, C., Panagos, P., Jones, A., Schmid, M.W., Orgiazzi, A., Briones, M.J.I. Ecosystem type drives soil eukaryotic diversity and composition in Europe. *Global change biology*, 1-14 (2023).

Lilja, M. A. et al. Comparing earthworm biodiversity estimated by DNA metabarcoding and morphology-based approaches. *Applied Soil Ecology* 185, 104798 (2023).

References:

1. Liao, H. et al. Herbicide Selection Promotes Antibiotic Resistance in Soil Microbiomes. *Molecular Biology and Evolution* 38, 2337–2350 (2021).
2. Maggi, F., Tang, F. H. M. & Tubiello, F. N. Agricultural pesticide land budget and river discharge to oceans. *Nature* 620, 1013–1017 (2023).
3. Zheng, X. et al. Organochlorine contamination enriches virus-encoded metabolism and pesticide degradation associated auxiliary genes in soil microbiomes. *ISME J* 1–12 (2022).
4. Liu, Y.-R. et al. Soil contamination in nearby natural areas mirrors that in urban greenspaces worldwide. *Nat Commun* 14, 1706 (2023).
5. Edlinger, A. et al. Agricultural management and pesticide use reduce the functioning of beneficial plant symbionts. *Nat Ecol Evol* 6, 1145–1154 (2022).

Many thanks for providing this relevant literature. We included the references in the revised manuscript.

- We refer to prevailing pesticide contamination as found by Liu et al. (2023) (line 44) and the important role soil life plays in the contaminant's degradation in the introduction (Maggi et al. 2023/Zheng et al. 2022) (lines 50-51).
- We compare our results with those of Edlinger et al. (2022) in the discussion (line 337), who found similar results regarding negative effects on AMF. In the introduction, this study is also mentioned in the context of past studies focusing on specific organism groups (line 54).
- Liao et al. (2021) has been included in the discussion: "For instance, excessive mineral fertilisation may lead to phosphorus loss when combined with organophosphorus pesticide (Liu et al. 2020), while antibiotic contamination can worsen with concurrent pesticide application (Zhao et al., 2023; Liao et al. 2021)." (lines 451-454).

Line-by-line feedback

Line 1: The title is too broad. Is pesticide occurrence or pesticide concentration the main driver of soil biodiversity? Moreover, the data reported in this study do not prove that pesticides are the main driver. In Figure 3, the soil physicochemical properties are shown to be the primary factors.

Following your suggestion and in agreement with reviewer 1, we have changed the title to "Pesticide residues in soils differentially affect soil biodiversity". In the discussion, we have toned down our statement about pesticide concentrations being the main driver of soil biodiversity (lines 317-319): "This study confirms these observations in the belowground environment, indicating that pesticides significantly influence soil biodiversity composition."

Line 41: This manuscript uses pesticide residue concentrations. Are there any reports on pesticide residue concentrations?

We replaced the sales statistics with reports on pesticide residue concentrations found by Vieira et al. (2023) (lines 45-47).

Vieira, D., Franco, A., De Medici, D., Martin Jimenez, J., Wojda, P. and Jones, A. Pesticides residues in European agricultural soils - Results from LUCAS 2018 soil module. (Publications Office of the European Union, Luxembourg, 2023).

Line 60: The article only uses the glyphosate metabolite, aminomethylphosphonic acid (AMPA), and the authors should explain why.

The selection of analytes was defined within the available budget for the LUCAS Survey by general criteria of scientific relevance (potential risk, including the previous 2015 campaign) and policy interest. However, there was no defined criteria, and therefore no explicit reason for exclusion of any substance not included. AMPA is probably the most known metabolite of any pesticide, but almost all active ingredients have relevant metabolites. [Redacted]

In our analysis, AMPA was the only metabolite detected above its limit of quantification (LOQ) in the 373 sites of interest; however, several metabolites were considered at first among the 118 substances tested, for example:

Metabolites of atrazine: atrazine-deisopropyl and atrazine-desethyl.

Metabolites of carbofuran: carbofuran, -3-hydroxy and carbofuran, -keto

Metabolites of DDT: DDD o,p'- (TDE), DDD p,p'- (TDE), DDE o,p'-, DDE p,p'-, DDT o,p'-, DDT p,p'-

Metabolite of terbuthylazine: terbuthylazine-desethyl

Lines 65-68: What were the reasons or criteria that the authors used to select the nine functional groups?

The nine functional groups were selected based on: (i) their presence across ecosystem types, (ii) their relevance as biological indicators (Karpouzias et al., 2022) and (iii) the availability of their functional annotations in reference databases or former studies (Louca et al., 2016; Mazel et al., 2022; Pölme et al., 2020; van den Hoogen et al., 2020). Functional databases hosting taxonomy-based functional annotations for other eukaryotes such as arthropods do not exist yet.

Karpouzias, D. G., Vryzas, Z. & Martin-Laurent, F. Pesticide soil microbial toxicity: setting the scene for a new pesticide risk assessment for soil microorganisms (IUPAC Technical Report). *Pure and Applied Chemistry* 94, 1161-1194 (2022).

Louca, S. et al. High taxonomic variability despite stable functional structure across microbial communities. *Nature ecology & evolution* 1, 0015 (2016).

Mazel, F. et al. Soil protist function varies with elevation in the Swiss Alps. *Environmental Microbiology* 24, 1689-1702 (2022).

Pölme, S. et al. FungalTraits: a user-friendly traits database of fungi and fungus-like stramenopiles. *Fungal diversity* 105, 1-16 (2020).

van den Hoogen, J. et al. A global database of soil nematode abundance and functional group composition. *Scientific data* 7, 103 (2020).

Line 88: Is it appropriate to use pesticide persistence as metrics? Persistence does not directly reflect biological effects, merely the residence time.

Thank you for raising this point. We agree to leave this variable out, and it has been removed from the analyses.

Line 131: The selection of broad-spectrum pesticides for analysis appears to be rational for assessing their widespread biological impacts. However, how is the “broad spectrum” defined? None of the pesticides mentioned are broad-spectrum for archaea or bacteria, which are two important members of “soil biodiversity”.

The broad-spectrum is defined based on the available information in the Pesticide Properties DataBase (PPDB; Lewis et al., 2016). A fungicide is considered broad-spectrum when targeting a broad range of fungal pathogens. For herbicides, it corresponds to a broad-spectrum control of weeds and grasses. For insecticides, it corresponds to a substance targeting a large range of insects (e.g., certain Lepidoptera, cockroaches and ants). We added this explanation to the Methods section in the manuscript (lines 675-680).

Indeed, based on the reported purpose, none of the fungicides, herbicides or insecticides primarily target archaea or bacteria. However, direct and indirect non-target effects of pesticides have been reported in the literature for these groups, e.g., fungicides were found to decrease bacterial

community diversity after direct treatments and dosages (Ma et al., 2021). Similarly, archaea involved in nitrification were impacted by two fungicides in Puglisi et al., 2012.

Lewis, K. A., Tzilivakis, J., Warner, D. J. & Green, A. An international database for pesticide risk assessments and management. *Human and Ecological Risk Assessment: An International Journal* 22, 1050-1064 (2016).

Ma, G., Gao, X., Nan, J., Zhang, T., Xie, X., & Cai, Q. (2021). Fungicides alter the distribution and diversity of bacterial and fungal communities in ginseng fields. *Bioengineered*, 12(1), 8043–8056. <https://doi.org/10.1080/21655979.2021.1982277>

Puglisi, E., Vasileiadis, S., Demiris, K., Bassi, D., Karpouzas, D. G., Capri, E., Cocconcelli, P. S., & Trevisan, M. (2012). Impact of fungicides on the diversity and function of non-target ammonia-oxidizing microorganisms residing in a litter soil cover. *Microbial ecology*, 64(3), 692–701. <https://doi.org/10.1007/s00248-012-0064-4>

Line 133: The relationship between each pesticide and microbiota is not a simple linear relationship but characterized by distinct dose thresholds. Summing concentrations of different types of pesticides or even aggregating concentrations of all pesticides for correlation analysis could lead to confounding results.

We agree that summing concentrations leads to confounding results. However, using aggregated concentrations permits to quantify the overall pesticide effect on biodiversity (via the correlation matrices, e.g., Figure 2). In addition, we have also run generalised linear models with pesticide concentrations (per active ingredient or metabolite), accounting for these concentrations next to ecosystem type, soil properties and climate. To better understand the different analysis conducted, we have included a graphical abstract in the Methods section to better describe our methodological approach (Supplementary Figure S15).

Line 138: How is the multidiversity index calculated? Please provide the calculation formula for multidiversity index.

We have added the formula in **lines 725-728** of the Methods section, which is based on Delgado-Baquerizo et al. (2020).

Delgado-Baquerizo, M. et al. Multiple elements of soil biodiversity drive ecosystem functions across biomes. *Nature Ecology & Evolution* 4, 210–220 (2020).

Lines 140--141: Arbuscular mycorrhizal fungi (AMF) have been defined in the introduction.

Many thanks, we corrected this.

Line143: Please provide the R and p values of the correlation analysis.

We have added examples of values of correlation coefficients and p-values for the correlation analysis (**lines 144 to 151** of the revised manuscript). All coefficient values are available in **Supp. Figure S5**, where only the significant values ($p < 0.05$) are displayed.

Line 127-167: The authors claim there are complex correlations between soil biota to pesticide occurrence. However, the $|r|$ values for the Spearman correlation results in Figure 2 are all less than 0.4. This is not a notable correlation, even if the p value is significant. The data tell us the opposite of what they claim in the manuscript. Note: $|r| > 0.95$ indicates a significant correlation; ≥ 0.8 denotes a high correlation; $0.5 \leq |r| < 0.8$ indicates a moderate correlation; $0.3 \leq |r| < 0.5$ suggests a low correlation; and $|r| < 0.3$ indicates a very weak relationship, which is considered noncorrelated.

We have added the correlation values between parentheses in the revised manuscript (lines 144 to 151) so that the reader is aware of the strength of the relationships presented within the text (next to all numbers already presented in Supp. Figure S5). However, we think it is more important to consider the significance value (this demonstrates that the relationship is not random) than the coefficient value which express the slope of the relationship (i.e., the magnitude of the change of one variable relative to another). It is also important to emphasise that all our data are based on field measurements, which inherently present high spatial and temporal variability, making it difficult to obtain high correlation coefficients as already pointed out in previous works (Brulle et al., 2022; Gunstone et al., 2021). As we cover various landscapes across Europe, different soil types and biogeographical regions, we attribute the weak correlation to the complex context of our study rather than to an absence of relationships with pesticides.

To illustrate this last point, as an example, we have rerun the correlation analyses (for all 63 pesticides) in Italy, where the soil type is predominated by calcisols and cambisols (cf. Soil Atlas of Europe, pages 82-83). With a reduced set of data points geographically localised (4 annual croplands, 6 permanent croplands, 2 croplands recently converted to grasslands and 6 extensive grasslands), the correlation values range from -0.65 to 0.72 . However, high correlation coefficients derived from a small subset of observations risk being biased by local behaviours or outliers. Even in this focused framework, despite having the same soil types, soil properties can also vary greatly within a given soil type. A zoomed-in study looking in closer detail to variations of soil biodiversity at smaller spatial scales focusing on more homogenised soil and climatic conditions might render stronger correlations of soil biodiversity with pesticides, but it clearly diminishes the quality of our results compared to our analysis including 373 sites.

Figure: Correlation matrix between soil biodiversity and pesticide metrics for all 63 pesticides, for sites sampled in Italy.

Brulle, F., Amossé, J., Bart, S., Conrad, A., Mazerolles, V., Néliu, S., Lamy, I., Péry, A., & Pelosi, C. (2022). Toward a harmonized methodology to analyze field side effects of two pesticide products on earthworms at the EU level. *Integrated Environmental Assessment and Management*, 19, 254–271. <https://doi.org/10.1002/ieam.4650>

Gunstone, T., Cornelisse, T., Klein, K., Dubey, A., & Donley, N. (2021). Pesticides and soil invertebrates: A Hazard assessment. *Frontiers in Environmental Science*, 9. <https://doi.org/10.3389/fenvs.2021.643847>

Soil Atlas of Europe, European Soil Bureau Network European Commission, 2005, 128 pp Office for Official Publications of the European Communities, L-2995 Luxembourg

Line 152: In Figure 2, it is evident that cumulative concentration in croplands shows a positive correlation with archaeal richness, whereas insecticide concentration exhibits a negative correlation with archaeal richness. The authors should explain these results.

Thank you for raising this point. In Figure 2, the variable “Insecticides” corresponds to the number of insecticides detected, not their concentration. In addition, the cumulative concentration of pesticides is not correlated to the number of insecticides (Pearson’s $r=0.062$, $p\text{-value} = 0.3336$), which explains the divergence in relationships with archaeal richness in croplands. We have modified the y-axis and the caption of Figure 2 to indicate that “Insecticides” refers to the number of pesticides residues detected classified as insecticides, and not to their concentration.

Line 176: Based on a linear model or a unimodal model? Was detrended correspondence analysis conducted for model selection? Was collinearity among factors assessed? The authors should provide detailed descriptions of these methods in the methodology section.

The variation partitioning was performed on the generalised linear models (see Methods section, paragraph "Pesticide concentrations" lines 888-890, and lines 899-916) including any (distinct) pesticide concentrations and environmental variables that were identified as relevant predictors during the feature-selection steps. Collinearity among factors was reduced using feature-selection procedure (stepAIC function from MASS package) and a pre-filtering of important environmental background to avoid the biased selection of pesticide concentrations highly correlated with underlying environmental variables (e.g., soil properties, climatic conditions). We have now added additional sentences describing the use of variation inflation factor (VIF) on the final models to detect any remaining collinearity (lines 878-880) and discuss the two models where collinearity was detected in Supplementary results S4.

In addition, we provide a correlation matrix between all environmental variables and pesticide concentrations in Supplementary Fig. S12 of the Supplementary material, and Supplementary Fig. S4 of Supplementary Data File S3 for analyses of croplands only.

Line 252: The author needs to further describe and discuss the fitting results of the generalized linear model.

Thank you for pointing this out. While checking for performance and fitting of used generalized linear models, we realised that there is no consensus regarding the pseudo-R² metric for reporting the performance of these models (as discussed in Hemmert et al., 2018). In the updated manuscript, we replaced previous pseudo-R² by the squared correlation between predicted and observed values as an r² for model performance, together with other known metrics (e.g., RMSE, MAE, MSE). Please note that results remained consistent.

In addition, we have added a Supplementary results S4 section, discussing:

- the model performance (based on r² presented in Supplementary Tables S20 and S21 of Supplementary Data File S2 for metabarcoding models in all ecosystems and Supplementary Tables S12 and S13 of Supplementary Data File S4 for analyses in croplands only, as well as in Supplementary Tables S22-S24 of Supplementary Data File S2 for the metagenomics models in all ecosystem types or Supplementary Tables S14-S16 of Supplementary Data File S4 when focusing on croplands only).
- the QQ plots and residuals patterns (obtained with the simulateResiduals function from the DHARMA package)
- the interpretation of the shared variance in the variation partitioning analyses
- the interpretation of models showing collinearity

Additionally, we have extended the Methods section to further describe these methods:

"Model diagnostics also included QQ plots (using the simulateResiduals function from the DHARMA package) to assess the normality of residuals (see DHARMA plots files in outputs folders together with the R scripts from the Code availability section). Collinearity within all the fitted models was assessed using the vif function from the car package. Multicollinearity was detected in two models, and this is further discussed in Supplementary results S4. Spatial autocorrelation was tested on the residuals of each model using the geoR package, revealing no clear geographical trends." (lines 875-881)

Hemmert, G. A. J., Schons, L. M., Wieseke, J. & Schimmelpfennig, H. Log-likelihood-based pseudo-R² in logistic regression: Deriving sample-sensitive benchmarks. *Sociol. Methods Res.* 47, 507–531 (2018).

Lines 258-267: Environmental factors can influence the persistence of pesticides in soil, as mentioned in this section. A relevant question is, how did authors obtain pesticide persistence data (lines 83--167)? Did they directly use the data from soil samples?

We obtained this data from the Pesticide Properties DataBase (PPDB; Lewis et al., 2016, lines 98-100 of the Results and lines 671-673 of the Methods section). The persistence is based on reported Dt50 values in soils, i.e., the time required for the pesticide concentration to decline to 50% of the initial amount applied under field studies. This information is now included in the revised manuscript.

Lewis, K. A., Tzilivakis, J., Warner, D. J. & Green, A. An international database for pesticide risk assessments and management. *Human and Ecological Risk Assessment: An International Journal* 22, 1050-1064 (2016).

Additionally, the use of the Kruskal–Wallis test to analyse potential environmental conditions that may enhance the persistence of individual pesticides in soil is not rigorous. The results of the Kruskal–Wallis test can indicate significant differences, but the actual connection between environmental conditions and pesticide persistence requires experimental validation or more detailed data analysis, such as soil organic matter contents etc..

Thank you for raising this issue. We have further developed the need for experimental validation in the Discussion: *“Moreover, our findings reveal that soil type and climate correlate with pesticide presence in shaping soil biodiversity. Future studies should cover a wider range of soil types across different climatic zones to experimentally validate their interactive role in pesticide persistence in soils”* (lines 432-435).

Lines 277-279: The effect of pesticide concentration is likely one of the most complex factors. The authors need to clarify the nonlinear and intricate interactions between pesticides and microbiota. Is there a threshold for pesticides to impact the soil biota?

Reference: <https://doi.org/10.1038/s41467-023-41258-x>

Thank you for raising these crucial points. We modeled the complex relationships between pesticide concentrations (predictors) and soil (micro-) biota (response variable) using generalised linear models (GLMs) with a Gaussian log-link function. However, we now refer to these GLM models with caution (lines 871-873) as they might represent an approximation to the nonlinear, intricate interactions between pesticides and soil biodiversity. We also acknowledge that these models have been found reliable in analysing ecotoxicological effects of pesticides on soil biodiversity in previous studies (Szöcs & Schäfer, 2015; Szöcs et al., 2015).

Currently, no established threshold exists for determining the impact of pesticides on soil biota. This important research gap in the literature largely reflects limitations of past studies, which are often limited to a single pesticide type, single or few soil organism groups, or restricted in their geographical scales. Our study is the first step forward in addressing these limitations, though it was only feasible to include a subset of soil organisms or influential variables.

Pesticide impact likely depends on several factors, including the specific pesticide concentrations and the baseline conditions of soil communities before pesticide exposure. However, our dataset

does not contain this type of information, which would be necessary to define such thresholds across EU soils (lines 436-438, 445-447, 473 of the Discussion).

Our findings underscore that pesticides influence belowground biodiversity in different ways and highlight the need for more comprehensive studies, including exploratory and laboratory studies, to establish clear guidelines.

Szöcs, E. & Schäfer, R. B. Ecotoxicology is not normal: A comparison of statistical approaches for analysis of count and proportion data in ecotoxicology. *Environmental Science and Pollution Research* 22, 13990-13999 (2015).

Szöcs, E. et al. Analysing chemical-induced changes in macroinvertebrate communities in aquatic mesocosm experiments: a comparison of methods. *Ecotoxicology* 24, 760-769 (2015).

Lines 285-291: This statement seem to be a repetition with the results? The discussion should be a further analysis of the results, exploring deeper scientific issues as well as the significance of the study. It has been widely reported that pesticides can increase or decrease diversity. The authors need to further discuss this result.

Many thanks for raising this point. We have edited this paragraph and rearranged the discussion to better link our findings to the implications of potential community alterations and competitive advantages in response to pesticides (lines 328-330).

References:

1. Zhao, F., Yang, L., Yen, H. et al. Reducing risks of antibiotics to crop production requires land system intensification within thresholds. *Nat Commun* 14, 6094 (2023).
2. Ke, M. et al. Development of a machine-learning model to identify the impacts of pesticides characteristics on soil microbial communities from high-throughput sequencing data. *Environ Microbiol* 1462-2920.16175 (2022)
3. Guinet, M., Adeux, G., Cordeau, S. et al. Fostering temporal crop diversification to reduce pesticide use. *Nat Commun* 14, 7416 (2023).
4. Wang, C. N. et al. Effects of pesticide residues on bacterial community diversity and structure in typical greenhouse soils with increasing cultivation years in Northern China. *Sci Total Environ*, 710, 136321 (2020).

Many thanks for the suggested references. We see them complementing our statements in various sections of our manuscript. Please find their integration as follows:

- We integrated **Zhao et al. (2023)** in the discussion: "*For instance, excessive mineral fertilisation may lead to phosphorus loss when combined with organophosphorus pesticide (Liu et al. 2020), while antibiotic contamination can worsen with concurrent pesticide application (Zhao et al., 2023; Liao et al. 2021).*" (lines 451-454)
- We integrated **Ke et al. (2022)** discussing the limitations of our study: *Other pesticide properties, such as dissociation constant, molecular weight and water solubility could not be considered here, despite known impacts on bacterial diversity (Ke et al. 2022, Liao et al. 2021)* (lines 442-444)

- Also in the discussion, we integrated **Guinet et al. (2023)**: *“In addition, future studies should incorporate detailed land management data, as it sheds light on pesticide application patterns—such as lower pesticide use in crop rotations (Guinet et al. 2023) compared to greenhouse cultivation (Angioni et al. 2012, Bojaca et al. 2013).”* (lines 449-451)
- We referred to **Wang et al. (2020)** in the introduction, stating that the persistency of pesticides depends on their chemical properties (lines 44-45). We used this reference also to emphasise the importance of continuous studies: *“This information is crucial to enhance future studies’ reliability, especially given that pesticide responses may vary across different years (Wang et al. 2020).”* (see Supplementary Discussion)

Line313: What is the connection between bacterivore nematodes and plant productivity?

Bacterivore nematodes contribute to nutrient availability by stimulating the mineralisation and decomposition of organic matter. We added this information and cited Trap et al., 2016. This explanation has been now included in the paragraph dealing with the effects on beneficial organisms in the discussion section (lines 338-340).

Trap, J., Bonkowski, M., Plassard, C., Villenave, C. & Blanchart, E. Ecological importance of soil bacterivores for ecosystem functions. *Plant and Soil* 398, 1-24 (2016).

Lines 549-553: Why use Amplicon Sequence Variants (ASVs) for 18S analysis and operational taxonomic units (OTUs) for others? There are two completely different algorithms: OTUs are based on clustering, whereas ASVs involve denoising.

The decisions were taken based on the most suitable method for each dataset. ASVs have been found to be the most reliable method for assessing the 18S gene (Bukin et al. 2023) and hence, used in recent studies on soil protists and soil fauna using 18S data (Königer et al., 2023; Oliverio et al., 2020, Wang et al. 2024). For bacteria, we used zero-radius OTUs (zOTUs) that also represent exact sequence variants (100% similarity), not clustered sequences at a certain distance threshold. However, previous results from a study investigating the patterns of soil bacteria across Europe showed consistent outcomes between 99%-OTUs and zOTUs (unpublished data from Labouyrie et al., 2023). In addition, ASVs are not suitable for analysing data obtained with the PacBio platform, i.e., in the case of our archaeal and fungal datasets (Tedersoo et al., 2022), nor assessing full-length ITS sequences, due to random PCR errors, the presence of multiple/highly similar copies of the ITS region in eukaryote genomes (Lindner et al. 2013) and an increased elimination of taxa that are both rare and phylogenetically unique (Joos et al., 2020). Therefore, archaeal 97%-OTUs were consequently used in this study following Grant et al. (2023) and for fungi, we used 98%-OTUs as recommended in Tedersoo et al. (2021).

For 18S data, we compared the richness and Shannon diversity values obtained with a 97%-OTU table to the ones obtained with ASVs (see figure below). The smaller the organism group in size, the higher was the correlation obtained for richness (i.e., 0.90 for protists, 0.54 for nematodes and 0.49 for arthropods). A similar trend was observed for the Shannon diversity index, but with weaker correlation values (0.42 for protists, 0.23 for nematodes and 0.25 for arthropods). This further illustrates the consistency in outcomes between 18S 97%-OTUs and ASVs in our analyses. However, it also evidences the limitations of the dataset for bigger organism groups that are less well represented due to methodological constraints (see above). For annelids, 97% OTUs did not lead to

any results since no appropriate sequencing depth (Cmin) was possible to select in the normalization step. This finding provides further support for leaving the annelid data out of the analyses.

Consequently, for comparison with future studies, we propose to perform the pesticide analyses at a 100% similarity threshold for the 18S and bacterial 16S data, while keeping OTUs for the PacBio-derived datasets (16S archaea and ITS fungal data).

Bukin, Yuri S., et al. "The effect of metabarcoding 18S rRNA region choice on diversity of microeukaryotes including phytoplankton." *World Journal of Microbiology and Biotechnology* 39.9; 229 (2023).

Grant, A., Aleidan, A., Davies, C. S., Udochi, S. C., Fritscher, J., Bahram, M., & Hildebrand, F. (2023). Improved taxonomic annotation of Archaea communities using LotuS2, the Genome Taxonomy Database and RNAseq data. *bioRxiv*, 2023–08.

Joos, L. et al. Daring to be differential: metabarcoding analysis of soil and plant-related microbial communities using amplicon sequence variants and operational taxonomical units. *BMC Genomics* 21, 733 (2020).

Köninger, J., Ballabio, C., Panagos, P., Jones, A., Schmid, M.W., Orgiazzi, A., Briones, M.J.I. Ecosystem type drives soil eukaryotic diversity and composition in Europe. *Global change biology*, 1-14 (2023).

Labouyrie, M., Ballabio C., Romero F., Panagos P., Jones, A., Schmid, M.W., Mikryukov V., Dulya O., Tedersoo L., Bahram, M., Lugato, E., van der Heijden, M.G-A., Orgiazzi, A. Patterns in soil microbial diversity across Europe. *Nature Communications* (2023).

Lindner, D. L. et al. Employing 454 amplicon pyrosequencing to reveal intragenomic divergence in the internal transcribed spacer rDNA region in fungi. *Ecol. Evol.* 3, 1751–1764 (2013).41.

Oliverio, A. M. et al. The global-scale distributions of soil protists and their contributions to belowground systems. *Sci Adv* 6, eaax8787 (2020).

Tedersoo, L., Bahram, M., Zinger, L., Nilsson, R. H., Kennedy, P. G., Yang, T., Anslan, S., & Mikryukov, V. (2022). Best practices in metabarcoding of fungi: From experimental design to results. *Molecular ecology*, 31(10), 2769–2795. <https://doi.org/10.1111/mec.16460>

Tedersoo, L. et al. The Global Soil Mycobiome consortium dataset for boosting fungal diversity research. *Fungal Diversity* 111, 573–588 (2021).

Wang, Haotian, et al. "Profiling the eukaryotic soil microbiome with differential primers and an antifungal peptide nucleic acid probe (PNA): Implications for diversity assessment." *Applied Soil Ecology* 200, 105464 (2024).

Lines 559-561: Please provide the calculation formula.

We have added the formula **line 725** of the Methods section (following Delgado-Baquerizo et al. 2020).

Delgado-Baquerizo, M. et al. Multiple elements of soil biodiversity drive ecosystem functions across biomes. *Nature Ecology & Evolution* 4, 210–220 (2020).

Referee #3 (Remarks to the Author):

This manuscript describes field monitoring of combined soil pesticide residues and soil biota in over 300 sites in 26 European countries covering different ecosystems (annual and perennial cropland, grassland and woodland) and part of the LUCAS monitoring campaign in 2018. The paper attempts to relate the composition (richness, diversity and functional group abundance) of the soil biota to different pesticide residue metrics (number of detected pesticides, cumulative (additive) concentration and risk and individual pesticide residue concentrations), as well as ecosystem type, soil properties and climate variables and determine the most important variables for structuring the soil community. I think it is a potentially important study, though I have some issues with the inclusion of an excessive number of both dependent and predictor variables that in some cases are very correlated, many many different models are formulated rather than composing models that considers the data complexity (hierarchical or multivariate), the lack of considering other (agricultural) management related variables that are not monitored but of potential high importance in shaping soil communities and possibly correlated with the pesticide residues as well as the method to infer organismal abundance underlying the functional groups.

To summarize my major criticisms:

1) There are an excessive number of variables used for both the dependent side, but in particular for the predictor side of the x-y relations. It is evident from supplementary figure S6 that there is high correlation (mostly $r > 0.8$) among number of pesticides, herbicides, fungicides, mode of action, chemical groups, persistence, broad-spectrum, meaning that these variables would mostly explaining the same variance in dependent variables. Thus, only one of these should be used in analyses (I suggest number of pesticides detected for simplicity and comparability with others work) and it should be made clear that it would represent a number of other pesticide related metrics that are impossible to separate. The other metrics (cumulative concentration and risk + number of detected insecticides) are not as correlated and could be considered for inclusion. In addition to these metrics, individual concentrations are also included in some of the analyses, without considering their interrelatedness and correlation with the compounded pesticide metrics.

We thank you for your detailed feedback and thoughtful comments, which have significantly helped us refine and focus the manuscript, particularly the Methods section.

In response to your observations, we identified and acknowledged limitations in the formulation and presentation of the methodology used in this study. As a result, we have restructured and revised the Methods section to provide clearer explanations on the decisions made. Additional information has been incorporated to better justify our approach. Below is a non-exhaustive list of additional information that we hope addresses your concerns:

- we have reduced the Results section on the positive/negative relationships derived from the GLMs (lines 139-154) and placed some further details in Supplementary results S2;
- we approached some caveats on DNA-based analyses and functional annotations in a Supplementary discussion (please see the Supplementary material);
- we included details on which organisms are considered into the NOEC calculations (please see Methodes, lines 683-687);
- we reordered the Methods' Statistical section for better readability;
- we extended the description on how GLMs were designed and tested (please also see the Statistical section in the Methods);
- we have included a graphical abstract of the methodology to illustrate the analyses performed (see Supplementary Fig. S15);
- we have provided a correlation matrix between all variables initially considered as predictors in the GLMs (see, e.g., Supp. Figure S12, and Supp. Figure S4 of Supplementary Data File S3), and displayed their highest correlations ($|r| > 0.5$) (Supplementary Table S10 of Supplementary Data File S2 and Supplementary Table S9 of Supplementary Data File S4).

Below, we address your specific concerns regarding the number of variables, their correlations, and interrelatedness in greater detail.

Addressing the use of highly correlated pesticide metrics: We see your concern regarding the highly correlated pesticide metrics (e.g., number of pesticides, herbicides, fungicides, etc.). However, they are used exclusively for the correlative analyses between soil biodiversity metrics and pesticide metrics (e.g. Figure 2). These are simple pairwise Spearman correlations and do not involve any additional modelling or statistical interpretation beyond correlation coefficients. This approach allows us to explore potential associations without overstating causal implications. They are relevant metrics often used in ecology and policy-oriented studies on pesticide risk assessments for soil biodiversity, and hence, it is appropriate to report them in our manuscript, to enable future comparisons (Supp. Figure S5). However, in the main manuscript we focus on the most common ones (Figure 2). Pesticide persistence was removed in the revised manuscript, as well as the systemic access information (based on your comments below).

Handling individual pesticide concentrations in GLMs: For the GLMs, only individual pesticide concentrations are included, and they are not analysed with the rest of the pesticide metrics, to avoid issues of redundancy or interrelatedness. We can confirm that our modelling approach clearly separates these two types of analyses, minimizing the risk of confounding effects. Especially, the GLMs are done in two steps, but only the (final) GLMs from the second step are used for downstream analyses (we now emphasise this more in lines 849-850 and 863-870).

The first step of the GLMs is selecting the environmental drivers (soil properties, ecosystem type, climate). The second step is adding pesticide concentrations next to the previously selected environmental variables to see if the pesticide concentrations improve the model. A pre-filtering of the pesticide concentrations showing no or very little variation across sites is done between the first and second steps. In the second step, the remaining individual pesticide concentrations are added

together in the models from the first step (i.e., [pesticide A] + [pesticide B] + [pesticide C] + environment). The pesticide concentrations (A and/or B and/or C) will be retained in the model by a feature-selection step if they improve the model on top of the environment. Therefore, it is possible that no pesticide concentrations are selected at all, just one of them, or several.

From these last GLMs (including environment and pesticide concentrations), further analyses are derived, but no additional models are created per se:

- The variable importance (VIP) analyses use these GLMs (environment + pesticide concentrations) to hierarchise the most important pesticide concentrations retained as predictors
- The variation partitioning analyses also use the same GLMs to assess the unique and shared variance explained by each driver type, based on the principle of the varpart function from vegan R package (Oksanen et al., 2010), adapted to our GLMs. The four driver types considered are pesticide concentrations (the ones retained in the GLMs, if any), soil properties, climatic variables and ecosystem (the ones retained in the GLMs as well, if any, respectively).
- The Kruskal-Wallis tests assess the statistical significance of the differences between soil properties or climatic variables retained by the GLMs (during the first step), in soils where the pesticide concentration is positive versus soils where the pesticide concentration is not detected above LOQ. This is done for each pesticide concentration retained in the same GLMs during the second step, such that if the GLMs corresponds to:

Soil property S1 + Soil property S2 + Climate C1 + [Pesticide A] + [Pesticide B] + Ecosystem type

we tested if the average value of S1 differs between sites where Pesticide A is present or absent, then tested the same between sites where Pesticide B is present or absent, then moved on to assess any difference in S2 in the presence or absence of Pesticide A, Pesticide B, then move on to C1 in a similar way.

The two-step GLMs were thus done for each soil biodiversity metric, but only the GLM from the second step, including both environmental variables and pesticide concentration(s), is used for further analyses, such as VIP analyses and variation partitioning analyses. We thus work with 22 final models for metabarcoding data: 6 models for richness – one for each soil taxonomic group, 6 in total for their diversity, 9 for the relative abundance – one for each functional group, and 1 for multidiversity. In addition, comparable analysis for metagenomics functional data have been added (one model per functional gene group diversity).

Finally, all analyses were performed twice: (i) including all ecosystem types (e.g., n=373 sites) and (ii) croplands only (e.g., n=244 sites). The reasons for conducting separate analyses for croplands are developed in the detailed responses to your associated comments below.

Evaluating and managing potential collinearity in GLMs: We recognise the importance of addressing collinearity in our models. To this end, we have applied the variance inflation factor (VIF) test using the vif function from the car package to detect potential multicollinearity in the final GLMs. We have now explicitly included this procedure in the Methods section (lines 878-881) and describe the results in Supplementary results S4.

We address the rest of the concerns on land management practices and abundance underlying the functional groups below.

Oksanen, J. Vegan: community ecology package. <http://vegan.r-forge.r-project.org/> (2010)

This is a monitoring study with the strength of measuring the cumulative (mixture of) exposure to pesticide, it is not an experiment suitable for linking soil biota metrics to specific concentrations of individual substances (this is better done in e.g. empirical papers underlying the topically related meta-analysis by Beaumelle et al. 2024 (<https://doi.org/10.1111/1365-2664.14437>)). Inclusion of this part risk to be misleading rather than informative, based on both the correlation among pesticide metrics, the lack of including management related variables (point 3 below) and the uncertainties in the abundance inference underlying functional groups (point 4 below).

Since active ingredients differ according to application rates, detection limits, persistence, modes of action, and chemical properties, just considering cumulative pesticide concentration in GLMs (as used in Figure 2) would have offered only a limited perspective. To address this, our GLMs include individual pesticide concentrations to detect more specific positive or negative relationships with soil biodiversity metrics. These relationships are further supported by our partial plots, which isolate the effect of each pesticide concentration while accounting for other predictors in the models. We have now justified the use of partial plots in the Methods (lines 924-927) sections.

We address points 3 and 4 below.

For the dependent side, is there really value in presenting both richness and diversity metrics? At least consider putting one of these in the supplementary part and keeping the other in the main manuscript OR make a point of keeping them both in the manuscript and be clear about why including them both are informative (e.g. better highlight IMPORTANT differences and discuss why they may differ).

Richness captures the number of taxa present (independently of whether they are abundant or rare), while diversity accounts for the evenness of their distribution, which may respond differently to environmental stressors like pesticides. By presenting both, we provide a more comprehensive view of the impacts. To justify their inclusion, we have added the following explanation in the revised manuscript: *“Interestingly, richness and diversity metrics exhibited dissimilar results (Fig. 4, Supplementary Fig. S13, Supplementary Fig. S2 of Supplementary Data File S3), emphasising the different aspects of soil biodiversity they capture (Morris et al. 2014).”* (lines 252-254). This reference concludes based on several other studies finding similar results including Heino et al. (2008) and Stirling et al. (2001).

In particular, Stirling et al. (2001) suggest to present richness counts together with Shannon diversity

Heino, J., Mykrä, H., & Kotanen, J. (2008). Weak relationships between landscape characteristics and multiple facets of stream macroinvertebrate biodiversity in a boreal drainage basin. *Landscape Ecology*, 23, 417-426.

Morris, E. Kathryn, et al. Choosing and using diversity indices: insights for ecological applications from the German Biodiversity Exploratories. *Ecology and evolution*, 2014, vol. 4, no 18, p. 3514-3524.

Stirling, G., & Wilsey, B. (2001). Empirical relationships between species richness, evenness, and proportional diversity. *The American Naturalist*, 158(3), 286-299.

In addition, perhaps Beaumelle et al. (2024) could be informative in guiding the use of richness, diversity and abundance/biomass, with their conclusion that richness and diversity were more impacted by pesticides compared to abundance and biomass.

We agree and that is why it is so important to include several biodiversity metrics. In addition, following a request from reviewer 1, we have performed new analyses using bacterial and fungal biomass data from fatty acid methyl esters. We present these data in response to their comments (see pages 10-12 of this document) for croplands only (n=233 sites, 201 annual croplands and 32 permanent croplands). We compared the outputs from GLMs run for bacterial and fungal richness and diversity with those for bacterial and fungal biomass. We also run a GLM for the fungal-to-bacterial biomass ratio. Additionally, we performed correlation analyses between the new biomass metrics and the pesticide metrics (e.g., number of pesticides, of insecticides, of fungicides, of herbicides etc., similar to Figure 2).

The results from these analyses suggest that, in the presence of certain pesticides, soils may become dominated by a few more resilient taxa (with increased biomass of these groups), while more sensitive or less competitive taxa decline or disappear, leading to reduced OTU richness and diversity. Although our findings indicate potentially interesting patterns that could complement the work from Beaumelle et al., we believe the extensive metabarcoding analyses (presented in the initial version of the manuscript), combined with the new metagenomics analyses, already provide substantial insights into soil biodiversity complexity. This is further underscored by the fact that the biomass dataset represents only a subset of our sites (croplands) and a subset of our organism groups of interest (bacteria and fungi). However, if you feel they add value to our reported findings, we are happy to include them.

2) I can't even count how many individual statistical analyses that are done for this work; partly because there are so many dependent and predictor variables and partly because models are built for parts of the data rather than building fewer models that include interactions to evaluate e.g. the importance of sampled ecosystem or organism group in relation to the pesticide metrics using general/generalized mixed models or multivariate analysis approaches.

We have clarified some misunderstandings about the type of analyses performed and the justification behind them in this response letter but also in the revised manuscript. In addition, to improve the description of our methodology, we have added a graphical diagram illustrating the different analyses performed, and edited the Methods section to better describe/justify our approaches (Supplementary Figure S15).

Briefly, we have conducted our analyses for all ecosystem types (373 sites) as well as for croplands only (244 sites). Since some pesticides are predominantly found in croplands, GLMs were built specifically for cropland sites to better account for these pesticides. Otherwise, eight of the pesticide concentrations would have near-zero variance across all ecosystem types and would be discarded

from the analyses, despite their high relevance in croplands. This partial analysis aims to compare with the models based on the overall dataset and to test their consistency with independency of ecosystem type.

Also the GLMs and variance partitioning to identify influential variables need additional methods details, e.g. always stating the R package and function(s) used.

Thanks for highlighting this. In the Methods, we have added more information on how GLMs were built, including the main R packages and functions used. We hope that it is clear now that the same generalised linear models (GLMs) were used as the basis for the outputs produced.

In addition, I can't see any detailing of how well the chosen model(s) fit the data (residues) and if/how this has been evaluated even if this box is ticked in the reporting summary ("A description of any assumptions of corrections, such as test of normality..."). Let me know where it can be found if I have overlooked this.

Thank you for pointing this out. We agree that providing more details on the development and fitting of the generalised linear models (GLMs) would improve the transparency of our approach and clarify the decisions made. We have consequently expanded the Methods section by adding the following text:

"Model diagnostics included QQ plots (using the simulateResiduals function from the DHARMA package) to assess the normality of residuals (see DHARMA plots files in outputs folders together with the R scripts from the Code availability section). Collinearity within all fitted models was assessed using the vif function from the car package. Multicollinearity was detected in two models, and this is further discussed in Supplementary results S4. Spatial autocorrelation in the model residuals was tested using the geoR package⁹⁶, revealing no clear geographical trends." (lines 875-881)

3) It is great that you have included considerations of ecosystem, soil and climatic properties in addition to the pesticide residues. However, there is a lack of information and considerations of other, potentially related, agricultural management variables in addition to the pesticide use aspect. Since other agricultural management activities such as fertilizer regime and soil disturbance could be related to pesticide use regimes, caution should be taken when concluding on causal impact of pesticide residue on the soil biota community composition. If I understand correctly, agricultural management aspects were not monitored in LUCAS 2018, so this information can't be explored or considered? In that case, it is rather a caution in formulations and bringing this aspect up for discussion that is needed in the manuscript – something that I can't find currently.

Thank you for this comment. Unfortunately, we do not have quantified information on land management practices in the LUCAS Soil survey. As an alternative, we have considered an additional set of categorical variables for land management in our generalised linear models (GLMs), next to pesticides, soil properties, climate and ecosystem type. These variables were established by trained surveyors during soil sampling and included signs of ploughing (yes or no), presence of crop residues (yes or no), presence of grazing (yes or no), presence of grass margins (no, presence of grass margin of less than or equal to 1m width, presence of grass margin of more than 1m width), sign of water management (e.g., irrigation, drainage) or not. In addition, the degree (%) of soil imperviousness associated with land management was measured via satellite data (COPERNICUS).

These variables were, however, closely related to ecosystem types (e.g. annual croplands being ploughed). As a result of this redundancy, they did not improve the model explanation (e.g., in some cases, land practices were chosen instead of the ecosystem type) and raised some collinearity issues in the models (based on VIF values). For the models in which land management practices were retained (without collinearity issues), variation partitioning analyses showed that pesticide concentrations were still selected, explaining a unique part of variance that land practices did not capture.

Increasing the number of predictors might be skewed due to our number of observations (i.e., dimensionality issue). Therefore, we propose to leave these land practices variables out. However, in the revised manuscript we have added a critical discussion on the limitations of our interpretations due to the absence of quantitative data on land management practices (lines 449-457):

“In addition, future studies should incorporate detailed land management data, —such as lower pesticide use in crop rotations (Guinet et al. 2023) compared to greenhouse cultivation (Angioni et al. 2012, Bojaca et al. 2013) —and their intertwined effects on soil processes and biota. For instance, excessive mineral fertilisation may lead to phosphorus loss when combined with organophosphorus pesticide (Liu et al. 2020), while antibiotic contamination can worsen with concurrent pesticide application (Zhao et al., 2023; Liao et al. 2021). Additionally, conventional tillage can increase pesticide leaching (Summerton et al. 2023), while conservation tillage practices — often involving pesticide-treated seeds—may shift fungal communities (Mackay et al. 2023). Data on such practices could help disentangle land management effects from pesticide impacts on soil biodiversity.”

4) Organismal abundance is inferred based on read counts from the DNA metabarcoding. I’m not an expert on this topic, but even I know that this is a highly debated topic because of methodological challenges, e.g. biomass across and within species, taxon specific primer efficiency and amplification and post lab data processing (e.g. reviewed and discussed for terrestrial arthropods in Sickel et al. 2023 (<https://doi.org/10.3897/mbmg.7.112290>)) as well as the challenge of relict DNA remains in soil samples. I understand that you are trying to make the best out of this data, but I lack any discussion on this caveat (can be brief in the manuscript and more extended in the supplement) and much more details on the underlying methods.

We agree that there are several caveats in DNA metabarcoding that need to be fully considered for the interpretation of our results. We therefore added a section as a Supplementary discussion that emphasises the potential impact of relic DNA in our interpretations. This aspect was already highlighted in our previous study (Königer et al. 2023). In addition, we have also indicated how to improve current DNA methodologies to overcome some of these shortcomings: *“However, DNA from deceased organisms (necromass DNA) may accumulate in soils, potentially obscuring the active community profile. To disentangle this “DNA memory effect” from the current community composition, methods targeting intracellular DNA or RNA could offer valuable insights into community dynamics.”* (please see Supplementary discussion in the Supplementary material)

Königer, J., Ballabio, C., Panagos, P., Jones, A., Schmid, M.W., Orgiazzi, A., Briones, M.J.I. Ecosystem type drives soil eukaryotic diversity and composition in Europe. *Global change biology*, 1-14 (2023).

You are referring to other papers (62-64) but that doesn’t exclude the need for a method summary also here, that includes the main considerations that the reader should keep in mind.

Many thanks for pointing this out. We added a method summary including the primers and sequencing platforms used (PacBio for 16S archaea and ITS fungi, Illumina for 16S bacteria and 18S data) as follows: *“Specific primers were used to amplify these regions: SSU1ArF and SSU1000ArR for archaeal 16S rRNA, 515F and 926R for bacterial 16S rRNA, ITS9mun and ITS4ngsUni for fungal ITS, and Euk575F and Euk895R for eukaryotic 18S rRNA. Bacterial and eukaryotic sequencing was performed using the Illumina MiSeq platform, while archaeal and fungal sequencing used the PacBio Sequel II platform.”* (lines 703-708)

Here I’m also wondering if the richness (i.e. number) of ASVs/OTUs could be used in place to form the functional groups and the dependent variables used in the analysis since these metrics are less questioned than the methods for abundance estimation.

We think that calculating the relative abundance of taxa is more relevant than richness as it avoids giving the same weight to OTUs with high richness but not being well represented (low counts) or vice versa.

That said, this seems like an interesting, timely and potentially significant study that could substantially moves along our understanding of how real-world pesticide exposure via soil might be affecting the soil biota community in Europe – though the above and below points should be addressed. Since environmental risk assessment (ERA) of pesticides usually consider the potential impacts of exposure to one (or perhaps a few if co-formulated in a single product) active ingredients for a set of model species (mainly annelids), the reality is that organisms encounter pesticide exposure as mixtures. Studies like can provide insights to how the ERA process could be revised or supplemented to cover mixtures and the diversity of the soil biota. In relation to this, it would be interesting to have an improved and extended discussion in relation to the conclusions made in Franco et al. (2024, <https://doi.org/10.1002/ieam.4917>) in a mixture ERA context. E.g. the ecological risk concluded there based on standard toxicity information vs the soil biota data that you have included here.

Thank you for this suggestion. We have now further discussed the importance of assessing the effects of pesticide mixtures (see lines 405-408, 436-438, 472 in the discussion). Additionally, Franco et al. (2024) evaluated the potential risk to non-target soil macroinvertebrates using conventional regulatory risk assessment for pesticide mixture, but many organism types (e.g., archaea, bacteria, fungi) are not accounted for when calculating the risk based on the no observed effect concentration (NOEC). We now highlight the discrepancy between the organisms considered in risk assessments and the organisms included in our study (see lines 393-396 in the discussion, lines 688-689 in the Methods section, and Supplementary results S1).

Without the full considerations of more complex soil communities and functional groups (including non-target ones) and the potential cascade effects on the soil food web, the current regulatory assessment framework remains limited and fails to provide a valid threshold for “acceptable doses” of pesticides (lines 396-398). These points should be investigated in future experiments, as we indicate in the discussion section, by (a) better highlighting the shortcomings of current risk assessments (lines 400-411) and (b) aiming at better guiding future research to overcome current regulatory gaps (lines 442-445, 457-460, 466-469).

It is really a pity that annelid data is not included in the current manuscript, since most standardized toxicity data comes from this group.

We agree that it is unfortunate that we could not include the annelid data. However, this was done to base our conclusions on a robust dataset. The sampling protocols in the LUCAS Survey and the small soil volume sampled do not allow for accurate data on macrofauna. Previous studies have recommended using larger soil volumes to better account for bigger organisms (Königer et al. 2023), and this limitation is now addressed in the supplementary discussion. In our study and across all sites investigated, we only found two different ASVs, both assigned to *Lumbricus polyphemus*, while the majority of identified annelid species were enchytraeids (mesofauna). This confirms that earthworm diversity was not fully captured by the metabarcoding approaches.

Another issue that deserves further consideration is that although earthworms are used in standardised toxicity tests, the model species are not the most representative of the field earthworm communities. For example, the NOEC widely used in pesticide risk assessment is established for *Eisenia fetida*, *E. andrei*, *Aporrectodea caliginosa*, *A. longa*, *A. icterica*, *Lumbricus rubellus*, *L. terrestris* and *Perionyx excavatus*. While some of these species are commonly found in soils, others such as *E. fetida*, *E. adrei* and *P. excavatus* are vermicomposting worms and produced commercially.

Another paper that could be worth relating to is Alsani et al. (2024, <https://doi.org/10.1016/j.soilbio.2024.109459>), since this has authors from the group, and specifically tackle land use intensification in relation to part of the soil biota community.

Many thanks for pointing to this paper. It is now included in the Supplementary discussion.

Pesticide use is often an inseparable part of land use intensification, so they often have to be discussed as a bundle (e.g. as discussed in Rigal et al. 2023, <https://doi.org/10.1073/pnas.2216573120>).

We agree with this comment, and further discussed how pesticides and land use management practices are interlinked: “*In addition, future studies should incorporate detailed land management data, —such as lower pesticide use in crop rotations compared to greenhouse cultivation—and their intertwined effects on soil processes and biota. For instance, excessive mineral fertilisation may lead to phosphorus loss when combined with an organophosphorus pesticide, while antibiotic contamination can worsen with concurrent pesticide application. Additionally, conventional tillage can increase pesticide leaching, while conservation tillage practices —often involving pesticide-treated seeds—may shift fungal communities. Data on such practices could help disentangle land management effects from pesticide impacts on soil biodiversity*” (lines 449-457).

The authors have though done a good job of setting the work into the appropriate context and cite relevant literature.

Abstract

L35-38: Consider caveating with the lack of agricultural management practice information possibly related to pesticide use and soil residues.

Thank you for the suggestion. We have now expanded the discussion section to include these caveats. In the abstract however, the limited space makes it difficult to include this aspect and instead, we had to focus on our main results and policy-related conclusions.

Main manuscript and figures

L115: number of pesticide residues = number of detected pesticides?

Yes, indeed we refer here to the number of detected pesticides. We edited the sentence accordingly, thank you.

L116-118: Are the different ecosystem sites evenly distributed over countries and latitude/longitude? Could this be presented in some way in the methods? I can see that there are figures with the point distribution, but it is hard to see if the different ecosystem sites are well distributed since the maps are very dominated by the annual crop land sites. Consider including supplementary maps that show the site distribution by ecosystem (i.e. one map for each ecosystem type).

Thank you for this suggestion, we have now created different panels for each ecosystem type, which are included in Supplementary Fig. S1.

L134: For the cumulative risk metric, consider including a reference to where this same approach has been used before (i.e. in Franco et al. 2024).

Thank you, done (line 70).

L136: Replace responses with relations.

Thank you, done (line 171).

L144: Why the changed direction (from positive to negative relation) for bacterial N-fixers when using all sites vs only cropland sites?

Our findings suggest that N-fixing bacteria may benefit from pesticide inputs in croplands while there is a negative correlation when all sites are presented. In our dataset, N-fixing bacteria were the most abundant in extensive grasslands (EG; see Supplementary Fig. S7a and boxplot below, with Pairwise Wilcoxon test), where 0.8 pesticide residues were recorded on average per site, while we found more pesticides (5.8 residues on average per site in annual croplands (AC) and 3.3 in permanent croplands (PC)) and a lower relative abundance of N-fixers in croplands. This opposite trend between pesticides and N-fixers could lead to a negative correlation when all sites are compared together, while this is not the case when only considering croplands (i.e., N-fixers abundance increased with increasing pesticide presence there – see figure below). The results below can be included in the supplementary materials if you think they would provide additional value. *While our findings in croplands align with*

previously found stimulated growth of N-fixing bacteria by pesticides (Das et al. 2012, Sun et al. 2021), other studies reported a decline in nitrogen-fixing bacterial genes with increasing concentrations of individual pesticide residues (Walder et al. 2021). Even though we added this sentence to the discussion (lines 422-424), given the limited relative abundance of N-fixers in our dataset (up to 3%, see boxplot below) we are uncertain about its importance for the discussion and further research is needed here.

Figure: Relative abundance of bacterial N-fixers (in %) in the five ecosystem types (AC: annual croplands, PC: permanent croplands, FC: former croplands recently converted to grasslands, EG: extensive grasslands, WL: woodlands). Significant p-values in the Pairwise Wilcoxon test indicate differences in relative abundance between two ecosystem types.

Sun, T., Li, M., Saleem, M., Zhang, X. & Zhang, Q. The fungicide “fluopyram” promotes pepper growth by increasing the abundance of P-solubilizing and N-fixing bacteria. *Ecotoxicology and Environmental Safety* 188, 109947 (2020).

Walder, F. et al. Soil microbiome signatures are associated with pesticide residues in arable landscapes. *Soil Biology and Biochemistry* 174, 108830 (2022).

Das, Amal C.; Nayek, Hemanta; Nongthombam, S. Devi. Effect of pendimethalin and quizalofop on N 2-fixing bacteria in relation to availability of nitrogen in a Typic Haplustept soil of West Bengal, India. *Environmental Monitoring and Assessment*, 2012, vol. 184, p. 1985-1989.

L149-150: This is an important conclusion, though it can be made just by looking at the strong correlation in supplementary figure S6. Consider using your own conclusion to simplify the manuscript.

This conclusion is based on the correlation matrices (Fig. 2, Supplementary Fig. S5), where the Spearman correlation coefficients for the cumulative risk against biodiversity metrics were found to be similar to those observed for the other pesticide metrics (e.g., number of pesticides). In the revised manuscript, we have clarified on which results this conclusion is based (lines 117-120).

L160-162: Since you are relying on composite pesticide metrics here, there appears to be no need, and actually contra productive, to only include the top 20/28 occurring pesticides. Would the result be different if you included all detected pesticides? Based on supplementary figure S7, that would not be the case, so I suggest to just rely on data from all detected pesticides (and in addition possibly reduce the number of predictors based on the strong correlations in supplementary figure S6).

In the original submitted version, we provided correlation analyses for the top 20/28 most occurring pesticides to: (i) confirm that the pesticides detected less frequently did not influence the correlations, and (ii) link the results from the correlation analyses to those derived from the GLMs of these most occurring pesticides. However, in the revised version, we have drawn all figures using 63 pesticides. We could indicate that we also investigated the correlations for the top 20/28 pesticides and found similar outcomes in the caption of the figures if necessary.

Figure 2: Herbicides, Insecticides, Fungicides – are these referring to the number of detected compounds of these groups or something else?

Thank you for pointing out this confusion. We refer here to the number of detected pesticides identified as herbicides, insecticides, and fungicides. We have modified the legend and y-axis labels of Figure 2 to make this clearer.

L180-181: Here is a place where caution is needed in relation to pesticide residues vs other, possibly related, land use intensification metrics that are unmeasured. It is rather pesticide concentration and other related variables.

Many thanks for pointing this out. We also agree that there are limitations in these interpretations, and therefore, we have extensively discussed them in the discussion section, by better acknowledging how the lack of information on land management practices represents an important limitation in our conclusions based on pesticide concentrations (lines 449-457): *“In addition, future studies should incorporate detailed land management data, —such as lower pesticide use in crop rotations compared to greenhouse cultivation—and their intertwined effects on soil processes and biota. For instance, excessive mineral fertilisation may lead to phosphorus loss when combined with an organophosphorus pesticide, while antibiotic contamination can worsen with concurrent pesticide application. Additionally, conventional tillage can increase pesticide leaching, while conservation tillage practices —often involving pesticide-treated seeds—may shift fungal communities. Data on such practices could help disentangle land management effects from pesticide impacts on soil biodiversity”.*

L185-194: Also here there is need for nuance, since it could equally well be unmeasured variables that correlate with pesticide residues. This is highly likely given the relation between pesticide use and other aspects of (agricultural) land use intensification.

We acknowledge that unmeasured variables remain a potential source of unexplained variance in the model, which is inherent in all ecological studies at this scale. However, we have now stressed this point in the revised manuscript (lines 892-894 of the Methods): *“The unexplained variance, represented by the residuals of the model, reflects the potential influence of unmeasured factors.”*), in Supplementary results S3, and extended the discussion in relation to the links between agricultural practices and pesticide residues (lines 449-458 of the discussion).

L201-203: Could the negative shared variance be related to model overfitting?

Negative shared variance arises as a mathematical artifact of how variation is partitioned among sets of predictors. According to Legendre and Legendre (2012), “such values indicate that groups of predictors collectively explain the dependent variable better than the sum of the individual effects of these predictors”. In general, it is recommended to interpret negative values as zero. We have explained this concept in Supplementary results S4.

To avoid overfitting and reduce the model complexity, AIC-based stepwise selection of the predictors was used.

Legendre, P. and Legendre, L. (2012) *Developments in Environmental Modeling*. In: *Numerical Ecology*, 3rd Edition, Elsevier, Amsterdam.

L210-: I’m considering this part inappropriate given the nature of the data with monitoring of pesticide residues rather than experimental + potential for correlation among residue concentrations. It may be more misleading than informative, which is why I suggest to exclude this part where individual pesticide concentrations are included as predictors. Such information could rather be used to point to which pesticides drive potential exposure and risk overall or in particular ecosystems (perhaps this is already done in previous publications?).

Thank you for raising this point. We have now substantially shortened this section and moved most to the Supplement (see Supplementary Results S3). In our view, exploratory studies better reflect the reality of co-occurring pesticide residues in the field and their potential interactions with a broad range of environmental properties. Our analysis indicate that several (distinct) pesticide concentrations can be retained as relevant predictors in our models, and these results can be interpreted as the importance of a given residue concentration under certain environmental conditions and in the presence of one to several other co-occurring residues (illustrated by partial plots).

However, we agree that our results need further sustentation by experimental studies manipulating several pesticides, and now discuss the limitations of current interpretations: *“While our study provides the first exploratory study of the in-field effects of pesticides on soil biodiversity, it also shows some limitations regarding the lack of detailed data on pesticide application rates, timing, and composition of pesticide mixtures applied.”* (lines 436-438) and *“To further establish causal links between pesticide effects and soil biota, experimental studies that manipulate pesticide presence, concentration, and combinations of pesticides are needed to complement observational studies, accounting for direct, indirect and cascade effects.”* (lines 445-447)

Regarding the potential for correlation among residue concentrations, we now clearly indicate that collinearity was checked (see the Methods section lines 843-850), and the correlation matrix between all variables initially considered as predictors is now provided for the GLMs (Supp. Figure S12 and Supp. Figure S4 of Supplementary Data File S3). We also show the higher correlations ($|r| > 0.5$) in the dataset across all ecosystem types (n=373 sites) and in croplands (n=244 sites) (Supplementary Table S10 of Supplementary Data File S2 and Supplementary Table S9 of Supplementary Data File S4).

Pesticide risk on soils was analysed in a previous study by Franco et al. (2024), but it did not include any links to measured soil biodiversity, and our study provides this novel information.

Franco, A., Vieira, D., Clerbaux, L-A., Orgiazzi, A., Labouyrie M., Köninger, J., Silva, V., van Dam, R, Carneseccchi, E., Dorne J-J. CM, Vuaille, J., Vicente, J. L., Jones, A. Evaluation of the ecological risk of pesticides residues from the European LUCAS soil monitoring 2018 survey. Integrated Environmental Assessment and Management (2024).

L211-212: Are the concentrations of individual (or several) pesticides correlated?

There were no strong correlations between individual pesticide concentrations (see Supp. Figure S12 and Supp. Figure S4 of Supplementary Data File S3), except for glyphosate and its metabolite AMPA (Pearson $r = 0.68$ across all ecosystems and $r = 0.64$ for croplands, p -value < 0.05 in both cases). Collinearity between pesticide concentrations was not detected according to the VIF calculations (vif function from car package).

L233: Delete “very”.

Corrected, thank you.

L234: Delete “significantly” – if not significant, there is no relation to mention. Makes me wonder about the significance of the “negatively related” in the following sentence.

Corrected, thank you.

L246: responses = relations

Corrected, thank you.

L507-508: Are these sites isolated from cropland? That would be a requirement for considering these for background contamination and possible benchmarking. I’m curious to know the distance of isolation from cropland and in general on the landscape context land use/cover composition around these sampled points.

Thank you for raising this important question regarding the distance of non-cropland sites to croplands. To assess the isolation of non-cropland sites from croplands, we have analysed the land uses of the surrounding areas using CORINE land cover satellite data from 2018. This analysis revealed that 39 out of 129 non-cropland sites (27 extensive grasslands, 11 former croplands, and 1 woodland) were located within CORINE cropland polygons. Those polygons are a vectorised version of CORINE land cover raster (100 m resolution). This information has been added to the Methods

(lines 646-650) and the discussion sections (lines 428-431), including the potential implications for pesticide exposure at these sites.

L521-523: Add information on what percent of the sites that these numbers represent for each of the ecosystems (e.g. 13% for cropland).

We added these percentages in the text (lines 664-668):

- Annual croplands: 13.33%
- Permanent croplands: 38.24%
- Recently converted grasslands (former croplands): 42.11%
- Extensive grasslands: 53.61%
- Woodlands: 84.62%

L528-529: Systemic vs non systemic is a debated character since it can refer to both the target pest and the crop plant to be protected. What is the reference here?

We acknowledge that the terms "systemic" and "non-systemic" are subject to debate, as they can refer to how the pesticide affects the crop plant or the target pest. We are using the PPDB's definition referring to the target region of pesticides: "systemic pesticides are affecting or distributed throughout the whole body". However, to avoid this confusion but also to keep the focus on the main message, we have now omitted this variable from our analyses.

L534: NOEC for which organisms?

The list of taxonomic groups and species used to establish NOEC values includes annelids (*Eisenia fetida*, *E. andrei*, *Aporrectodea caliginosa*, *A. longa*, *A. icterica*, *Lumbricus rubellus*, *L. terrestris*, *Perionyx excavatus*, *Enchytraeus albidus*), collembolans (*Folsomia candida*, *F. fimetaria*, *Heteromurus nitidus*) and mites (*Hypoaspis aculeifer*). We have added this information in the Methods section (lines 782-687), together with a discussion on how NOEC data are not available for other organisms considered in our studies (lines 688-696 of the Methods) and stress that this is a knowledge gap in current pesticide risk assessment (Supplementary results S1, lines 393-396 of the discussion).

L547-549: What are the methodological challenges here? Is it just few taxonomic units within these groups that is the limitation. It is a pity that annelids can't be included since much of the standard toxicity testing is based on this group and the cumulative risk metric use here should be particular relevant to explore in relation to this group.

Thank you for raising this point, please see our earlier answer about the methodological caveats in the earthworm species selected for laboratory experiments and ecotoxicological risk assessments (the majority focusing on *E. fetida*). (see answer above)

Also, in the Supplementary discussion, we discuss further limitations in assessing soil biodiversity, such as methodological challenges that hamper the assessment of earthworm diversity with metabarcoding methods: "Improved methods (e.g., by considering larger volumes of soil) could capture a wider spectrum of taxonomic groups, including larger and more mobile organisms like annelids." In the Methods, we also mention: "Three eukaryotic groups (rotifers, tardigrades and

annelids) were excluded due to methodological constraints (limited soil volume considered⁶⁹) and lack of reference databases.)” (lines 709-713)

L560: Could the same approach be used for the functional groups? In such case, the questionable use of abundance proxies could be avoided.

Please see our previous response on the advantage of abundance versus richness-based calculations of functional groups.

L562-564: How reliable is this method for inferring abundance for these organism groups? It would be good with a supplementary section on this and possibly a caveat somewhere in the main text.

Many thanks for raising this question, we added the caveats linked to functional annotations inferred on organism taxonomy in the Supplementary discussion.

L573-: This would be an appropriate place to also mention the lack of agricultural management data. Additionally, it would be informative to have a correlation matrix (similar to supplementary figure S6) that also included soil and climatic properties, in addition to the pesticide metrics to understand the interrelationship among all explanatory variables. Finally, was there any systematic difference in soil and climatic properties among ecosystems?

Regarding the lack of the agricultural management practices and the correlation matrix, please see our response above addressing your points (especially the new Supp. Figure S12 and Supp. Figure S4 of Supplementary Data File S3). The sampling points were selected to prevent any systematic bias in the distribution of soil/climate/land cover.

L599-600: Again, these data are unsuitable for exploring the many individual pesticide concentrations as predictors.

By using GLMs and their partial plots, we ensured that we can confidently discuss the relationships between any given pesticide concentration and each of the biodiversity metrics while accounting for all other predictors in the models (i.e., environmental properties or any other pesticide concentration in addition to the pesticide concentration discussed).

L602-604: To support the use of separate models here, ecosystem could be included as a predictive factor with inclusion of the interaction(s) with other predictors. If $p < 0.05$, it would support the use of post hoc testing of differences in relation within ecosystems.

Thank you for this suggestion. By modelling the cropland sites separately, we could account for certain pesticides that exhibit near-zero variance when all ecosystem types are considered. Some of these pesticides show relevant variation patterns in these ecosystems, in particular annual and permanent croplands, deserving to be investigated further.

L607-615: It is evident from supplementary figure S6 that there is high correlation (mostly $r > 0.8$) among number of pesticides, herbicides, fungicides, mode of action, chemical groups, persistence, broad-spectrum, meaning that these variables would mostly explaining the same variance in dependent variables. Thus only one of these should be used in analyses (I suggest number of

pesticides detected for simplicity and comparability with others work) and it should be made clear that it would represent a number of other pesticide related metrics that are impossible to separate. The other metrics (cumulative concentration and risk + number of detected insecticides) are not as correlated and could be considered for inclusion. Another solution to focusing on the most relevant variables would be to use multivariate statistics.

Please see our response addressing this point above (i.e., correlation analyses versus GLMs, the use of two-steps models with feature-selection, and the verification of collinearity).

L615-628: Again, I find it inappropriate to conduct predictive testing for the individual concentrations of pesticides based on the underlying monitoring data. For the compounded pesticide metrics, it defeats their purpose to exclude some pesticides.

These lines referred to how the correlation matrices between soil biodiversity metrics and pesticides metrics (e.g., number of pesticides, numbers of insecticides/fungicides/herbicides, cumulative concentration) were built. The correlation matrices in Supplementary Fig. S7 (now S5) were elaborated for the 63 pesticides detected, 20 most occurring pesticides across all ecosystem types and 28 in croplands to (i) verify that the pesticides detected less frequently did not influence the overall correlations, and (ii) link the correlation analyses to the GLMs using these most occurring pesticides. In the revised manuscript we now focus on the 63 detected pesticides only.

L661-662: Would it be possible to here also model the importance of unmeasured variables? For example by using counterfactual analysis?

Thank you for the suggestion. We agree that counterfactual analysis could provide valuable insights into the importance of unmeasured variables. However, counterfactual methods typically require specific conditions, such as well-defined treatment and control groups, which does not apply to our large-scale ecological dataset. The complexity of multiple interacting factors across diverse ecosystems further complicates the application of this approach.

However, as we acknowledge the importance of unmeasured variables, we have discussed this issue in the revised manuscript and highlighted the need for more experimental laboratory and field studies and provided some recommendations for future studies to fill the remaining knowledge gaps (see Supplementary discussion).

L676: I'm wondering if it would not be more informative to replace the cumulative pesticide concentration here and in related figures with the number of detected pesticides since this metric was so strongly correlated to many other of the explored pesticide metrics?

The variation partitioning is based on the GLMs outputs that included one or more pesticide concentrations and that were retained as relevant predictors along with soil properties, climate and ecosystem. The cumulative concentration and the number of detected pesticides were used as pesticide metrics for the correlation analyses only (Fig. 2, Supplementary Fig. S5) and they did not influence the GLMs.

L692-: This details the underlying datasets of pesticide residues and soil biota DNA metabarcoding, but not the dataset that matches the two for the 373 included sites. So in addition to the underlying dataset, I suggest to also archive the specific pesticide-soil biota dataset used in this manuscript.

L700-: Consider also archiving the R code used for the data analyses done in this manuscript. The referenced code is just for part of the underlying data, if I understand correctly.

Thank you for raising this point. We propose to create a dedicated ESDAC page where both the specific pesticide-soil biota dataset and the R scripts would be available. [Redacted]

Supplementary material

Supplementary figure S1a: Number of pesticide residues = number of detected pesticides

Corrected, thank you.

Supplementary figure S1: Possibly also cumulative pesticide risk on a map, but that may be very similar to the map for number of detected pesticides. Again, consider mapping each ecosystem in a separate map panel since it is hard to see anything but the abundant cropland sites.

A map showing the cumulative pesticide risk is very similar to the map of the number of detected pesticides. However, we agree that the different ecosystem types are overlapping, making it difficult to see. Therefore, in agreement with your suggestion, we have produced separate maps for each ecosystem type (see Supplementary Fig. S1).

Supplementary results S1: It is actually Roundup and not glyphosate that is studied in ref 88 – please correct accordingly.

Thank you for pointing out this error. We have corrected this section.

Supplementary figure S3: Please give units on the axes when appropriate.

The unit “in mg/kg” was added for the cumulative concentration. For the cumulative risk, we specified the calculated ratio in the figure legend.

Supplementary figure S4 and others: N values are missing in many of the figure texts – please include these consistently.

We have added the number of sites (n = 373 or n = 244) to the figures.

Supplementary figure S7: I’m wondering about the inclusion of panel a here, since panel b should be sufficient and they are very similar. Is there a specific reason for including both? And for the identification of the top 20-28 pesticide compounds?

As mentioned in previous responses, we have performed correlation analyses for the top 20/28 most occurring pesticides to (i) verify that the pesticides detected less frequently did not influence the correlations, and (ii) link the correlation analyses to the GLMs that use these most occurring pesticides. In the revised manuscript we only include the analyses based on all 63 pesticides.

Supplementary figure S8: Also here it would probably be more interesting to have number of detected pesticides rather than cumulative concentration since the former related strongly to many of the other pesticide metrics. Why was the cumulative concentration chosen over the other metrics?

The pesticide concentrations are the (different, not lumped) residue concentrations retained in the GLMs as relevant predictors, not the cumulative concentration lumping all pesticides together.

Supplementary table S6-7: Where are the model specifications that these numbers come from?

The results are based on the GLMs used for further analyses (i.e., those used for variable importance calculations, and for variation partitioning). This table is now presented in Table S11 of Supp. Data file S2. For each pesticide concentration retained in the GLMs, we assessed the existence of significant differences in the mean values of the environmental variables selected in the same model between sites in which the pesticide was present versus sites in which it was not detected above LOQ. For this, we used a Kruskal-Wallis test. As explained in a previous comment, if the GLMs correspond to:

Soil property S1 + Soil property S2 + Climate C1 + [Pesticide A] + [Pesticide B] + Ecosystem type

we tested if the average value of S1 differs between sites where Pesticide A is present or not detected above LOQ, then tested the same between sites where Pesticide B is present or “absent”, and then moved on to assessing any difference in S2 in the presence or “absence” of Pesticide A, Pesticide B, then moved on to C1 in a similar way.

Referees' comments:

Referee #1 (Remarks to the Author):

The revised manuscript “Pesticide residues in soils differentially affect soil biodiversity” by Koninger et al evaluated the effects of pesticides on soil microbial and faunal communities across multiple ecosystems in Europe. They report that ~70% sites contained pesticide residues and different soil communities responded differentially to pesticide residues. In revised version they also analysed functional genes involved in carbon, nitrogen and phosphorus cycle which mainly responded positively while soil fungi community declined. Authors concluded that these findings highlight the need for comprehensive assessment of pesticide effects on soil biodiversity for improved regulation.

In my opinion, authors have addressed most technical issues raised by reviewers by including new data from metagenomics, utilising other statistical tools (e.g. Structural equation models) and providing more details where needed. Two key weaknesses remain in my opinion

- a. There is a need for better articulation of scientific needs and socio-economic and regulation policy. Combined with space constraint, the authors have understandably taken a more cautious approach, and rather focused on research requirements for future regulation/ policy impact. However, in my opinion, there is still some room for this to be strengthened and I have highlighted below where and how.
- b. The Discussion section seems disjointed. This is partially due to address multiple issues raised by reviewers. Below I have provided some notes to improve this.

We thank you for your positive feedback on previous revisions brought to the manuscript. In response to your comment on the articulation of scientific needs and regulatory context, we have revised the Introduction to more clearly position our study within current regulatory gaps (e.g., lines 65-66). In the Discussion, we have restructured the text to ensure clearer focus and flow. Specifically, as suggested, we now dedicated distinct, thematically cohesive sections to key issues: regulatory challenges together with policy implications (lines 304-323) and future research needs to address (lines 324-335).

We also moved some extended examples and supporting context to a Supplementary Discussion, allowing us to maintain clarity and narrative focus within the main text while still providing depth for interested readers.

Other comments

1. L44. It should be ‘59% of the Earth’s biodiversity—’

Done, thank you. (line 42)

2. L46. Please provide reference for impact of soil adsorption and absorption impact on pesticide residue persistence.

We have added following reference (line 44): Wauchope, R. D., Yeh, S., Linders, J. B. H. J., Kloskowski, R., Tanaka, K., Rubin, B., ... & Unsworth, J. B. (2002). Pesticide soil sorption parameters: theory, measurement, uses, limitations and reliability. *Pest management science*, 58(5), 419-445

3. L52. In my opinion, it should read- ‘--soil organisms in ecosystem functions including pesticide degradation’.

Done, thank you (line 51).

4. Line 66. This will be good place to include one or two sentence on limitation of current regulations i.e. based on assay of single compounds and on specific organisms.

Thank you, we have extended this point to also include how different field conditions as well as long-term exposure effects are not adequately considered in pesticide authorisation processes (as also pointed by Riedo et al. 2025; lines 65-66).

Riedo, J., Rillig, M. C. & Walder, F. Beyond Dosage: The Need for More Realistic Research Scenarios to Understand Pesticide Impacts on Agricultural Soils. *Journal of Agricultural and Food Chemistry* (2025).

5. L183-185. Impact on N₂ fixing taxa- have authors accounted for presence of legume plants? Presence of legume plants will increase relative abundance of symbiotic N₂ fixer which can potentially bias this dataset.

Thank you for raising this point. To test this suggestion (even though soil DNA extracts are dominated by DNA from free-living organisms rather than strict endosymbionts), we have examined the effect of crop type (legume vs. non-legume) on N-fixing taxa relative abundance (metabarcoding) and N-fixation genes abundance and diversity (metagenomics) (Fig A below). The ANOVA output showed no effect of legumes. However, there was a slightly higher abundance of N-fixing taxa in legume crops. Further analysis revealed that N-fixation genes abundance and diversity increases with humidity, which can be explained by the fact that legumes are typically cultivated in mesic regions (Fig B). This led us to the conclusion that the bias from legumes is likely negligible to be included in our models (GLMs). Also, we have now removed the correlation analysis from the present version of the manuscript, reshaping and relocating them in Supplementary Results S4 following comments from other reviewers. There, we only describe the stronger correlation patterns (Spearman’s coefficient value around or above |0.4|) such that the results on bacterial N fixation are not approached as significant relationships anymore (based on a comment from reviewer 2).

6. L302-313. Could this finding be linked to microbial activity, and pesticide adsorption? For example, lower temperature means lower microbial activities that in turn can reduce rate of pesticide degradation.

We agree and hence, we have added the following sentence to the interpretations in the Supplementary Results S3: “These findings point to lower pesticide degradation, potentially connected to lower microbial activities in colder areas”. In addition, in order to streamline the manuscript’s narrative (as requested by reviewers 4 and 5), all results referring to the relationships between environmental variables and the presence of pesticides in shaping soil biodiversity have been transferred to Supplementary Results S3.

7. L316-317. You have already said this earlier, suggest to delete.

Done, thank you.

8. L330-333. This belongs to results - not in discussion section. Authors should highlight rationale or implication of this finding.

Thank you for raising this point. We have more implicitly linked this finding to the capacity of bacteria to metabolise pesticides, which may potentially give them an advantage in pesticide-rich environments (lines 278-283 of the discussion).

9. L336. Suggest use the term – with reduced diversity of ‘some’ beneficial organisms,--

Done, thank you (line 261 of the discussion).

10. L343- suggest – soil food web and ‘potentially’ ecosystem functioning by—

Thank you for this suggestion. We have reformulated the sentence to link it to controlled experiments (lines 257-258).

11. L428-435. Not sure what does this section add. There are too many – ‘future research is needed’ in the discussion section. This is good that authors should highlight those but that should be based on following in my opinion. How does the present work advance the discipline and how does this open up new area of research (future research) and what other evidence are needed for societal impact- in this case regulation policy. So, my suggestion to authors will be to consolidate future research requirements in one paragraph (may be paragraph just before conclusions section).

Many thanks for this valuable suggestion. We added a paragraph listing all the future research needs as well as how our work can help in advancing the discipline (lines 324-335 of the discussion).

12. L445-458. Suggest to reducing this section to half and bring about 2- 3 sentence on requirements for improving specific regulations.

We have combined this section with the one dealing with future research and improving specific regulations (lines 324-335 of the discussion).

Overall, I believe that the revision has significantly improved the manuscript and, in my opinion, if articulated appropriately, it can have significant impacts on both the scientific discipline and relevant policies.

Many thanks for your feedback and suggestions. We integrated those into our revised manuscript to improve the narrative towards a more impactful message.

Referee #1 (Remarks on code availability):

The analyses seem straight forward to me and believe any one can repeat those analyses.

Regarding code, I do not consider myself an expert on this and hopefully other reviewers can provide more informed advice.

Referee #2 (Remarks to the Author):

I appreciate the authors' efforts to address the concerns pointed out in the first-round review, particularly in clarifying the novelty of study, expanding methodological details, and strengthening the discussion. However, I still suggest rejecting this manuscript because some of the explanations are not reasonable, and does not resolve the fundamental science questions. Therefore, the new version does not reach the quality standard of Nature and requires more-than-major revisions to do so.

1. "Large scale"

The authors repeatedly emphasize the "large scale" of their study, considering multiple pesticides, species, countries, and ecosystems. However, I believe this approach, as presented, is not innovative enough to justify publication in Nature. For instance, while the authors analyze the responses of multiple species to pesticides, do the conclusions differ from those of studies focusing on a single species? Have any new scientific insights been gained? Conclusions such as "Soil biodiversity (taxonomic and functional) responded differently to pesticides depending on organism type" and "the effects of soil pesticide residues may extend beyond their target organisms" are just expected findings on the basis of prior single-species studies.

We understand your concerns. However, the scale covered in this study (Europe), the number of pesticides analyses, the number of ecosystem types, and the wide range of soil organisms studied has never been done before. Additionally, no previous work has involved such a variety of functional gene groups' responses (for C, N, P cycling) to pesticides at the continental scale either. Most importantly, our results demonstrate for the first time the importance of pesticides compared to other factors (soil properties, ecosystem type and climate) in influencing soil communities. That is a major step forward, and such analysis can only be done with the large dataset included here.

In order to stress these facts, we have reformulated the conclusions to better emphasise the strength of our findings from early on (please see the abstract).

Moreover, the authors claim their study is unprecedented in scale and attempt to consider multiple species, yet they overlook annelids, despite offering some justification for this omission. Additionally, soil properties, climate, and other factors can influence the impact of pesticides on soil organisms. Can the European

sampling sites truly represent global climate types (e.g., as classified by the Köppen Climate classification) or soil properties (e.g., as described by SoilGrids)?

We did not claim that the present study exhaustively covered all environmental conditions nor represented global climate and soil property types (please let us know where this was stated otherwise). However, we do cover a wide range of sites (373 sites in 26 countries - see Figure 2) and our sampling sites represent major ecosystem types and climate in Europe. The Köppen Climate classification, although valuable and very informative, represents a classification by vegetation type. Studies focusing on soil biodiversity show that they do not follow the same patterns as above-ground organisms (Cameron et al. 2019)

Annelids were not overlooked. However, the methodology used in this study does not allow for a representative sample of these organisms as we discussed before (Köninger et al. 2023) and trying to interpret these results would have led to too much speculation in our point of view. We have now specifically addressed this in the discussion, stating that soil biodiversity assessment is not exhaustive and missing several groups, among which annelids (lines 290-294), linking this gap to methodological constraints (e.g., soil volume for annelids or limitations of reference databases for tardigrades and rotifers; see Methods lines 519-521). Still, this study analyses many more biodiversity metrics (a wide range of taxonomic groups and functional gene responses)” as previous studies and at a much larger scale.

Cameron, Erin K., et al. Global mismatches in aboveground and belowground biodiversity. *Conservation Biology*, 2019, vol. 33, no 5, p. 1187-1192.

Köninger, Julia, et al. Ecosystem type drives soil eukaryotic diversity and composition in Europe. *Global change biology*, 2023, vol. 29, no 19, p. 5706-5719.

2. Pesticide

The authors acknowledged that “pesticide occurrence” is misleading and replaced it with “pesticide metrics.” However, this change in terminology does not resolve the fundamental issue: the manuscript still lacks a clear and rigorous definition of how pesticide presence is quantified and interpreted. Simply renaming the term does not clarify the methodology or address my concern that lumping various pesticide-related parameters into a single term obscures their distinct ecological implications. The authors should explicitly define how each pesticide metric is measured, normalized, and integrated into their analysis.

To address your concerns, we have produced a new correlation matrix only including aggregated metrics for pesticide occurrence and risk commonly used in the literature. Pesticide occurrence metrics included (i) the total number of pesticide residues detected per sites, the number of (ii) herbicides, (iii) fungicides and (iv) insecticides. These metrics were compared to the cumulative risk, and additional risk indicators tailored per pesticide type (with one risk indicator for herbicides, one for fungicides, one for insecticides). The calculations and use of the aggregated pesticide metrics are further detailed in the Methods (lines 716-738 of the

manuscript). We present the correlations as a simplistic overview of trends between soil biodiversity and pesticides and present the limitation of using aggregated metrics. Following a request from reviewers 4/5 to streamline the manuscript's content, we have transferred the correlations analyses to the supplementary material as complementary analyses (please see Supplementary Results S4).

Additionally, their response does not clarify how increased detection (e.g., owing to more sensitive analytical techniques) is distinguished from actual changes in pesticide presence in the environment.

We apologize for not addressing this point correctly. In the initial submission, we used the term "increased pesticide occurrence" to describe the consistent trends observed in the correlation analyses. For example, bacterial richness was positively correlated with multiple pesticide metrics (e.g., the number of residues detected per site, the cumulative concentration, and the number of fungicides). Therefore, rather than listing each association separately, we summarized these findings using the broader term "pesticide occurrence". After reading your comment, we now understand your point and we agree we should avoid using a broad common term for pesticide metrics representing different measurements. In the Supplementary Results S4, we have now reduced the number of metrics to describe pesticide occurrence and risk.

We would also like to clarify that all samples were analysed using a uniform analytical protocol, with standardized extraction, instrumentation, and detection procedures applied across all sites. Our limits of quantification (LOQs) fall within the range commonly reported in the literature; therefore, variations in the number of detected pesticides reflect true differences in environmental presence rather than inconsistencies in detection methodology. It was not our intention to state that pesticide detection in the environment increased (this we cannot address with this study and that was also not our goal).

The authors argue that the cumulative risk is based on multiple taxa (annelids, collembolans, mites) rather than just arthropods. However, this still does not justify the application of NOEC-derived risk assessments to soil microbiota, which differ significantly in sensitivity and ecological function. The fact that NOEC is derived from a limited number of model organisms makes its extrapolation to the broader soil community problematic. Simply acknowledging this limitation in the Methods and Discussion does not resolve the core issue: the risk metric does not adequately capture microbial responses to pesticides. This methodological weakness undermines the reliability of the ecological risk assessments presented.

The cumulative risk has only been included in correlation analyses (old matrix Fig. 2), which are inherently correlative and not intended to imply causation. While we acknowledge the limitations of using NOEC values established for a limited number of model organisms (now in Supplementary Results S4 and lines 305-310 of the discussion), the NOEC remains one of the most established metrics in

ecotoxicological risk assessment, and we believe it is important to report it. Presenting how microbial and other biodiversity variables correlate with cumulative risk and other risk indicators – despite their clear limitations – can still be informative for readers (in particular ecotoxicologists). However, to address this issue more cautiously, we have relocated the risk-related figures to the supplementary material (Supplementary Results S4), where we deepen the discussion on their limitations. This allows interested readers to access the information while supporting our broader argument that risk assessment frameworks should be expanded to better include (microbial) community responses and their functions (lines 305-313 of the Discussion).

The authors state that they initially tested risk quotient models (pesticide concentration/NOEC_{min}) but did not include them owing to distortions caused by NOEC limitations. However, this does not address my concern: the impact of different pesticide active ingredients varies widely, and merely considering pesticide concentrations without accounting for specific toxicities is insufficient. The manuscript lacks an analysis of how different active ingredients contribute to the observed biodiversity changes. Without this, the study provides an oversimplified view of pesticide effects. A more robust approach would involve grouping pesticides on the basis of their mode of action or toxicity profiles to different soil biota rather than treating them as a uniform risk factor.

Our analysis is focused on exploring patterns between pesticide residue concentrations in soils and soil biodiversity at a certain point in time. The goal of our study is not to provide a risk assessment of pesticides but help to identify shortcomings in current regulatory framework using pesticide concentration in NOEC-based risk assessments. The use of GLMs including distinct pesticide concentrations next to environment (as already presented in previous versions of the manuscript) provides how the concentrations of different active ingredients contribute to the observed biodiversity changes.

Notably, mode of action is provided for target organisms, and cannot be applied to non-target groups, which are the primary focus of our study. Toxicity profiles are relevant to non-target groups, but such data are often inconsistently reported (and do not embrace all the diversity of our groups of interest), making it impossible to run standardised analyses. We agree that this is a major limitation in the field and a key area where further harmonized data collection would greatly enhance ecological risk assessments. This is now addressed in the Supplementary Results S4 and lines 308-311 of the discussion.

We agree that incorporating pesticide properties to explain observed effects is a constructive approach. Hence, we further analysed this based on your suggestion. To this end, we hypothesised that the pesticide effects on particular components and properties of soil biota can be explained by pesticide properties. To test this hypothesis, we related pesticides to each other based on:

1. Empirical and regulatory properties that may reflect potential hazards, including regulation status (approved or not), target group (herbicides, fungicides, insecticides), effect spectrum (broad or narrow), access type (systemic or non-systemic), soil mobility (based on Kf, Kfoc, 1/n), and persistence (DT50-field).
2. Structural similarities and physicochemical properties, using the ChemMine tool and 32 JOELib descriptors.
3. Using the GLM results from our study, we created an effect profile for each pesticide, indicating the type of effect (negative, positive, or none) on soil biota properties described by the 63 variables (please see the “pesticide_molecular_properties” table as additional files for review); central to our analysis. We then generated a similarity matrix based on these effect profiles.

Based on these data we performed Procrustes analysis (please see “Data for Procrustes” table as additional files for review). While it showed significant links between molecular and empirically measured properties ($R^2 = 0.60$, $p = 0.001$), both of these were not significantly related to effect on the studied parameters: $R^2 = 0.62$, $p = 0.17$ for the comparison of molecular similarity vs effect similarity; $R^2 = 0.35$, $p = 0.97$ for empirical properties vs effect similarity (please see the “Figure_Procrustes” in the additional files for review).

We feel that involving these additional pesticide properties gave no information gain. In an attempt to focus the narrative on fewer analyses/information in the main manuscript and associated supplements, as requested by other reviewers, we propose to leave this additional analysis out. However, we would be happy to include it in the supplements upon request, should you or the Editor consider it necessary.

3. Soil biodiversity

The authors acknowledge that certain groups are excluded owing to methodological constraints but do not provide a convincing justification for omitting them entirely. Other studies have successfully incorporated earthworm communities via DNA metabarcoding or morphological approaches¹¹.

The additional analysis of annelids is appreciated, but the authors ultimately decided to exclude them on the basis of low dataset quality. This exclusion, rather than improving the study, highlights a major limitation: the study still lacks a true representation of soil macrofauna, which weakens its conclusions about soil biodiversity.

The previous rebuttal letter and manuscript already approached the limitations of annelid assessment with LUCAS sampling design, owing to a much lower soil volume than the one used in Pansu et al. (2015). The latter involved 10 soil cores per subplot of 0.5m².

We agree that this methodological constraint prevent us from having a robust dataset on soil macrofauna (e.g. 40g of soil were used by Lilja et al. (2023) and 80g

in Cuartero et al. (2025)), but unlike those which targeted annelids specifically, we targeted a wider community including archaea, bacteria, fungi, protists, nematodes and arthropods.

Therefore, to better inform the reader on the scope of organisms covered in our analyses, we have listed the major groups considered in this study as well as the groups not included (lines 290-294 of the discussion), referring to methodological constraints (lines 519-520 of the Methods) and flagging it for future research (line 334 of the discussion).

Cuartero, J., et al. Earthworm and enchytraeid indicator taxa of different land-use types identified using soil DNA metabarcoding, *Applied Soil Ecology* (2025)

Lilja, M. A. et al. Comparing earthworm biodiversity estimated by DNA metabarcoding and morphology-based approaches. *Applied Soil Ecology* 185, 104798 (2023).

Pansu, J. et al. Landscape-scale distribution patterns of earthworms inferred from soil DNA. *Soil Biology and Biochemistry* 83, 100-105 (2015).

The authors now recognize the limitations of assigning functions on the basis solely of taxonomic classification, yet their response does not sufficiently address my original concern. Inferring microbial functional roles on the basis solely of genus-level taxonomy is problematic because functional potential can vary significantly within a genus. Without direct evidence from functional gene profiling or metabolic analyses, these assignments remain speculative.

We are aware of functional trait variability and specifically focused on a limited number of most well-studied traits. Moreover, all approaches to functional annotation, including gene profiling and metabolic analyses, have inherent limitations because metabolic activity is highly context-dependent. This is especially true for complex traits like lifestyle (e.g., pathogenic, symbiotic, or endobiotic), which are often performed by the same organism and depend more on the host's species, genotype, and immune status than on the microbe itself.

Furthermore, databases such as FAPROTAX and FungalTraits are based on expert-curated observations from multiple studies, and their functional annotations are more relevant to ecological contexts than those inferred solely from gene profiling. This approach is widely accepted, as evidenced by over 2800 published research papers employing FAPROTAX and FungalTraits for functional inference.

Therefore, we believe that grouping organisms into functional categories is supported by the literature (e.g., Fierer et al. 2017; Ferris et al. 2001; Tedersoo et al. 2014; Delgado-Baquerizo et al. 2016; Oliverio et al. 2020), adding important information to our analyses. Also, functional groups have been used in recent articles (see van den Hoogen et al. 2019, Liu et al. 2022, Labouyrie et al. 2023).

However, we are also aware of their limitations and these are now further addressed in the Supplementary Discussion.

Labouyrie, M., et al. Patterns in soil microbial diversity across Europe. *Nature Communications*, 2023, vol. 14, no 1, p. 3311.

Liu, S., et al. Phylotype diversity within soil fungal functional groups drives ecosystem stability. *Nature Ecology & Evolution*, 2022, vol. 6, no 7, p. 900-909.

van den Hoogen, J., et al. Soil nematode abundance and functional group composition at a global scale. *Nature*, 2019, vol. 572, no 7768, p. 194-198.

Although the authors mention incorporating metagenomic insights, they do not clarify whether these were used to redefine functional groups or simply to provide additional results. If functional annotations are still largely based on taxonomy rather than gene function, the issue remains unresolved.

Metagenomic data were intended to provide a more detailed view of the community's functional response to pesticides at the gene level. Effects on genes involved in three different cycles (carbon, nitrogen and phosphorus) were assessed. These were analysed separately from the metabarcoding data. Regarding the use of taxonomic functional groups, please see our responses in the comment above.

The authors state that they have expanded their analyses to include 48 functional gene groups related to biogeochemical cycles, but there are still major concerns: The selection of functional genes appears arbitrary, and it is unclear how these gene groups were defined or validated. Without a clear methodological explanation, the robustness of these functional assignments is questionable.

The approaches for gene identification in metagenomes are described in the Methods (see lines 552-569). They are state-of-the-art yet conceptually standard (Bahram et al., 2018 *Science*; Dulya et al., 2024 *Environment International*).

The choice of genes of interest is non-arbitrary as they inform on the most important biogeochemical cycles and encode the soil biota functions critical for such ecosystem services as greenhouse gas regulation, nutrient and carbon retention. However, we acknowledge that we did not provide justification for this choice, and it has been corrected (see lines 572-582 in the Methods).

Bahram, M., et al. (2018). Structure and function of the global topsoil microbiome. *Nature*, 560(7717), 233–237. <https://doi.org/10.1038/s41586-018-0386-6>

Dulya, O., et al. (2024). A trait-based ecological perspective on the soil microbial antibiotic-related genetic machinery, *Environment International*, Volume 190, 108917, ISSN 0160-4120, <https://doi.org/10.1016/j.envint.2024.108917>

4. Analysis

The new correlation analyses and GLMs do not fundamentally resolve the problem of pesticide impact assessment, as functional group definitions remain vague. The relationship between pesticides and functional diversity needs more rigorous exploration, potentially through direct metagenomic or transcriptomic analysis rather than inferred functions.

The conclusions of studies on the relationship between pesticides and microbial diversity contradict widely accepted perspectives. Specifically, the analysis indicated that bacterial richness is positively correlated with most pesticides, which deviates from mainstream findings. While the authors propose a reasonable explanation—that pesticide exposure reduces fungal diversity, alleviates competition and promotes pesticide-degrading bacterial communities—this hypothesis lacks robust data support. The current discussion is largely speculative and lacks concrete statistical or experimental validation. The authors are encouraged to provide more detailed supporting data.

We agree that direct metagenomics offers more precise insight into the relationship between pesticides and functional diversity; this is why we used such data in the previously revised manuscript already. Regarding the functional group analysis, we acknowledge the limitations of taxonomic-based annotations. However, as noted above, they are grounded in established ecological trait databases and provide functional insights beyond community composition alone.

A substantial body of experimental research has shown that pesticide applications can confer a competitive advantage to those bacteria capable of degrading or tolerating these compounds (e.g., metabarcoding-based work Ni et al., 2025) due to the release from predator pressure. These alterations in trophic community structure, have been comprehensively reviewed in the classical work by Staley et al., 2015. Also, a recent study by Romero et al. 2025, demonstrated that fungal richness is suppressed by pesticides while bacterial richness is not affected and their functional role increases. Note, while studies indeed do report negative pesticide effects on bacteria (e.g., the recent meta-analysis by Wan et al., 2025), these studies often focus on physiological responses in model strains, such as *Bacillus*.

In our study, we observed reduced fungal diversity associated with glyphosate concentrations, and increased diversity of bacterial genes involved in phosphonate degradation with rising AMPA levels. However, we acknowledge that our study is not a controlled food web experiment and cannot confirm competition mechanisms. We now present the proposed mechanisms more cautiously in the discussion (lines 273-283) and linked them to the supporting studies of Ni et al. and Staley et al. (line 277).

Ni et al., 2025. Increasing pesticide diversity impairs soil microbial functions.

Proceedings of the National Academy of Sciences 122, e2419917122

<https://doi.org/10.1073/pnas.2419917122>

Romero, F., Jiao, S., & van der Heijden, M. G. (2025). Impact of microbial diversity and pesticide application on plant growth, litter decomposition and carbon substrate use. *Soil Biology and Biochemistry*, 109866.

Staley et al., 2015. A synthesis of the effects of pesticides on microbial persistence in aquatic ecosystems. *Critical reviews in toxicology* 45, 813-836
<https://doi.org/10.3109/10408444.2015.1065471>

Wan et al., 2025. Pesticides have negative effects on non-target organisms. *Nature Communications* 16, 1360 (2025). <https://doi.org/10.1038/s41467-025-56732-x>

5. Line-by-line feedback:

Lines 192-221: The authors have replaced sales statistics with pesticide residue concentration data from Vieira et al. (2023), which is a more relevant dataset. However, the manuscript does not clarify whether measured concentrations or modeled estimates were used for analysis. Furthermore, pesticide concentrations alone do not reflect actual exposure levels to soil biota—other factors such as bioavailability, degradation kinetics, and interactions with soil organic matter should be considered.

Pesticide concentrations used in this study were obtained by measurements of soil samples collected within the framework of the LUCAS module named LUCAS Soil Pesticides. To clarify this point, we have added this information in the first paragraph of the Methods (lines 468-471).

While the measurement methods used in our study are well established for pesticide residues and widely used in soil biodiversity studies, we agree that bioavailability data would provide additional insights into actual exposure levels. However, bioavailability is highly organism-specific, and given the wide diversity of organism groups we examined, it is not feasible to incorporate a single, standardised measure of bioavailability across all of them. We also included soil adsorption coefficients (K_f , K_{foc}) in additional analyses (Procrustes analyses; please see our response in the current response letter about the link between pesticide properties and revealed effects on biota). Subsequently, we added this lack of data on pesticide bioavailability, degradation kinetics and interaction with soil organic matter as limitation of the current study (please see Supplementary Discussion). There, we also added how pesticide persistence does not reflect soil biota exposure to pesticides.

Lines 76-83: The authors cite their presence across ecosystem types, but they do not justify why these specific groups were prioritized over others.

Their relevance as biological indicators is cited from Karpouzas et al. (2022), but the authors do not provide evidence that these functional groups respond strongly to

pesticide exposure. The reliance on taxonomy-based functional assignments (rather than metagenomic/metatranscriptomic evidence) is problematic, as it does not accurately reflect functional capabilities.

Functional groups were selected based on their primary roles in soil ecosystem processes that are critical for soil functioning, such as plant productivity, nitrogen loss and retention, and organic carbon release (Ferris et al. 2001, Tedersoo et al. 2014, Delgado-Baquerizo et al. 2016, Oliverio et al. 2020). The text in the Methods has been further extended to illustrate the ecological roles of these groups (lines 538-546). Please see our previous response regarding the reliance of functional assignments based on taxonomy.

Delgado-Baquerizo, M. et al. Microbial diversity drives multifunctionality in terrestrial ecosystems. *Nature communications* 7, 10541 (2016).

Ferris, H., Bongers, T. & de Goede, R. G. A framework for soil food web diagnostics: extension of the nematode faunal analysis concept. *Applied soil ecology* 18, 13-29 (2001).

Oliverio, A. M. et al. The global-scale distributions of soil protists and their contributions to belowground systems. *Science advances* 6, eaax8787 (2020).

Tedersoo, L. et al. Global diversity and geography of soil fungi. *science* 346, 1256688 (2014).

Line 88: The authors have removed pesticide persistence from the analysis, which is appropriate, as persistence alone does not equate to biological effects. However, the manuscript should still discuss why persistence is not a reliable indicator of pesticide impact on biodiversity—for example, highly persistent pesticides may have low bioavailability, and rapidly degrading pesticides can still exert significant effects on microbial communities.

This issue is now discussed in the Supplementary Discussion.

Line 139: The Pesticide Properties Database categorization only applies to target organisms (weeds, fungi, insects), not to nontarget soil biota (archaea, bacteria, protists, etc.). The authors admit that the pesticides analyzed do not directly target archaea or bacteria, yet they still assume broad-spectrum effects on microbial communities. The cited studies (Ma et al. 2021; Puglisi et al. 2012) suggest potential indirect effects of fungicides on bacterial diversity, but this does not justify treating these pesticides as broad-spectrum agents for all soil biotas. If the authors want to claim broad-spectrum effects, they should provide quantitative toxicity data for archaea, bacteria, and fungi, rather than assuming indirect effects.

Thank you for raising this point. The broad spectrum has been removed from the analyses (please see correlation analyses transferred to Supplementary Results S4),

also in an attempt to focus the narrative on fewer analyses/information in the main manuscript (as asked by reviewers 4/5).

Lines 152-155: The authors acknowledge that summing pesticide concentrations leads to potential confounding but still argue that it is a useful approach for quantifying overall pesticide effects on biodiversity. However, this method introduces serious statistical biases: 1) Nonadditive toxicity: Different pesticides have distinct mechanisms of action and toxicity thresholds. Summing concentrations ignore potential synergistic or antagonistic effects and assume a linear dose–response relationship, which is biologically unrealistic. 2) Varying environmental persistence: Pesticides degrade at different rates, and their bioavailability depends on soil properties. Aggregating concentrations does not account for differential persistence and degradation kinetics, leading to misleading conclusions.

The authors claim to have conducted generalized linear models (GLMs) for individual pesticides, but they do not present a direct comparison between the aggregated and individual models. Without such a comparison, the aggregated pesticide approach remains fundamentally flawed.

We agree on the shortcomings derived from summing all pesticide concentrations per site to obtain an overall cumulative concentration because this would assume a linear-dose response, ignoring the synergistic or antagonistic and the non-additive effects of pesticides. Accordingly, this aggregated metric has been removed from the new Supplementary Results S4. In addition, we have further acknowledged that information on pesticide degradation kinetics and bioavailability would provide more insights into soil biota exposure to pesticides (please see Supplementary Discussion).

Importantly, we would like to stress that we did not use any aggregated pesticide metrics in the GLMs (avoiding the claimed linear dose-response assumptions). The aggregated metrics were only used for correlation analysis (Spearman correlation) without considering other factors such as soil properties and climate. As a result, a direct comparison between aggregated correlation analysis and GLM analysis is not relevant.

Lines 140-155: the authors report that all Spearman correlation coefficients in Figure 2 are below 0.4, which is considered "low" or "very weak" based on standard statistical guidelines. Despite this, they argue that statistical significance (p values) is more important than the strength of the correlation. This is a fundamental misinterpretation of statistical significance. A statistically significant result ($p < 0.05$) merely indicates that the observed correlation is unlikely to be due to chance; it does not imply a strong or biologically meaningful relationship. The effect size (correlation coefficient $|r|$) determines the biological relevance of the relationship. Correlations below 0.3 suggest a negligible association between pesticides and biodiversity metrics. The authors attributed the weak correlations to high spatial and temporal

variability in the field data. While variability is expected, the lack of strong correlations suggests that pesticides are not the main driver of soil biodiversity, which contradicts the manuscript's primary claim.

We thank you for your inputs on the correlation analyses. The new correlation matrix (Supplementary Fig. S11) displays the correlations greater than or equal to |0.3|. Associated results describe the stronger correlations (around or above |0.4|) in Supplementary Results S4.

The claim regarding pesticides being the main driver of soil biodiversity has already been removed from the revised version (January 2025), and was initially based on a different analysis, i.e. the variation partitioning performed on the models (GLMs) presented in previous Fig. 3 (Fig. 4 in the current version). This analysis showed that a big share of the explained variance in certain soil biodiversity metrics was attributed to pesticide concentrations. We have reported this observation in lines 251-252 of the discussion, while acknowledging that pesticides are the second major driver of soil biodiversity patterns after soil properties. Consequently, it is important to note that our initial claim and its revised statement were not based on correlation analyses and their coefficient values.

To justify the weak correlations, the authors conduct a subset analysis in Italy, reporting higher correlations (ranging from -0.65--0.72). However, this analysis is flawed for several reasons: 1) small sample size bias: the Italy dataset includes only 18 sites, making it highly susceptible to random variation and outliers. High correlations in small samples do not generalize to the entire dataset. 2) selection bias: The authors do not explain how this subset was selected. If the region was chosen post hoc because it shows stronger correlations, this introduces confirmation bias. 3) Ignoring confounding factors: The greater correlations in Italy could be due to local environmental or land-use factors rather than pesticide effects. Without controlling for these factors, the results are inconclusive.

A more appropriate statistical approach would involve multivariate models with interaction terms rather than selectively reporting high correlations from small subsets.

The previous rebuttal letter explained that this subset of sites has been selected to perform analyses on a more homogeneous set of sites, reducing the high spatial variability in landscapes, soil types and biogeographical regions that could lead to weaker correlations. These sites were geographically localised and encompassed a limited range of soil types (calcisol and cambisol; based on the Soil Atlas of Europe). All limitations (sample size bias, confounding factors) raised here were already acknowledged and discussed by us in a previous letter.

Regarding the statistical approach, the aim of the aggregated pesticide metrics used in the correlation analyses is to give a simplified view of the associations between, e.g., pesticide occurrence and soil biodiversity. Such a correlative approach is

indeed inherently not causative nor exhaustive, as it fails to account for other factors. We now highlight how these correlations can only be interpreted as indicative trends in Supplementary Results S4. We feel these analyses provide added value as explained here, but we can also remove this if needed as this is not central to our observations and conclusions, we agree.

References:

1. Pansu, J. et al. Landscape-scale distribution patterns of earthworms inferred from soil DNA. *Soil Biology and Biochemistry* 83, 100-105 (2015).
2. Kotték, M., J. Grieser, C. Beck, B. Rudolf, and F. Rubel, 2006: World Map of the Köppen-Geiger climate classification updated. *Meteorol. Z.*, 15, 259-263 (2006).

Referee #4 (Remarks to the Author):

We have reviewed the manuscript with special emphasis to the comments of Ref 3 and the rebuttal. In general, we agree with the conclusion of Ref 3 that 'this seems like an interesting, timely and potentially significant study that could substantially moves along our understanding of how real-world pesticide exposure via soil might be affecting the soil biota community in Europe'. The paper shows how widespread agricultural pesticide residues persist in soils and its analyses suggest that soil microbial communities are strongly modified in response. From a cautionary principle these results are of great importance for policy decisions and regulations, although more definite proofs will require further work (as detailed below).

We also agree with Referee #3 that the manuscript presents an overly complex and apparently incoherent narrative, largely due to the sheer number of analyses and datasets included, a problem that only has been partly corrected in the rebuttal and the new version of the manuscript. A few further changes, and some deliberate choices, are advisable to improve the transparency and clarity of the paper.

(1) Bring overview and structure in the many analyses for the purpose of answering the main question of the paper – evaluating the impacts of pesticides on soil biodiversity

The two paragraphs introducing the approach and methods (ll. 67-91) only partly solve the problem of bringing coherence in the many analyses done. The authors have constructed a Methodology diagram which is now hidden as Fig S15 in the Supplementary Material. In our view, this diagram (in modified form) is needed up front to bring clarity and justify the choices made. The left part of the diagram portraits very well the complexity of both pesticides and soil biodiversity. The

diagram can line out the approach of tackling this complexity, dealing with colinearities (middle part), resulting in downstream analyses addressing key aspects of the main question of the paper (right part). Here analyses on the full dataset vs. croplands only should be clearly separated. Hence, a conceptual rather than methodological diagram (although methods can be summarized in the caption) will explain the hierarchy of the analyses, and how and why subsequent steps contribute to addressing the main objective of the paper. It also presents a 'roadmap' as a guidance for the subsequent results to come. A place for the diagram in the main text would be very helpful.

We recommend that the authors use this diagram to explain their deliberate choices about which analyses are essential to the central narrative and which could be relegated to supplementary materials. For example, the title emphasizes the effects of pesticide residues on soil biodiversity, yet the inclusion of functional gene analyses is not well justified; it simply pops up in the manuscript.

Many thanks for this suggestion. We have followed your recommendations and focused the narrative on fewer deliberate choices. Consequently, the manuscript content has been streamlined to base all results on our GLMs (variable importance Fig. 3, variation partitioning Fig. 4). This places cropland analyses up front, and the rationale for focusing on these analyses is now detailed at the end of the introduction. The main conclusions from these analyses are compared with those including both croplands and other ecosystem types in the last paragraph of the results section.

Additionally, in the revised manuscript, we describe the analyses at the functional gene level earlier than in the previous version: the title has been edited to show that the study covers both taxonomic and functional levels, and the abstract introduces the concept of functional genes involved in C, N, P cycling earlier on (lines 29, 32, 34).

The correlation analyses formerly presented in Fig. 2 were refined and moved to the supplementary material as complementary analyses (Supplementary Results S4). All outputs using NOEC and derived risk have been removed from the supplementary figures and were regrouped in Supplementary Results S4. There, some risk indicators were correlated to soil biodiversity and were compared to other aggregated metrics for pesticide occurrence in the correlation analyses. The Results paragraph describing the differences in environmental variables in presence/absence of pesticides shaping soil biodiversity have also been moved to complementary analyses (Supplementary Results S3). Results for Shannon diversity have been moved to the Supplements, for example Supplementary Fig. S8.

Following your recommendation, we have converted the previous Supplementary Fig. S15 into a conceptual diagram (Fig. 1). As the analyses are performed for croplands-only, and then repeated for croplands and other ecosystems together, we have included these two approaches in the diagram with the aim of assessing the spill-over effects and broader ecological impacts of pesticides. Complementary

analyses (Kruskal-Wallis, correlation analyses with pesticide risk/NOEC and occurrence) are listed on the right of this conceptual diagram, with their respective output.

(2) Highly correlated pesticide metrics

The high correlation between variables, acknowledged in the rebuttal, should be addressed in the main text or in the caption of Fig. 2. Although the authors note that Figure 2 is intended to illustrate general positive correlations and is not used in GLMs, we still find it problematic. Including all highly correlated variables risks overstating the implications. We understand the choice of showing the metrics as they are often used in scientific studies and policy making. However, it should be made clear, early in the caption of Fig 2, that for columns with a similar colour, we are essentially looking at one and the same effect, referring to the relevant Suppl Fig that gives the correlations between the variables.

Otherwise, the solution for handling the correlations between the metrics and the collinearities are appropriate, with a two-step GLM where the first step selects the pesticide concentrations to be included.

Many thanks for raising this point. The following sentence has been added to the caption of Supplementary Fig. S11: "Columns with similar colours indicate the same direction of correlation between pesticide metrics and soil biodiversity, reflecting underlying linear relationships among the pesticide metrics (see Supplementary Fig. S12)".

(3) Interpreting results for croplands in relation to agricultural intensification

Pesticide use is typically embedded within broader land management regimes, often co-occurring with practices like excessive use of manure and chemical fertilizer, which also strongly affect soil biota. Although the authors attempted to include some management-related variables, these do not capture long-term land-use intensity. The Discussion (ll.449-458) states the problem as focal points for future studies, not as consequential for the interpretation of the current results. This is an important omission. It should be more clearly acknowledged that the observed patterns represent early-warning signals of effects of pesticides rather than evidence for direct causation. Works such as Tsiafouli et al. (2015) and de Vries et al. (2013) have shown how land-use intensity affects soil biodiversity, and with the seminal dataset that the authors present, the question remains, are effects of pesticides in fact caused by agricultural intensification or, vice versa, are pesticides a causal factor behind land-use intensification effects on soil biodiversity.

Thank you for raising this important point regarding the interpretation of pesticide effects in the context of broader agricultural intensification. We fully agree that pesticide application is rarely an isolated practice and often co-occurs with other intensification measures such as excessive fertilizer use or tillage, all of which can

influence soil biodiversity. As you pointed out, this co-variation makes it difficult to isolate the specific contribution of pesticides to observed changes in soil biota, especially when using observational, field-based data.

In the revised manuscript, we have made a clearer statement (lines 301-303 of the Discussion) that our results should be interpreted as early-warning signals of potential pesticide impacts, rather than definitive evidence of direct causation. However, we would like to stress that this does not diminish the relevance of our results. On the contrary, it highlights the complexity of observational studies and the importance of accounting for multiple interacting stressors in future assessments. We also emphasize the limited availability of comprehensive land management data as a shortcoming of the study and a direction for future research (lines 297-299).

Regarding the Structural Equation Modeling (SEM) that was previously included with the aim at partitioning the effects of agriculture into components—such as soil properties, pesticide loads, and other environmental variables—we have decided not to include it in the revised version for several reasons:

1. **Lack of comparability between analyses:** Our SEM analysis used aggregated metrics of pesticide occurrence and cumulative risk, rather than compound-specific concentration data as in the GLMs. SEMs based on (distinct) pesticide concentrations encountered methodological limitations. As a result, SEMs could neither support nor be directly compared to the GLMs, since they relied on different pesticide information. Given that the manuscript now focuses on results from the GLMs, retaining SEMs based on aggregated pesticide metrics would offer limited added value and could introduce narrative confusion.
2. **Scope and focus of the study:** Our primary goal was to assess associations between pesticides and changes in soil biodiversity across a large spatial scale. Including a complex SEM framework could risk diluting the focus of the paper and may give an impression of a more definitive causal inference than the available data support. By keeping the analysis straightforward, we aim to transparently communicate the limitations and early-warning nature of our findings, while avoiding over-interpretation.
3. **Future research direction:** We view SEM as a valuable approach for follow-up studies where more detailed land-use and management data (e.g., temporal pesticide application history, fertilizer use, tillage intensity) are available.

Overall, we believe the revised manuscript more clearly and directly addresses the relationship with agricultural intensification in the discussion (linking the limitations of our study to the works of Tsiafouli et al., and de Vries et al., line 299), without introducing unnecessary complexity in the analyses.

Despite these limitations, we acknowledge the strength of the study in terms of its broad taxonomic scope and geographic coverage. In particular, the consistent effects observed across ecosystems, including woodlands, are compelling, as they may allow for interpretation independent of land-use intensity. Comparing these effects more systematically with those for croplands would strengthen the case that the paper makes.

Many thanks for this comment. In the results section, we have compared the analyses conducted in croplands with analysis conducted on all ecosystems (including croplands, grasslands and woodlands (apart for metagenomics analysis)), see lines 216-242). We further highlight the importance of including non-croplands to detect spill-over effects and broader ecological patterns (lines 241-242, lines 329-332, Supplementary Discussion).

Tsiafouli, M. A., Thébault, E., Sgardelis, S. P., De Ruiter, P. C., Van Der Putten, W. H., Birkhofer, K., ... & Hedlund, K. (2015). Intensive agriculture reduces soil biodiversity across Europe. *Global change biology*, 21(2), 973-985.

De Vries, F. T., Thébault, E., Liiri, M., Birkhofer, K., Tsiafouli, M. A., Bjørnlund, L., ... & Bardgett, R. D. (2013). Soil food web properties explain ecosystem services across European land use systems. *Proceedings of the National Academy of Sciences*, 110(35), 14296-14301.

(4) Microbial richness and diversity metrics

While the authors provide a valid (albeit theoretical) rationale in the rebuttal for including both richness and diversity, the different results for richness vs diversity are unconvincing and the main text does not adequately discuss the contrasting richness and diversity patterns. Presenting both metrics thus remains unwarranted as it stands.

We would suggest transferring the results on diversity to the Supplementary. The differences between richness and diversity responses in Fig. 3 are quantitative rather than qualitative. In Fig. 4, the different lines to richness and diversity can hardly be distinguished. Removing diversity would increase its readability and its value to the paper. It thus seems that the added value of including diversity is limited; excluding it from the main text would simplify and strengthen the narrative of the paper.

Thank you for raising this point. We have moved the diversity to the supplementary materials (e.g., please see new Supplementary Fig. S8).

The rebuttal states: '... in the presence of certain pesticides, soils may become dominated by a few more resilient taxa (with increased biomass of these groups), while more sensitive or less competitive taxa decline or disappear, leading to reduced OTU richness and diversity.' This would be a valuable addition to the main text (and also suggests that responses of richness and diversity are not divergent as suggested beforehand).

Thank you for this suggestion. However, we need to stress that this statement is based on the biomass dataset presented in previous rebuttal letter, which is limited to bacteria and fungi only, for which the richness and diversity are more correlated (Pearson's $r=0.90$ for bacteria, $r=0.85$ for fungi) than for other organisms, such as archaea and fauna. Also, including the biomass dataset in the study would add another layer of complexity, which might not support the simplification of the analyses and narrative.

Referee #4 (Remarks on code availability):

NA

Referee #5 (Remarks to the Author):

I co-reviewed this manuscript with one of the reviewers who provided the listed reports.

Referees' comments:

Referee #1 (Remarks to the Author):

The revised manuscript “Pesticide residues in soils affect soil taxonomic and functional biodiversity” by Koninger et al evaluated the effects of pesticides on soil microbial and faunal communities across multiple ecosystems in Europe. They report that ~70% sites contained pesticide residues and different soil communities responded differentially to pesticide residues. In revised version they also analysed functional genes involved in carbon, nitrogen and phosphorus cycle which mainly responded positively while soil fungi community declined. Authors concluded that these findings highlight the need for comprehensive assessment of pesticide effects on soil biodiversity for improved regulation.

In my opinion, authors have addressed most technical issues raised by reviewers by including new data from metagenomics, utilising other statistical tools (e.g. Structural equation models) and providing more details where needed. Two key weaknesses remain in my opinion

a. There is a need for better articulation of scientific needs and socio-economic and regulation policy. Combined with space constraint, the authors have understandably taken a more cautious approach, and rather focused on research requirements for future regulation/ policy impact. However, in my opinion, there is still some room for this to be strengthened and I have highlighted below where and how.

b. The Discussion section seems disjointed. This is partially due to address multiple issues raised by reviewers. Below I have provided some notes to improve this.

We thank you for your positive feedback on previous revisions brought to the manuscript. In response to your comment on the articulation of scientific needs and regulatory context, we have revised the Introduction to more clearly position our study within current regulatory gaps (e.g., lines 65-66). In the Discussion, we have restructured the text to ensure clearer focus and flow. Specifically, as suggested, we now dedicated distinct, thematically cohesive sections to key issues: regulatory challenges together with policy implications (lines 304-323) and future research needs to address (lines 324-335).

We also moved some extended examples and supporting context to a Supplementary Discussion, allowing us to maintain clarity and narrative focus within the main text while still providing depth for interested readers.

Other comments

1. L44. It should be ‘59% of the Earth’s biodiversity—’

Done, thank you. (line 42)

2. L46. Please provide reference for impact of soil adsorption and absorption impact on pesticide residue persistence.

We have added following reference (line 44): Wauchope, R. D., Yeh, S., Linders, J. B. H. J., Kloskowski, R., Tanaka, K., Rubin, B., ... & Unsworth, J. B. (2002). Pesticide soil sorption parameters: theory, measurement, uses, limitations and reliability. *Pest management science*, 58(5), 419-445

3. L52. In my opinion, it should read- ‘--soil organisms in ecosystem functions including pesticide degradation’.

Done, thank you (line 51).

4. Line 66. This will be good place to include one or two sentence on limitation of current regulations i.e. based on assay of single compounds and on specific organisms.

Thank you, we have extended this point to also include how different field conditions as well as long-term exposure effects are not adequately considered in pesticide authorisation processes (as also pointed by Riedo et al. 2025; lines 65-66).

Riedo, J., Rillig, M. C. & Walder, F. Beyond Dosage: The Need for More Realistic Research Scenarios to Understand Pesticide Impacts on Agricultural Soils. *Journal of Agricultural and Food Chemistry* (2025).

5. L183-185. Impact on N₂ fixing taxa- have authors accounted for presence of legume plants? Presence of legume plants will increase relative abundance of symbiotic N₂ fixer which can potentially bias this dataset.

Thank you for raising this point. To test this suggestion (even though soil DNA extracts are dominated by DNA from free-living organisms rather than strict endosymbionts), we have examined the effect of crop type (legume vs. non-legume) on N-fixing taxa relative abundance (metabarcoding) and N-fixation genes abundance and diversity (metagenomics) (Fig A below). The ANOVA output showed no effect of legumes. However, there was a slightly higher abundance of N-fixing taxa in legume crops. Further analysis revealed that N-fixation genes abundance and diversity increases with humidity, which can be explained by the fact that legumes are typically cultivated in mesic regions (Fig B). This led us to the conclusion that the bias from legumes is likely negligible to be included in our models (GLMs). Also, we have now removed the correlation analysis from the present version of the manuscript, reshaping and relocating them in Supplementary Results S4 following comments from other reviewers. There, we only describe the stronger correlation patterns (Spearman's coefficient value around or above |0.4|) such that the results on bacterial N fixation are

not approached as significant relationships anymore (based on a comment from reviewer 2).

6. L302-313. Could this finding be linked to microbial activity, and pesticide adsorption? For example, lower temperature means lower microbial activities that in turn can reduce rate of pesticide degradation.

We agree and hence, we have added the following sentence to the interpretations in the Supplementary Results S3: “These findings point to lower pesticide degradation, potentially connected to lower microbial activities in colder areas”. In addition, in order to streamline the manuscript’s narrative (as requested by reviewers 4 and 5), all results referring to the relationships between environmental variables and the presence of pesticides in shaping soil biodiversity have been transferred to Supplementary Results S3.

7. L316-317. You have already said this earlier, suggest to delete.

Done, thank you.

8. L330-333. This belongs to results - not in discussion section. Authors should highlight rationale or implication of this finding.

Thank you for raising this point. We have more implicitly linked this finding to the capacity of bacteria to metabolise pesticides, which may potentially give them an advantage in pesticide-rich environments (lines 278-283 of the discussion).

9. L336. Suggest use the term – with reduced diversity of ‘some’ beneficial organisms,--

Done, thank you (line 261 of the discussion).

10. L343- suggest – soil food web and ‘potentially’ ecosystem functioning by—

Thank you for this suggestion. We have reformulated the sentence to link it to controlled experiments (lines 257-258).

11. L428-435. Not sure what does this section add. There are too many – ‘future research is needed’ in the discussion section. This is good that authors should highlight those but that should be based on following in my opinion. How does the present work advance the discipline and how does this open up new area of research (future research) and what other evidence are needed for societal impact- in this case regulation policy. So, my suggestion to authors will be to consolidate future research requirements in one paragraph (may be paragraph just before conclusions section).

Many thanks for this valuable suggestion. We added a paragraph listing all the future research needs as well as how our work can help in advancing the discipline (lines 324-335 of the discussion).

12. L445-458. Suggest to reducing this section to half and bring about 2- 3 sentence on requirements for improving specific regulations.

We have combined this section with the one dealing with future research and improving specific regulations (lines 324-335 of the discussion).

Overall, I believe that the revision has significantly improved the manuscript and, in my opinion, if articulated appropriately, it can have significant impacts on both the scientific discipline and relevant policies.

Many thanks for your feedback and suggestions. We integrated those into our revised manuscript to improve the narrative towards a more impactful message.

Referee #1 (Remarks on code availability):

The analyses seem straight forward to me and believe any one can repeat those analyses.

Regarding code, I do not consider my self expert on this and hopefully other reviewers can provide more informed advice.

Referee #2 (Remarks to the Author):

I appreciate the authors' efforts to address the concerns pointed out in the first-round review, particularly in clarifying the novelty of study, expanding methodological details, and strengthening the discussion. However, I still suggest rejecting this manuscript because some of the explanations are not reasonable, and does not resolve the fundamental science questions. Therefore, the new version does not reach the quality standard of Nature and requires more-than-major revisions to do so.

1. "Large scale"

The authors repeatedly emphasize the "large scale" of their study, considering multiple pesticides, species, countries, and ecosystems. However, I believe this approach, as presented, is not innovative enough to justify publication in Nature. For instance, while the authors analyze the responses of multiple species to pesticides, do the conclusions differ from those of studies focusing on a single species? Have any new scientific insights been gained? Conclusions such as "Soil biodiversity (taxonomic and functional) responded differently to pesticides depending on organism type" and "the effects of soil pesticide residues may extend beyond their target organisms" are just expected findings on the basis of prior single-species studies.

We understand your concerns. However, the scale covered in this study (Europe), the number of pesticides analyses, the number of ecosystem types, and the wide range of soil organisms studied has never been done before. Additionally, no previous work has involved such a variety of functional gene groups' responses (for C, N, P cycling) to pesticides at the continental scale either. Most importantly, our results demonstrate for the first time the importance of pesticides compared to other factors (soil properties, ecosystem type and climate) in influencing soil communities. That is a major step forward, and such analysis can only be done with the large dataset included here.

In order to stress these facts, we have reformulated the conclusions to better emphasise the strength of our findings from early on (please see the abstract).

Moreover, the authors claim their study is unprecedented in scale and attempt to consider multiple species, yet they overlook annelids, despite offering some justification for this omission. Additionally, soil properties, climate, and other factors can influence the impact of pesticides on soil organisms. Can the European sampling sites truly represent global climate types (e.g., as classified by the Köppen Climate classification) or soil properties (e.g., as described by SoilGrids)?

We did not claim that the present study exhaustively covered all environmental conditions nor represented global climate and soil property types (please let us know where this was stated otherwise). However, we do cover a wide range of sites (373 sites in 26 countries - see Figure 2) and our sampling sites represent major ecosystem types and climate in Europe. The Köppen Climate classification, although valuable and very informative, represents a classification by vegetation type. Studies focusing on soil biodiversity show that they do not follow the same patterns as above-ground organisms (Cameron et al. 2019)

Annelids were not overlooked. However, the methodology used in this study does not allow for a representative sample of these organisms as we discussed before (Köninger et al. 2023) and trying to interpret these results would have led to too much speculation in our point of view. We have now specifically addressed this in the discussion, stating that soil biodiversity assessment is not exhaustive and missing several groups, among which annelids (lines 290-294), linking this gap to methodological constraints (e.g., soil volume for annelids or limitations of reference databases for tardigrades and rotifers; see Methods lines 519-521). Still, this study analyses many more biodiversity metrics (a wide range of taxonomic groups and functional gene responses)” as previous studies and at a much larger scale.

Cameron, Erin K., et al. Global mismatches in aboveground and belowground biodiversity. *Conservation Biology*, 2019, vol. 33, no 5, p. 1187-1192.

Köninger, Julia, et al. Ecosystem type drives soil eukaryotic diversity and composition in Europe. *Global change biology*, 2023, vol. 29, no 19, p. 5706-5719.

2. Pesticide

The authors acknowledged that “pesticide occurrence” is misleading and replaced it with “pesticide metrics.” However, this change in terminology does not resolve the fundamental issue: the manuscript still lacks a clear and rigorous definition of how pesticide presence is quantified and interpreted. Simply renaming the term does not clarify the methodology or address my concern that lumping various pesticide-related parameters into a single term obscures their distinct ecological implications. The authors should explicitly define how each pesticide metric is measured, normalized, and integrated into their analysis.

To address your concerns, we have produced a new correlation matrix only including aggregated metrics for pesticide occurrence and risk commonly used in the literature. Pesticide occurrence metrics included (i) the total number of pesticide residues detected per sites, the number of (ii) herbicides, (iii) fungicides and (iv) insecticides. These metrics were compared to the cumulative risk, and additional risk indicators tailored per pesticide type (with one risk indicator for herbicides, one for fungicides, one for insecticides). The calculations and use of the aggregated pesticide metrics are further detailed in the Methods (lines 716-738 of the manuscript). We present the correlations as a simplistic overview of trends between soil biodiversity and pesticides and present the limitation of using aggregated metrics. Following a request from reviewers 4/5 to streamline the manuscript's content, we have transferred the correlations analyses to the supplementary material as complementary analyses (please see Supplementary Results S4).

Additionally, their response does not clarify how increased detection (e.g., owing to more sensitive analytical techniques) is distinguished from actual changes in pesticide presence in the environment.

We apologize for not addressing this point correctly. In the initial submission, we used the term "increased pesticide occurrence" to describe the consistent trends observed in the correlation analyses. For example, bacterial richness was positively correlated with multiple pesticide metrics (e.g., the number of residues detected per site, the cumulative concentration, and the number of fungicides). Therefore, rather than listing each association separately, we summarized these findings using the broader term "pesticide occurrence". After reading your comment, we now understand your point and we agree we should avoid using a broad common term for pesticide metrics representing different measurements. In the Supplementary Results S4, we have now reduced the number of metrics to describe pesticide occurrence and risk.

We would also like to clarify that all samples were analysed using a uniform analytical protocol, with standardized extraction, instrumentation, and detection procedures applied across all sites. Our limits of quantification (LOQs) fall within the range commonly reported in the literature; therefore, variations in the number of detected pesticides reflect true differences in environmental presence rather than inconsistencies in detection methodology. It was not our intention to state that pesticide detection in the environment increased (this we cannot address with this study and that was also not our goal).

The authors argue that the cumulative risk is based on multiple taxa (annelids, collembolans, mites) rather than just arthropods. However, this still does not justify the application of NOEC-derived risk assessments to soil microbiota, which differ significantly in sensitivity and ecological function. The fact that NOEC is derived from a limited number of model organisms makes its extrapolation to the broader soil community problematic. Simply acknowledging this limitation in the Methods and

Discussion does not resolve the core issue: the risk metric does not adequately capture microbial responses to pesticides. This methodological weakness undermines the reliability of the ecological risk assessments presented.

The cumulative risk has only been included in correlation analyses (old matrix Fig. 2), which are inherently correlative and not intended to imply causation. While we acknowledge the limitations of using NOEC values established for a limited number of model organisms (now in Supplementary Results S4 and lines 305-310 of the discussion), the NOEC remains one of the most established metrics in ecotoxicological risk assessment, and we believe it is important to report it. Presenting how microbial and other biodiversity variables correlate with cumulative risk and other risk indicators – despite their clear limitations – can still be informative for readers (in particular ecotoxicologists). However, to address this issue more cautiously, we have relocated the risk-related figures to the supplementary material (Supplementary Results S4), where we deepen the discussion on their limitations. This allows interested readers to access the information while supporting our broader argument that risk assessment frameworks should be expanded to better include (microbial) community responses and their functions (lines 305-313 of the Discussion).

The authors state that they initially tested risk quotient models (pesticide concentration/NOEC_{min}) but did not include them owing to distortions caused by NOEC limitations. However, this does not address my concern: the impact of different pesticide active ingredients varies widely, and merely considering pesticide concentrations without accounting for specific toxicities is insufficient. The manuscript lacks an analysis of how different active ingredients contribute to the observed biodiversity changes. Without this, the study provides an oversimplified view of pesticide effects. A more robust approach would involve grouping pesticides on the basis of their mode of action or toxicity profiles to different soil biota rather than treating them as a uniform risk factor.

Our analysis is focused on exploring patterns between pesticide residue concentrations in soils and soil biodiversity at a certain point in time. The goal of our study is not to provide a risk assessment of pesticides but help to identify shortcomings in current regulatory framework using pesticide concentration in NOEC-based risk assessments. The use of GLMs including distinct pesticide concentrations next to environment (as already presented in previous versions of the manuscript) provides how the concentrations of different active ingredients contribute to the observed biodiversity changes.

Notably, mode of action is provided for target organisms, and cannot be applied to non-target groups, which are the primary focus of our study. Toxicity profiles are relevant to non-target groups, but such data are often inconsistently reported (and do not embrace all the diversity of our groups of interest), making it impossible to run standardised analyses. We agree that this is a major limitation in the field and a key area where

further harmonized data collection would greatly enhance ecological risk assessments. This is now addressed in the Supplementary Results S4 and lines 308-311 of the discussion.

We agree that incorporating pesticide properties to explain observed effects is a constructive approach. Hence, we further analysed this based on your suggestion. To this end, we hypothesised that the pesticide effects on particular components and properties of soil biota can be explained by pesticide properties. To test this hypothesis, we related pesticides to each other based on:

1. Empirical and regulatory properties that may reflect potential hazards, including regulation status (approved or not), target group (herbicides, fungicides, insecticides), effect spectrum (broad or narrow), access type (systemic or non-systemic), soil mobility (based on K_f , K_{foc} , $1/n$), and persistence (DT50-field).
2. Structural similarities and physicochemical properties, using the ChemMine tool and 32 JOELib descriptors.
3. Using the GLM results from our study, we created an effect profile for each pesticide, indicating the type of effect (negative, positive, or none) on soil biota properties described by the 63 variables (please see the “pesticide_molecular_properties” table as additional files for review); central to our analysis. We then generated a similarity matrix based on these effect profiles.

Based on these data we performed Procrustes analysis (please see “Data for Procrustes” table as additional files for review). While it showed significant links between molecular and empirically measured properties ($R^2 = 0.60$, $p = 0.001$), both of these were not significantly related to effect on the studied parameters: $R^2 = 0.62$, $p = 0.17$ for the comparison of molecular similarity vs effect similarity; $R^2 = 0.35$, $p = 0.97$ for empirical properties vs effect similarity (please see the “Figure_Procrustes” in the additional files for review).

We feel that involving these additional pesticide properties gave no information gain. In an attempt to focus the narrative on fewer analyses/information in the main manuscript and associated supplements, as requested by other reviewers, we propose to leave this additional analysis out. However, we would be happy to include it in the supplements upon request, should you or the Editor consider it necessary.

3. Soil biodiversity

The authors acknowledge that certain groups are excluded owing to methodological constraints but do not provide a convincing justification for omitting them entirely. Other studies have successfully incorporated earthworm communities via DNA metabarcoding or morphological approaches¹¹.

The additional analysis of annelids is appreciated, but the authors ultimately decided to exclude them on the basis of low dataset quality. This exclusion, rather than improving

the study, highlights a major limitation: the study still lacks a true representation of soil macrofauna, which weakens its conclusions about soil biodiversity.

The previous rebuttal letter and manuscript already approached the limitations of annelid assessment with LUCAS sampling design, owing to a much lower soil volume than the one used in Pansu et al. (2015). The latter involved 10 soil cores per subplot of 0.5m².

We agree that this methodological constraint prevent us from having a robust dataset on soil macrofauna (e.g. 40g of soil were used by Lilja et al. (2023) and 80g in Cuartero et al. (2025)), but unlike those which targeted annelids specifically, we targeted a wider community including archaea, bacteria, fungi, protists, nematodes and arthropods.

Therefore, to better inform the reader on the scope of organisms covered in our analyses, we have listed the major groups considered in this study as well as the groups not included (lines 290-294 of the discussion), referring to methodological constraints (lines 519-520 of the Methods) and flagging it for future research (line 334 of the discussion).

Cuartero, J., et al. Earthworm and enchytraeid indicator taxa of different land-use types identified using soil DNA metabarcoding, *Applied Soil Ecology* (2025)

Lilja, M. A. et al. Comparing earthworm biodiversity estimated by DNA metabarcoding and morphology-based approaches. *Applied Soil Ecology* 185, 104798 (2023).

Pansu, J. et al. Landscape-scale distribution patterns of earthworms inferred from soil DNA. *Soil Biology and Biochemistry* 83, 100-105 (2015).

The authors now recognize the limitations of assigning functions on the basis solely of taxonomic classification, yet their response does not sufficiently address my original concern. Inferring microbial functional roles on the basis solely of genus-level taxonomy is problematic because functional potential can vary significantly within a genus. Without direct evidence from functional gene profiling or metabolic analyses, these assignments remain speculative.

We are aware of functional trait variability and specifically focused on a limited number of most well-studied traits. Moreover, all approaches to functional annotation, including gene profiling and metabolic analyses, have inherent limitations because metabolic activity is highly context-dependent. This is especially true for complex traits like lifestyle (e.g., pathogenic, symbiotic, or endobiotic), which are often performed by the same organism and depend more on the host's species, genotype, and immune status than on the microbe itself.

Furthermore, databases such as FAPROTAX and FungalTraits are based on expert-curated observations from multiple studies, and their functional annotations are more

relevant to ecological contexts than those inferred solely from gene profiling. This approach is widely accepted, as evidenced by over 2800 published research papers employing FAPROTAX and FungalTraits for functional inference.

Therefore, we believe that grouping organisms into functional categories is supported by the literature (e.g., Fierer et al. 2017; Ferris et al. 2001; Tedersoo et al. 2014; Delgado-Baquerizo et al. 2016; Oliverio et al. 2020), adding important information to our analyses. Also, functional groups have been used in recent articles (see van den Hoogen et al. 2019, Liu et al. 2022, Labouyrie et al. 2023). However, we are also aware of their limitations and these are now further addressed in the Supplementary Discussion.

Labouyrie, M., et al. Patterns in soil microbial diversity across Europe. *Nature Communications*, 2023, vol. 14, no 1, p. 3311.

Liu, S., et al. Phylotype diversity within soil fungal functional groups drives ecosystem stability. *Nature Ecology & Evolution*, 2022, vol. 6, no 7, p. 900-909.

van den Hoogen, J., et al. Soil nematode abundance and functional group composition at a global scale. *Nature*, 2019, vol. 572, no 7768, p. 194-198.

Although the authors mention incorporating metagenomic insights, they do not clarify whether these were used to redefine functional groups or simply to provide additional results. If functional annotations are still largely based on taxonomy rather than gene function, the issue remains unresolved.

Metagenomic data were intended to provide a more detailed view of the community's functional response to pesticides at the gene level. Effects on genes involved in three different cycles (carbon, nitrogen and phosphorus) were assessed. These were analysed separately from the metabarcoding data. Regarding the use of taxonomic functional groups, please see our responses in the comment above.

The authors state that they have expanded their analyses to include 48 functional gene groups related to biogeochemical cycles, but there are still major concerns: The selection of functional genes appears arbitrary, and it is unclear how these gene groups were defined or validated. Without a clear methodological explanation, the robustness of these functional assignments is questionable.

The approaches for gene identification in metagenomes are described in the Methods (see lines 552-569). They are state-of-the-art yet conceptually standard (Bahram et al., 2018 *Science*; Dulya et al., 2024 *Environment International*).

The choice of genes of interest is non-arbitrary as they inform on the most important biogeochemical cycles and encode the soil biota functions critical for such ecosystem services as greenhouse gas regulation, nutrient and carbon retention. However, we acknowledge that we did not provide justification for this choice, and it has been corrected (see lines 572-582 in the Methods).

Bahram, M., et al. (2018). Structure and function of the global topsoil microbiome. *Nature*, 560(7717), 233–237. <https://doi.org/10.1038/s41586-018-0386-6>

Dulya, O., et al. (2024). A trait-based ecological perspective on the soil microbial antibiotic-related genetic machinery, *Environment International*, Volume 190, 108917, ISSN 0160-4120, <https://doi.org/10.1016/j.envint.2024.108917>

4. Analysis

The new correlation analyses and GLMs do not fundamentally resolve the problem of pesticide impact assessment, as functional group definitions remain vague. The relationship between pesticides and functional diversity needs more rigorous exploration, potentially through direct metagenomic or transcriptomic analysis rather than inferred functions.

The conclusions of studies on the relationship between pesticides and microbial diversity contradict widely accepted perspectives. Specifically, the analysis indicated that bacterial richness is positively correlated with most pesticides, which deviates from mainstream findings. While the authors propose a reasonable explanation—that pesticide exposure reduces fungal diversity, alleviates competition and promotes pesticide-degrading bacterial communities—this hypothesis lacks robust data support. The current discussion is largely speculative and lacks concrete statistical or experimental validation. The authors are encouraged to provide more detailed supporting data.

We agree that direct metagenomics offers more precise insight into the relationship between pesticides and functional diversity; this is why we used such data in the previously revised manuscript already. Regarding the functional group analysis, we acknowledge the limitations of taxonomic-based annotations. However, as noted above, they are grounded in established ecological trait databases and provide functional insights beyond community composition alone.

A substantial body of experimental research has shown that pesticide applications can confer a competitive advantage to those bacteria capable of degrading or tolerating these compounds (e.g., metabarcoding-based work Ni et al., 2025) due to the release from predator pressure. These alterations in trophic community structure, have been comprehensively reviewed in the classical work by Staley et al., 2015. Also, a recent study by Romero et al. 2025, demonstrated that fungal richness is suppressed by pesticides while bacterial richness is not affected and their functional role increases.

Note, while studies indeed do report negative pesticide effects on bacteria (e.g., the recent meta-analysis by Wan et al., 2025), these studies often focus on physiological responses in model strains, such as *Bacillus*.

In our study, we observed reduced fungal diversity associated with glyphosate concentrations, and increased diversity of bacterial genes involved in phosphonate degradation with rising AMPA levels. However, we acknowledge that our study is not a controlled food web experiment and cannot confirm competition mechanisms. We now present the proposed mechanisms more cautiously in the discussion (lines 273-283) and linked them to the supporting studies of Ni et al. and Staley et al. (line 277).

Ni et al., 2025. Increasing pesticide diversity impairs soil microbial functions. *Proceedings of the National Academy of Sciences* 122, e2419917122
<https://doi.org/10.1073/pnas.2419917122>

Romero, F., Jiao, S., & van der Heijden, M. G. (2025). Impact of microbial diversity and pesticide application on plant growth, litter decomposition and carbon substrate use. *Soil Biology and Biochemistry*, 109866.

Staley et al., 2015. A synthesis of the effects of pesticides on microbial persistence in aquatic ecosystems. *Critical reviews in toxicology* 45, 813-836
<https://doi.org/10.3109/10408444.2015.1065471>

Wan et al., 2025. Pesticides have negative effects on non-target organisms. *Nature Communications* 16, 1360 (2025). <https://doi.org/10.1038/s41467-025-56732-x>

5. Line-by-line feedback:

Lines 192-221: The authors have replaced sales statistics with pesticide residue concentration data from Vieira et al. (2023), which is a more relevant dataset. However, the manuscript does not clarify whether measured concentrations or modeled estimates were used for analysis. Furthermore, pesticide concentrations alone do not reflect actual exposure levels to soil biota—other factors such as bioavailability, degradation kinetics, and interactions with soil organic matter should be considered.

Pesticide concentrations used in this study were obtained by measurements of soil samples collected within the framework of the LUCAS module named LUCAS Soil Pesticides. To clarify this point, we have added this information in the first paragraph of the Methods (lines 468-471).

While the measurement methods used in our study are well established for pesticide residues and widely used in soil biodiversity studies, we agree that bioavailability data would provide additional insights into actual exposure levels. However, bioavailability is highly organism-specific, and given the wide diversity of organism groups we examined,

it is not feasible to incorporate a single, standardised measure of bioavailability across all of them. We also included soil adsorption coefficients (K_f , K_{foc}) in additional analyses (Procrustes analyses; please see our response in the current response letter about the link between pesticide properties and revealed effects on biota). Subsequently, we added this lack of data on pesticide bioavailability, degradation kinetics and interaction with soil organic matter as limitation of the current study (please see Supplementary Discussion). There, we also added how pesticide persistence does not reflect soil biota exposure to pesticides.

Lines 76-83: The authors cite their presence across ecosystem types, but they do not justify why these specific groups were prioritized over others. Their relevance as biological indicators is cited from Karpouzias et al. (2022), but the authors do not provide evidence that these functional groups respond strongly to pesticide exposure. The reliance on taxonomy-based functional assignments (rather than metagenomic/metatranscriptomic evidence) is problematic, as it does not accurately reflect functional capabilities.

Functional groups were selected based on their primary roles in soil ecosystem processes that are critical for soil functioning, such as plant productivity, nitrogen loss and retention, and organic carbon release (Ferris et al. 2001, Tedersoo et al. 2014, Delgado-Baquerizo et al. 2016, Oliverio et al. 2020). The text in the Methods has been further extended to illustrate the ecological roles of these groups (lines 538-546). Please see our previous response regarding the reliance of functional assignments based on taxonomy.

Delgado-Baquerizo, M. et al. Microbial diversity drives multifunctionality in terrestrial ecosystems. *Nature communications* 7, 10541 (2016).

Ferris, H., Bongers, T. & de Goede, R. G. A framework for soil food web diagnostics: extension of the nematode faunal analysis concept. *Applied soil ecology* 18, 13-29 (2001).

Oliverio, A. M. et al. The global-scale distributions of soil protists and their contributions to belowground systems. *Science advances* 6, eaax8787 (2020).

Tedersoo, L. et al. Global diversity and geography of soil fungi. *science* 346, 1256688 (2014).

Line 88: The authors have removed pesticide persistence from the analysis, which is appropriate, as persistence alone does not equate to biological effects. However, the manuscript should still discuss why persistence is not a reliable indicator of pesticide impact on biodiversity—for example, highly persistent pesticides may have low bioavailability, and rapidly degrading pesticides can still exert significant effects on

microbial communities.

This issue is now discussed in the Supplementary Discussion.

Line 139: The Pesticide Properties Database categorization only applies to target organisms (weeds, fungi, insects), not to nontarget soil biota (archaea, bacteria, protists, etc.). The authors admit that the pesticides analyzed do not directly target archaea or bacteria, yet they still assume broad-spectrum effects on microbial communities. The cited studies (Ma et al. 2021; Puglisi et al. 2012) suggest potential indirect effects of fungicides on bacterial diversity, but this does not justify treating these pesticides as broad-spectrum agents for all soil biotas. If the authors want to claim broad-spectrum effects, they should provide quantitative toxicity data for archaea, bacteria, and fungi, rather than assuming indirect effects.

Thank you for raising this point. The broad spectrum has been removed from the analyses (please see correlation analyses transferred to Supplementary Results S4), also in an attempt to focus the narrative on fewer analyses/information in the main manuscript (as asked by reviewers 4/5).

Lines 152-155: The authors acknowledge that summing pesticide concentrations leads to potential confounding but still argue that it is a useful approach for quantifying overall pesticide effects on biodiversity. However, this method introduces serious statistical biases: 1) Nonadditive toxicity: Different pesticides have distinct mechanisms of action and toxicity thresholds. Summing concentrations ignore potential synergistic or antagonistic effects and assume a linear dose–response relationship, which is biologically unrealistic. 2) Varying environmental persistence: Pesticides degrade at different rates, and their bioavailability depends on soil properties. Aggregating concentrations does not account for differential persistence and degradation kinetics, leading to misleading conclusions.

The authors claim to have conducted generalized linear models (GLMs) for individual pesticides, but they do not present a direct comparison between the aggregated and individual models. Without such a comparison, the aggregated pesticide approach remains fundamentally flawed.

We agree on the shortcomings derived from summing all pesticide concentrations per site to obtain an overall cumulative concentration because this would assume a linear-dose response, ignoring the synergistic or antagonistic and the non-additive effects of pesticides. Accordingly, this aggregated metric has been removed from the new Supplementary Results S4. In addition, we have further acknowledged that information on pesticide degradation kinetics and bioavailability would provide more insights into soil biota exposure to pesticides (please see Supplementary Discussion).

Importantly, we would like to stress that we did not use any aggregated pesticide metrics in the GLMs (avoiding the claimed linear dose-response assumptions). The aggregated metrics were only used for correlation analysis (Spearman correlation) without considering other factors such as soil properties and climate. As a result, a direct comparison between aggregated correlation analysis and GLM analysis is not relevant.

Lines 140-155: the authors report that all Spearman correlation coefficients in Figure 2 are below 0.4, which is considered "low" or "very weak" based on standard statistical guidelines. Despite this, they argue that statistical significance (p values) is more important than the strength of the correlation. This is a fundamental misinterpretation of statistical significance. A statistically significant result ($p < 0.05$) merely indicates that the observed correlation is unlikely to be due to chance; it does not imply a strong or biologically meaningful relationship. The effect size (correlation coefficient $|r|$) determines the biological relevance of the relationship. Correlations below 0.3 suggest a negligible association between pesticides and biodiversity metrics. The authors attributed the weak correlations to high spatial and temporal variability in the field data. While variability is expected, the lack of strong correlations suggests that pesticides are not the main driver of soil biodiversity, which contradicts the manuscript's primary claim.

We thank you for your inputs on the correlation analyses. The new correlation matrix (Supplementary Fig. S11) displays the correlations greater than or equal to $|0.3|$. Associated results describe the stronger correlations (around or above $|0.4|$) in Supplementary Results S4.

The claim regarding pesticides being the main driver of soil biodiversity has already been removed from the revised version (January 2025), and was initially based on a different analysis, i.e. the variation partitioning performed on the models (GLMs) presented in previous Fig. 3 (Fig. 4 in the current version). This analysis showed that a big share of the explained variance in certain soil biodiversity metrics was attributed to pesticide concentrations. We have reported this observation in lines 251-252 of the discussion, while acknowledging that pesticides are the second major driver of soil biodiversity patterns after soil properties. Consequently, it is important to note that our initial claim and its revised statement were not based on correlation analyses and their coefficient values.

To justify the weak correlations, the authors conduct a subset analysis in Italy, reporting higher correlations (ranging from -0.65--0.72). However, this analysis is flawed for several reasons: 1) small sample size bias: the Italy dataset includes only 18 sites, making it highly susceptible to random variation and outliers. High correlations in small

samples do not generalize to the entire dataset. 2) selection bias: The authors do not explain how this subset was selected. If the region was chosen post hoc because it shows stronger correlations, this introduces confirmation bias. 3) Ignoring confounding factors: The greater correlations in Italy could be due to local environmental or land-use factors rather than pesticide effects. Without controlling for these factors, the results are inconclusive.

A more appropriate statistical approach would involve multivariate models with interaction terms rather than selectively reporting high correlations from small subsets.

The previous rebuttal letter explained that this subset of sites has been selected to perform analyses on a more homogeneous set of sites, reducing the high spatial variability in landscapes, soil types and biogeographical regions that could lead to weaker correlations. These sites were geographically localised and encompassed a limited range of soil types (calcisols and cambisols; based on the Soil Atlas of Europe). All limitations (sample size bias, confounding factors) raised here were already acknowledged and discussed by us in a previous letter.

Regarding the statistical approach, the aim of the aggregated pesticide metrics used in the correlation analyses is to give a simplified view of the associations between, e.g., pesticide occurrence and soil biodiversity. Such a correlative approach is indeed inherently not causative nor exhaustive, as it fails to account for other factors. We now highlight how these correlations can only be interpreted as indicative trends in Supplementary Results S4. We feel these analyses provide added value as explained here, but we can also remove this if needed as this is not central to our observations and conclusions, we agree.

References:

1. Pansu, J. et al. Landscape-scale distribution patterns of earthworms inferred from soil DNA. *Soil Biology and Biochemistry* 83, 100-105 (2015).
2. Kottek, M., J. Grieser, C. Beck, B. Rudolf, and F. Rubel, 2006: World Map of the Köppen-Geiger climate classification updated. *Meteorol. Z.*, 15, 259-263 (2006).

Referee #4 (Remarks to the Author):

We have reviewed the manuscript with special emphasis to the comments of Ref 3 and the rebuttal. In general, we agree with the conclusion of Ref 3 that ‘this seems like an interesting, timely and potentially significant study that could substantially moves along our understanding of how real-world pesticide exposure via soil might be

affecting the soil biota community in Europe'. The paper shows how widespread agricultural pesticide residues persist in soils and its analyses suggest that soil microbial communities are strongly modified in response. From a cautionary principle these results are of great importance for policy decisions and regulations, although more definite proofs will require further work (as detailed below).

We also agree with Referee #3 that the manuscript presents an overly complex and apparently incoherent narrative, largely due to the sheer number of analyses and datasets included, a problem that only has been partly corrected in the rebuttal and the new version of the manuscript. A few further changes, and some deliberate choices, are advisable to improve the transparency and clarity of the paper.

(1) Bring overview and structure in the many analyses for the purpose of answering the main question of the paper – evaluating the impacts of pesticides on soil biodiversity

The two paragraphs introducing the approach and methods (ll. 67-91) only partly solve the problem of bringing coherence in the many analyses done. The authors have constructed a Methodology diagram which is now hidden as Fig S15 in the Supplementary Material. In our view, this diagram (in modified form) is needed up front to bring clarity and justify the choices made. The left part of the diagram portrays very well the complexity of both pesticides and soil biodiversity. The diagram can line out the approach of tackling this complexity, dealing with colinearities (middle part), resulting in downstream analyses addressing key aspects of the main question of the paper (right part). Here analyses on the full dataset vs. croplands only should be clearly separated. Hence, a conceptual rather than methodological diagram (although methods can be summarized in the caption) will explain the hierarchy of the analyses, and how and why subsequent steps contribute to addressing the main objective of the paper. It also presents a 'roadmap' as a guidance for the subsequent results to come. A place for the diagram in the main text would be very helpful.

We recommend that the authors use this diagram to explain their deliberate choices about which analyses are essential to the central narrative and which could be relegated to supplementary materials. For example, the title emphasizes the effects of pesticide residues on soil biodiversity, yet the inclusion of functional gene analyses is not well justified; it simply pops up in the manuscript.

Many thanks for this suggestion. We have followed your recommendations and focused the narrative on fewer deliberate choices. Consequently, the manuscript content has been streamlined to base all results on our GLMs (variable importance Fig. 3, variation partitioning Fig. 4). This places cropland analyses up front, and the rationale for focusing on these analyses is now detailed at the end of the introduction. The main conclusions from these analyses are compared with those including both croplands and other ecosystem types in the last paragraph of the results section.

Additionally, in the revised manuscript, we describe the analyses at the functional gene level earlier than in the previous version: the title has been edited to show that the study covers both taxonomic and functional levels, and the abstract introduces the concept of functional genes involved in C, N, P cycling earlier on (lines 29, 32, 34). The updated title now is “Pesticide residues in soils affect soil taxonomic and functional biodiversity”.

The correlation analyses formerly presented in Fig. 2 were refined and moved to the supplementary material as complementary analyses (Supplementary Results S4). All outputs using NOEC and derived risk have been removed from the supplementary figures and were regrouped in Supplementary Results S4. There, some risk indicators were correlated to soil biodiversity and were compared to other aggregated metrics for pesticide occurrence in the correlation analyses. The Results paragraph describing the differences in environmental variables in presence/absence of pesticides shaping soil biodiversity have also been moved to complementary analyses (Supplementary Results S3). Results for Shannon diversity have been moved to the Supplements, for example Supplementary Fig. S8.

Following your recommendation, we have converted the previous Supplementary Fig. S15 into a conceptual diagram (Fig. 1). As the analyses are performed for croplands-only, and then repeated for croplands and other ecosystems together, we have included these two approaches in the diagram with the aim of assessing the spill-over effects and broader ecological impacts of pesticides. Complementary analyses (Kruskal-Wallis, correlation analyses with pesticide risk/NOEC and occurrence) are listed on the right of this conceptual diagram, with their respective output.

(2) Highly correlated pesticide metrics

The high correlation between variables, acknowledged in the rebuttal, should be addressed in the main text or in the caption of Fig. 2. Although the authors note that Figure 2 is intended to illustrate general positive correlations and is not used in GLMs, we still find it problematic. Including all highly correlated variables risks overstating the implications. We understand the choice of showing the metrics as they are often used in scientific studies and policy making. However, it should be made clear, early in the caption of Fig 2, that for columns with a similar colour, we are essentially looking at one and the same effect, referring to the relevant Suppl Fig that gives the correlations between the variables.

Otherwise, the solution for handling the correlations between the metrics and the collinearities are appropriate, with a two-step GLM where the first step selects the pesticide concentrations to be included.

Many thanks for raising this point. The following sentence has been added to the caption of Supplementary Fig. S11: “Columns with similar colours indicate the same direction of correlation between pesticide metrics and soil biodiversity, reflecting underlying linear relationships among the pesticide metrics (see Supplementary Fig. S12)”.

(3) Interpreting results for croplands in relation to agricultural intensification

Pesticide use is typically embedded within broader land management regimes, often co-occurring with practices like excessive use of manure and chemical fertilizer, which also strongly affect soil biota. Although the authors attempted to include some management-related variables, these do not capture long-term land-use intensity. The Discussion (ll.449-458) states the problem as focal points for future studies, not as consequential for the interpretation of the current results. This is an important omission. It should be more clearly acknowledged that the observed patterns represent early-warning signals of effects of pesticides rather than evidence for direct causation. Works such as Tsiafouli et al. (2015) and de Vries et al. (2013) have shown how land-use intensity affects soil biodiversity, and with the seminal dataset that the authors present, the question remains, are effects of pesticides in fact caused by agricultural intensification or, vice versa, are pesticides a causal factor behind land-use intensification effects on soil biodiversity.

Thank you for raising this important point regarding the interpretation of pesticide effects in the context of broader agricultural intensification. We fully agree that pesticide application is rarely an isolated practice and often co-occurs with other intensification measures such as excessive fertilizer use or tillage, all of which can influence soil biodiversity. As you pointed out, this co-variation makes it difficult to isolate the specific contribution of pesticides to observed changes in soil biota, especially when using observational, field-based data.

In the revised manuscript, we have made a clearer statement (lines 301-303 of the Discussion) that our results should be interpreted as early-warning signals of potential pesticide impacts, rather than definitive evidence of direct causation. However, we would like to stress that this does not diminish the relevance of our results. On the contrary, it highlights the complexity of observational studies and the importance of accounting for multiple interacting stressors in future assessments. We also emphasize the limited availability of comprehensive land management data as a shortcoming of the study and a direction for future research (lines 297-299).

Regarding the Structural Equation Modeling (SEM) that was previously included with the aim at partitioning the effects of agriculture into components—such as soil properties, pesticide loads, and other environmental variables—we have decided not to include it in the revised version for several reasons:

1. **Lack of comparability between analyses:** Our SEM analysis used aggregated metrics of pesticide occurrence and cumulative risk, rather than compound-specific concentration data as in the GLMs. SEMs based on (distinct) pesticide concentrations encountered methodological limitations. As a result, SEMs could neither support nor be directly compared to the GLMs, since they relied on different pesticide information. Given that the manuscript now focuses on results from the GLMs, retaining SEMs based on aggregated pesticide metrics would offer limited added value and could introduce narrative confusion.
2. **Scope and focus of the study:** Our primary goal was to assess associations between pesticides and changes in soil biodiversity across a large spatial scale. Including a complex SEM framework could risk diluting the focus of the paper and may give an impression of a more definitive causal inference than the available data support. By keeping the analysis straightforward, we aim to transparently communicate the limitations and early-warning nature of our findings, while avoiding over-interpretation.
3. **Future research direction:** We view SEM as a valuable approach for follow-up studies where more detailed land-use and management data (e.g., temporal pesticide application history, fertilizer use, tillage intensity) are available.

Overall, we believe the revised manuscript more clearly and directly addresses the relationship with agricultural intensification in the discussion (linking the limitations of our study to the works of Tsiafouli et al., and de Vries et al., line 299), without introducing unnecessary complexity in the analyses.

Despite these limitations, we acknowledge the strength of the study in terms of its broad taxonomic scope and geographic coverage. In particular, the consistent effects observed across ecosystems, including woodlands, are compelling, as they may allow for interpretation independent of land-use intensity. Comparing these effects more systematically with those for croplands would strengthen the case that the paper makes.

Many thanks for this comment. In the results section, we have compared the analyses conducted in croplands with analysis conducted on all ecosystems (including croplands, grasslands and woodlands (apart for metagenomics analysis)), see lines 216-242). We further highlight the importance of including non-croplands to detect spill-over effects and broader ecological patterns (lines 241-242, lines 329-332, Supplementary Discussion).

Tsiafouli, M. A., Thébault, E., Sgardelis, S. P., De Ruiter, P. C., Van Der Putten, W. H., Birkhofer, K., ... & Hedlund, K. (2015). Intensive agriculture reduces soil biodiversity

across Europe. *Global change biology*, 21(2), 973-985.

De Vries, F. T., Thébault, E., Liiri, M., Birkhofer, K., Tsiafouli, M. A., Bjørnlund, L., ... & Bardgett, R. D. (2013). Soil food web properties explain ecosystem services across European land use systems. *Proceedings of the National Academy of Sciences*, 110(35), 14296-14301.

(4) Microbial richness and diversity metrics

While the authors provide a valid (albeit theoretical) rationale in the rebuttal for including both richness and diversity, the different results for richness vs diversity are unconvincing and the main text does not adequately discuss the contrasting richness and diversity patterns. Presenting both metrics thus remains unwarranted as it stands.

We would suggest transferring the results on diversity to the Supplementary. The differences between richness and diversity responses in Fig. 3 are quantitative rather than qualitative. In Fig. 4, the different lines to richness and diversity can hardly be distinguished. Removing diversity would increase its readability and its value to the paper. It thus seems that the added value of including diversity is limited; excluding it from the main text would simplify and strengthen the narrative of the paper.

Thank you for raising this point. We have moved the diversity to the supplementary materials (e.g., please see new Supplementary Fig. S8).

The rebuttal states: '... in the presence of certain pesticides, soils may become dominated by a few more resilient taxa (with increased biomass of these groups), while more sensitive or less competitive taxa decline or disappear, leading to reduced OTU richness and diversity.' This would be a valuable addition to the main text (and also suggests that responses of richness and diversity are not divergent as suggested beforehand).

Thank you for this suggestion. However, we need to stress that this statement is based on the biomass dataset presented in previous rebuttal letter, which is limited to bacteria and fungi only, for which the richness and diversity are more correlated (Pearson's $r=0.90$ for bacteria, $r=0.85$ for fungi) than for other organisms, such as archaea and fauna. Also, including the biomass dataset in the study would add another layer of complexity, which might not support the simplification of the analyses and narrative.

Referee #4 (Remarks on code availability):

NA

Referee #5 (Remarks to the Author):

I co-reviewed this manuscript with one of the reviewers who provided the listed reports.

Referee #1 (Remarks to the Author):

The manuscript 'Pesticide residues in soils affect soil taxonomic 1 and functional biodiversity' by Köninger et al investigated the effects of multiple pesticides on soil biodiversity and key functional gene groups across Europe. They report pesticide residues from ~70% sites and identify complex and widespread non-target impacts on soil biodiversity. Pesticides also impacted functional microbial groups specifically phosphorus and nitrogen cycling and suppressed many beneficial taxa, They concluded that key component of soil biodiversity should be included pesticide regulation to protect soil biodiversity.

I believe that the manuscript advances the discipline significantly and will have impact beyond academia and research. In my opinion methodological and statistical approaches are appropriate and robust. Authors have addressed all issues raised by me.

I also believe, Reviewer 2 comments were appropriately addressed in the revised version. I also agree with authors the including annelids based on data generated will be misleading and the methodological approach can not cover annelid diversity or community composition. Constructive comments such as incorporating pesticide properties to explain observed effects were satisfactorily addressed by carrying out additional statistical analyses. They have also revised sentences for clarity and limitations of current manuscript, and in my opinion, these are appropriate responses.

Many thanks for this positive feedback.

Some minor comments

L 39. integrate functional and taxonomic endpoints- 'should this be integrate functional and taxonomic characteristics---

Many thanks, integrated in line 35.

L67-68. Will this be better expression --"-highlighting a need for holistic approach in environmental risk assessment frameworks"

Many thanks, we edited the sentence as follows: *"As a result, the broader ecological impacts of pesticide use on soil life should be better represented into future risk assessments of regulations, moving towards a more holistic approach"* lines 65-67.

Figure 1: It has too much text and most of them too small fonts to read. Either improve the quality or better move this to supplementary materials.

Thank you for this comment. However, having the figure at the beginning of the manuscript was a request from other reviewers in order to guide the reader through our methods. Therefore, we think it would be better not to move it to the supplementary materials. We have edited the figure and hope it is more readable.

L131. Relevant impacts?

Many thanks, we have replaced "are relevant predictors" by "have relevant impacts on the" line 115.

L182. May be appropriate title would be 'Pesticide concentrations explain a significant variation in soil biodiversity of croplands' ?

Many thanks for this suggestion; as we need to shorten the subheadings to follow journal formatting guidelines (max 40 characters), we propose following alternative title: "*Pesticides as a driver of soil biota*" line 146.

L268. What does 'for a limited set of substances'—mean?

We referred here to the limited number of tested products in other studies. We agree that this message was not clear. We have edited the sentence and now reads as follows "*While some studies reported non-target effects of fungicides on nematode, protist, and bacterial diversity, these results are based on a very limited number of products. Our results covering a much broader range of fungicides show that these compounds can affect multiple components of the soil community*". We hope that it is clearer now lines 214-217.

L 337-345. Conclusion section should include a statement on trade off. For example, what impact will it have on food security if we do not use pesticides, and how to balance competing demands of food security and environmental sustainability.

Many thanks for pointing this out. We added the suggestion as follows in the conclusions:

"Only then will we be able to assess to what extent the unintended effects of pesticides on soil organisms may compromise the ecosystem functions that underpin long-term food security. Balancing the immediate need for high crop yields with efforts to enhance environmental sustainability will require investment in sustainable pest management solutions and agro-ecological practices that support both productivity and soil health" lines 292-297.

Referee #2 (Remarks to the Author):

I appreciate the authors' efforts to address the concerns I raised. I acknowledge the significant effort that authors have invested in the "large-scale" aspect of their work, including the number of pesticides analyzed, the variety of ecosystem types, and the wide range of soil organisms. However, a large scale does not in itself constitute innovation, I remain concerned about its novelty in theory is sufficient for publication in *Nature* (I do not, by all means, intend to discount the quality of this work). The conclusions drawn — even in their revised form, such as "Pesticides altered microbial functions, including phosphorus and nitrogen cycling and suppressed beneficial taxa, including arbuscular mycorrhizal fungi and bacterivore nematodes." — are almost verified by the previous literatures (see more details in the following references).

Thank you for your constructive comments. We appreciate the recognition of the scale and scope of our study and understand the concern regarding theoretical novelty and methodological innovation.

We also thank you for the list of references given below. However, they only add more proof to the value of our study and focus on specific pesticides, specific soil biota or specific ecosystems (see below) . These studies (references) clearly offer a partial view of what is happening in the soil which is a heterogeneous environment due to its interacting components. Our study offers a much more holistic view on the effects of different pesticides residues (instead of single types as in many of the papers listed below) on the whole biodiversity community and across different ecosystem types. Only with real data obtained from such a large geographical scale covering different soil types and climates, we will be able to develop integrated approaches that make sure that the physical, chemical and biological condition of the soil is maintained.

What I think could be more significant and novel is to address functional traits using real measurements, and delineate the link between pesticides diversity and biodiversity and food web dynamics.

Although this idea is interesting, the measurement of functional traits at this scale is not currently feasible within the LUCAS Soil framework. In addition, it has been addressed before in literature reviews. See for example, the meta-analysis performed by Beaumelle et al. (2023) where they used two functional traits to test the sensitivity of soil fauna to pesticides: (i) body size by assuming that macrofauna will be more sensitive than meso- and microfauna due to longer generation time and lower

population densities and (ii) presence of exoskeleton as it might act as a protective barrier against pesticides.

In the previous reviewing round, we (1) explicitly discussed this limitation in the Discussion (lines 224-226), (2) incorporated trait-based interpretations from metagenomic functional gene profiles (lines 220-231), and (3) added text outlining how these patterns may reflect cascading effects on soil food webs (lines 224-226).

In addition to its limited novelty, this paper also does not present significant technical breakthroughs: the methods used (DNA metabarcoding, metagenomics, and GLMs) are all common tools in research related to microbial community (see more details in references). While the paper reported the adverse effects of use on the soil microbiome, it fails to comprehensively quantify these impacts. A dataset of such "large scale" is immensely valuable to propose more meaningful criteria for identifying "healthy soil microbiota" and then quantitatively synthesizing the impacts of pesticides on it. But regrettably, such a valuable dataset has resulted in a simple report rather than targeted, constructive, and policy-relevant recommendations.

See below our response

References (just for example):

Pesticides alter microbial functions

Zhang, X. et al. The fate and ecological risk of typical diamide insecticides in soil ecosystems under repeated application. *Journal of Hazardous Materials* 494, 138440, (2025). (Pesticides application inhibited core nitrogen and carbon cycling microbes)

Wu, G. et al. Impacts of organophosphate pesticide types and concentrations on aquatic bacterial communities and carbon cycling. *Journal of Hazardous Materials* 475, 134824, (2024). (Pesticides application inhibited both Calvin-Benson-Bassham cycle and Wood-Ljungdahl pathway)

Sim, J. X. F. et al. Impact of twenty pesticides on soil carbon microbial functions and community composition. *Chemosphere* 307, 135820, (2022). (Pesticides application influenced soil carbon cycling; also analyzed with considerable number of pesticides)

Sim, J. X. F. et al. Pesticide effects on nitrogen cycle related microbial functions and community composition. *Science of The Total Environment* 807, 150734, (2022). (Pesticides application inhibited nitrogen cycle)

Liu, Y.-R. et al. Soil contamination in nearby natural areas mirrors that in urban

greenspaces worldwide. *Nature Communications* 14, 1706, doi:10.1038/s41467-023-37428-6 (2023). (Multiple soil contaminants, including pesticides, influence microbial functional traits in global scale)

Pesticides suppressed beneficial taxa

Edlinger, A. et al. Agricultural management and pesticide use reduce the functioning of beneficial plant symbionts. *Nature Ecology & Evolution* 6, 1145-1154, doi:10.1038/s41559-022-01799-8 (2022). (In their own previous work, the authors have already addressed the negative impacts of pesticides on arbuscular mycorrhizal fungi. Does the present study provide any new data or insights regarding this specific topic?)

Romero, F., Jiao, S. & van der Heijden, M. G. A. Impact of microbial diversity and pesticide application on plant growth, litter decomposition and carbon substrate use. *Soil Biology and Biochemistry* 208, 109866, (2025). (In this study from the authors' research group, the negative impacts of pesticides on organic matter degradation by beneficial taxa have already been discussed. Does the current paper provide any new findings or insights regarding this topic?)

Using DNA metabarcoding, metagenomics, and GLMs in research related to microbial community

Zhang, X. et al. The fate and ecological risk of typical diamide insecticides in soil ecosystems under repeated application. *Journal of Hazardous Materials* 494, 138440, (2025). (DNA metabarcoding, metagenomics)

Wu, G. et al. Impacts of organophosphate pesticide types and concentrations on aquatic bacterial communities and carbon cycling. *Journal of Hazardous Materials* 475, 134824, (2024). (DNA metabarcoding, RT-qPCR)

Sim, J. X. F. et al. Impact of twenty pesticides on soil carbon microbial functions and community composition. *Chemosphere* 307, 135820, (2022). (DNA metabarcoding, RT-qPCR, metagenomics)

Sim, J. X. F. et al. Pesticide effects on nitrogen cycle related microbial functions and community composition. *Science of The Total Environment* 807, 150734, (2022). (DNA metabarcoding, RT-qPCR)

Liu, Y.-R. et al. Soil contamination in nearby natural areas mirrors that in urban greenspaces worldwide. *Nature Communications* 14, 1706, doi:10.1038/s41467-023-37428-6 (2023). (metagenomics)

Edlinger, A. et al. Agricultural management and pesticide use reduce the functioning of beneficial plant symbionts. *Nature Ecology & Evolution* 6, 1145-1154, doi:10.1038/s41559-022-01799-8 (2022). (DNA metabarcoding)

Romero, F., Jiao, S. & van der Heijden, M. G. A. Impact of microbial diversity and

pesticide application on plant growth, litter decomposition and carbon substrate use. *Soil Biology and Biochemistry* 208, 109866, (2025). (DNA metabarcoding)

Kang, L. et al. Metagenomic insights into microbial community structure and metabolism in alpine permafrost on the Tibetan Plateau. *Nature Communications* 15, 5920, (2024). (metagenomics, GLMs)

Yu, Z., Zeng, X.-m., Cheng, X., Zhang, Q. & Zhang, K. Patterns and Environmental Drivers of Soil Microbial Succession. *Global Change Biology* 31, e70475, (2025). (DNA metabarcoding, GLMs)

Healthy soil microbiota under pesticides application (mentioned in previous studies, but can be improved by your datasets)

Swaine, M. et al. Impact of pesticides on soil health: identification of key soil microbial indicators for ecotoxicological assessment strategies through meta-analysis. *FEMS Microbiology Ecology* 101, fiab052, (2025). (ammonia-oxidizing microorganisms as indicators of the toxicity of pesticides on soil microbiota)

Xu, N. et al. Integrating Anthropogenic–Pesticide Interactions Into a Soil Health-Microbial Index for Sustainable Agriculture at Global Scale. *Global Change Biology* 30, e17596, (2024). (propose Health-Microbial Index and analyzed the impacts of pesticides on microbiota in global scale).

Many thanks for your evaluation and considerations. We agree that this is not the first study assessing effects of pesticides on soil organisms, and others studies have been published on pesticides, including our own work. However, the work done so far and the examples/references given above, including some of our own work, (i) focus on a very limited number of pesticides (e.g., Zhang et al., focusing on two substances in an experimental setting), (ii) focus on specific groups of soil biota (e.g., Edlinger et al. focusing on arbuscular mycorrhizal fungi) or (iii) have been spatially limited by focusing on specific countries or specific ecosystem types (e.g., Liu et al. focusing on urban greenspaces and adjacent natural areas). Several of the listed studies (e.g. Liu et al., Edlinger et al., Romero et al., Sim et al.) are already included in our Introduction, together with their limitations (lines 55-57).

Additionally, we acknowledge that the methodologies applied in our study (e.g., DNA metabarcoding, metagenomics, GLMs) have been used previously (Kang et al. 2024; Yu et al. 2025). The novelty of our work does not lie in new analytical techniques, but in their integration and upscaling to the continental level, leading to a system-level generalisation of pesticide impacts across ecosystem types and soil biodiversity taxa. This pan-European field approach indeed enables us to assess pesticide impacts under real environmental and management variability, identify organism- and function-

specific response patterns across ecological contexts, and quantify the relative influence of pesticides compared to soil properties and climate. The listed studies are therefore complementary to our work which represents the first large-continental scale integration of pesticide data with multi-trophic soil biodiversity and functional potential.

Finally, the large spatial coverage of our dataset and the inferred responses of soil biodiversity and pesticide residues provide the much needed baseline to guide the future policies and regulations defining meaningful criteria for maintaining healthy soils (and their microbiomes) across ecosystem types, climatic zones, and soil types in Europe. The LUCAS Soil Pesticides dataset is unique in unravelling continental scale patterns and identifying knowledge gaps that will steer future work. We strongly believe that, on this basis, our article constitutes an important, novel contribution.

Referee #4 (Remarks to the Author):

The authors have majorly revised their manuscript on the impact of pesticides on soil biodiversity, resulting in a more accessible paper, a better exposure of its insistent message with clear account of how it is based on the impressive European-wide database and series of analyses.

Many thanks for this positive feedback.

Our comments were well implemented, although I have a few minor additional suggestions. In particular

- The new conceptual Fig 1 is very welcome, giving overview of the analyses, their hierarchy and interrelationships. However, better match the description in the text (ll.84-96) with the figure. It is confusing to have 4 objectives (i – iv) in the text and 3 in the figure. Explain in the text why the complementary analyses (and refer to them as such) are needed.

Many thanks for pointing this out. We have re-drawn Figure 1 to improve its clarity and edited the caption to explain the complementary analyses. In addition, we are describing the three objectives in the text (lines 85-96) to match the ones in the figure.

•Analyses of cropland vs all ecosystems are now better in balance

Many thanks for this positive comment.

•The role of land use intensification and other factors are now appropriately discussed (ll 294-303), but I miss a note in the section on future research work. Establishing threshold values for pesticide impacts is an important research avenue but will only be meaningful if combined with effects of land use intensity measures.

Many thanks for pointing this out. We agree that land use intensity measures are indeed an important research avenue and added the note to the future research work as follows (lines 278-284):

"Future research should focus on: (i) systematically integrating soils adjacent to cropland areas to better assess the broader impacts of pesticide contamination and (ii) establishing threshold values for pesticide impacts on soil biodiversity across land use intensity gradients, soil types and climates⁴⁰, to support evidence-based updates to current pesticide approval and monitoring frameworks (see also Supplementary Discussion on robust soil biodiversity/pesticide data baseline)".

- Several analyses moved to Suppl (Correlation analysis; diversity measures), SEM omitted, making the paper more straightforward.

Many thanks for this positive comment.